# Impact of chlorine ion chemistry on ozone loss in the middle atmosphere during very large solar proton events

Monali Borthakur[1], Miriam Sinnhuber[1], Alexandra Laeng[1], Thomas Reddmann[1], Peter Braesicke[1], Gabriele Stiller[1], Thomas von Clarmann[1], Bernd Funke[2], Ilya Usoskin[3], Jan Maik Wissing[4], and Olesya Yakovchuk[4]

[1]Institute of Meteorology and Climate research, Karlsruhe Institute of Technology, Karlsruhe, Germany
[2]Instituto de Astrofísica de Andalucía, CSIC, Granada, Spain
[3]University of Oulu, Oulu, Finland
[4]University of Rostock, Rostock, Germany

**Correspondence:** Monali Borthakur (monali.borthakur@kit.edu)

**Abstract.**

Solar coronal mass ejections can accelerate charged particles, mostly protons, to high energies, causing Solar Proton Events (SPEs). Such energetic particles can precipitate upon the Earth's atmosphere, mostly in polar regions because of the geomagnetic shielding. Here, SPE induced chlorine activation due to ion-chemistry can occur and the activated chlorine depletes ozone in the polar middle atmosphere. We use state of the art 1D stacked-box Exoplanetary Terrestrial Ion Chemistry model (ExoTIC), of atmospheric ion and neutral composition to investigate such events in the Northern Hemisphere (NH). The Halloween SPE that occurred in late October 2003 is used as a test field for our study. This event has been extensively studied before using different 3D models and satellite observations. Our main purpose is to use such a large event that has been recorded by MIPAS on ENVISAT to evaluate the performance of the ion-chemistry model. Sensitivity tests were carried out for different model settings with a focus on the chlorine species of HOCl and ClONO$_2$ as well as O$_3$ and reactive nitrogen, NO$_y$. The model simulations were performed in the Northern Hemisphere at a high latitude of 67.5°N, inside the polar cap. Comparison of the simulated effects against MIPAS observations for the Halloween SPE revealed a rather good temporal agreement, also in terms of altitude range for HOCl, O$_3$ and NO$_y$. For ClONO$_2$, a good agreement was found in terms of altitude range. The model showed ClONO$_2$ enhancements after the peak of the event. The best model setting was the one with full ion-chemistry where O($^1$D) was set to photo-chemical equilibrium. HOCl and ozone changes are very well reproduced by the model, specially for nighttime. HOCl was found to be the main active chlorine species under nighttime conditions resulting in an increase of more than 0.2 ppbv. Further, ClONO$_2$ enhancements of 0.2-0.3 ppbv have been observed both during daytime and nighttime. Model settings that compared best with MIPAS observations were applied to an extreme solar event that occurred in 775 A.D., presumably once in a 1000 year event. With the model applied to this scenario, assessment can be made what is to be expected at worst for effects of a SPE on the middle atmosphere concentrating on effects of ion-chemistry compared to crude parameterisations. Here, a systematic analysis comparing the impact of the Halloween SPE and the extreme event on the Earth's middle atmosphere is presented. As seen from the model simulations, both events were able to perturb the polar stratosphere and mesosphere, with a high production of NO$_y$ and HO$_x$. Longer lasting and stronger stratospheric ozone loss was seen for

the extreme event. Qualitative difference between the two events and a long lasting impact on HOCl and HCl for the extreme event was found. Chlorine ion-chemistry contributed to a stratospheric ozone loss of 2.4% for daytime and 10% for nighttime during the Halloween SPE as seen with time dependent ionisation rates applied to the model. Furthermore, while comparing the Halloween SPE and the extreme scenario, with ionisation rate profiles applied just for the event day, the inclusion of chlorine ion-chemistry added an ozone loss of 10% and 20% respectively.

## 1 Introduction

High energy particles (e.g. electrons and protons) that precipitate at high latitudes can alter the chemical composition of the atmosphere by different photo-chemical reactions. This mainly happens due to primary collision processes and subsequent ion and neutral chemistry reactions. Such reactions ordered by increasing energy are, for example, excitation, photo-dissociation, photo-ionisation and dissociative ionisation. These particles can come from various sources in outer space, accelerated by different processes to different energies. They affect different altitude ranges of the atmosphere. Such sources are, for example, galactic cosmic rays (GCRs), with protons and heavier nuclei of energies ranging from hundreds of MeV to GeV; coronal mass ejections and SPEs with protons of energies from MeV to GeV; auroral electrons during substorms accelerated to energies from 10 keV to hundreds of keV; and medium and high energy electrons in the radiation belts to energies from tens of keV into the MeV range. This study involves SPEs which can also induce geomagnetic disturbances in the Earth's magnetosphere leading to energetic electron precipitation (EEP) events. Recent studies, such as Verronen et al. (2005), that studied energetic particle precipitation events (EPP) found significant co-variability in mesospheric ozone with proton and electron fluxes. $HNO_3$ increases measured during SPEs cannot be reproduced using the standard parameterization of $HO_x$ and $NO_x$ production, while models considering D-region ion-chemistry in detail agree with the observations (Verronen et al., 2016). This finding highlighted the need to improve ion chemistry modeling in the D-region for altitudes below 90 km in the ionosphere (Funke et al., 2011) to capture the EPP ozone interaction. The $HO_x$ and $NO_x$ parameterization cannot reproduce the longer-term effects of ion-chemistry on for example, reactive nitrogen partitioning, ozone, and dynamics of the middle atmosphere (Kvissel et al., 2012).

In the case of parameterized $NO_x$, NO is produced when $N(^2D)$ reacts with molecular oxygen, shown in Reaction R1 (Porter et al., 1976; Jackman et al., 2005). This reaction is a major source of NO in the stratosphere, mesosphere and lower thermosphere (Rusch et al., 1981; Barth, 1992).

$$N(^2D) + O_2 \longrightarrow NO + O(^3P, ^1D) \tag{R1}$$

NO can be destroyed by Reaction R2, which is an effective loss mechanism for $NO_x$, also known as the scavenging reaction (Jackman et al., 2005):

$$N(^4S) + NO \longrightarrow N_2 + O \tag{R2}$$

$O_2$ can also react with the ground state $N(^4S)$, that is temperature dependent and is a major source of NO in the thermosphere above $\sim 120$ km (Sinnhuber et al., 2012) (Barth, 1992). The excited states of N form NO, while ground state can destroy NO

which is relevant in the stratosphere, mesosphere and lower thermosphere. The partitioning between the ground and the excited states determine the amount of $NO_x$ formed (Sinnhuber et al., 2012; Nieder et al., 2014) thereby making Reaction R2 the driver of parameterised $NO_x$. Thus, Reaction R2 makes the difference between full ion-chemistry and parameterised $NO_x$ formation. The main processes responsible for the odd hydrogen ($HO_x$= H,OH,$HO_2$) formation during energetic particle precipitation events, along with the ion-chemistry processes leading to its release were considered by Solomon et al. (1981). They take place after the initial formation of ion pairs. Solomon et al. (1981) considered the ion-chemistry processes leading to a release of $HO_x$ during energetic particle precipitation events. They found that the main process responsible is the uptake of water vapour into large cluster ions and the subsequent release of H during recombination reactions of these cluster ions. Large cluster ions can then be formed by reaction pathways like (Sinnhuber et al., 2012):

$$O_2^+(H_2O) + H_2O \longrightarrow H_3O^+(OH) + O_2 \tag{R3}$$

$$O_2^+(H_2O) + H_2O \longrightarrow H^+(H_2O) + OH + O_2 \tag{R4}$$

These protonised water cluster ions can then recombine with electrons to form H and OH.

$$H_3O^+(OH) + e^- \longrightarrow H_2O + H + OH \tag{R5}$$

Hydrogen and nitrogen radicals lead to ozone destruction through catalytic cycles in the stratosphere and mesosphere. Different studies found ozone depletion in the mesosphere during SPEs, for example, Weeks et al. (1972) who studied a large polar cap absorption event in 1969 that was explained as a result of the formation of odd hydrogen (Swider and Keneshea, 1973).

## 1.1 Hydrogen catalytic cycles

Catalytic cycles involving $HO_2$ are very important in the lower stratosphere (10-30 km). The fastest of these cycles is shown in Reactions R6, R7 and R8.

$$OH + O_3 \longrightarrow HO_2 + O_2 \tag{R6}$$
$$HO_2 + O \longrightarrow OH + O_2 \tag{R7}$$
$$Net: O_3 + O \longrightarrow O_2 + O_2 \tag{R8}$$

Another example of $HO_x$ catalytic destruction cycles that is important in the middle and upper mesosphere (above 60 km), is shown in Reactions R9 and R10 (Bates and Nicolet, 1950). In every chain of Reactions R9 and R10, one molecule of $O_3$, $O(^3P)$ or $O(^1D)$ is lost while reforming H and OH and thereby producing a net ozone loss (Reaction R11).

$$H + O_3 \longrightarrow OH + O_2 \tag{R9}$$
$$OH + O \longrightarrow H + O_2 \tag{R10}$$
$$Net: O_3 + O \longrightarrow O_2 + O_2 \tag{R11}$$

## 1.2 Nitrogen catalytic cycle

In the lower stratosphere, ozone loss is mainly due to the catalytic cycle with $NO_x$ governed by the Reactions R12 and R13 in which case the loss of ozone is more persistent due to the longer lifetimes of $NO_x$.

$$NO + O_3 \longrightarrow NO_2 + O_2 \tag{R12}$$

$$NO_2 + O \longrightarrow NO + O_2 \tag{R13}$$

$$Net : O_3 + O \longrightarrow O_2 + O_2 \tag{R14}$$

## 1.3 Chlorine catalytic cycles

The focus of this paper is the impact of charged chlorine species during a SPE. Negative chlorine species constitute a significant part of the total anions in the mesosphere (Chakrabarty and Ganguly, 1989; Fritzenwallner and Kopp, 1998). The chlorine negative ion is an abundant ion of the lower D region during daytime and nighttime. Other D region negative ions like $O_2^-$, $O^-$, $CO_3^-$, $OH^-$, $NO_2^-$ and $NO_3^-$ can react with HCl to produce $Cl^-$ which forms $Cl^-(X)$, where X = (HCl, $H_2O$, $CO_2$ and $HO_2$) (Kopp and Fritzenwallner, 1997). $Cl^-$ and $Cl^-(H_2O)$ are the most abundant chlorine ions in the mesosphere as indicated by previous studies for e.g., Chakrabarty and Ganguly (1989), Fritzenwallner and Kopp (1998) and Turco (1977). Both species can react with atomic hydrogen re-releasing HCl and some of the recombination reactions of negative chlorine species with positive ions like $H^+$ release Cl, ClO, $ClNO_2$ and $Cl_2$. Since the ion reactions are faster, the SPE impacts due to chlorine ion-chemistry are expected to occur without any notable delay. The reactions involving charged and uncharged chlorine species along with the reaction rate coefficients are given in Table A1.

Apart from the $NO_x$ and $HO_x$ catalytic cycles, solar proton events can also affect stratospheric chlorine chemistry, but whether solar protons effectively activate or deactivate chlorine depends on illumination conditions. The ion production rates increase during a SPE and influence the chemistry of both charged and uncharged chlorine species. The neutral compounds of chlorine can then contribute to ozone loss. The chlorine catalytic cycles of ozone destruction are very efficient around 40 km (Lary, 1997). SPE induced changes of chlorine species can contribute to the short-term ozone depletion occurring after the SPE (von Clarmann et al., 2005). This influence is indirect and is mainly caused by $NO_x$ and $HO_x$ enhancements. The $ClO_x$ ozone loss catalytic cycle, where ClO photolyses:

$$ClO + O \longrightarrow Cl + O_2 \tag{R15}$$

$$Cl + O_3 \longrightarrow ClO + O_2 \tag{R16}$$

$$Net : O_3 + O \longrightarrow O_2 + O_2 \tag{R17}$$

is the main cycle responsible for ozone loss in the middle and upper stratosphere between 40 and 50 km (Daniel et al., 1999). O is formed by the photolysis of $O_2$ and $O_3$ and is available during daytime. The catalytic cycle involving hypochlorous acid (HOCl) and ClO acts as a link between chlorine and $HO_x$ enhancements as a result of the SPEs (Reactions R19, R20, R21, R22 and R23) (Lary, 1997). HOCl can photolyse during daytime and the OH formed can react with $O_3$ reforming $HO_2$ and Cl

reforming ClO thereby recycling HOCl again through Reaction R19. This cycle mainly plays a role in the sunlit polar lower
stratosphere (Lary, 1997). Reactive Cl can also be formed via reaction of OH with HCl.

$$OH + HCl \longrightarrow H_2O + Cl \tag{R18}$$

$$ClO + HO_2 \longrightarrow HOCl + O_2 \tag{R19}$$

$$HOCl + h\nu \longrightarrow Cl + OH \tag{R20}$$

$$Cl + O_3 \longrightarrow ClO + O_2 \tag{R21}$$

$$OH + O_3 \longrightarrow HO_2 + O_2 \tag{R22}$$

$$Net : 2O_3 \longrightarrow 3O_2 \tag{R23}$$

von Clarmann et al. (2005) showed an enhancement of chlorine monoxide, ClO and HOCl immediately after the SPE. They
concluded that this was due to the Reactions R21 and R19. During an SPE, HOCl and reactive Cl present in the stratosphere
can react with OH and $HO_2$ respectively, to form ClO. Other reactions of Cl with $HO_2$ and $H_2O_2$ can yield in the production
of HCl, which is the most important stratospheric reservoir species of Cl. The Reactions R24, R25, R26 and R27 are relevant
to the study in Sect. 4.

$$HOCl + OH \longrightarrow ClO + H_2O \tag{R24}$$

$$Cl + HO_2 \longrightarrow ClO + OH \tag{R25}$$

$$Cl + HO_2 \longrightarrow HCl + O_2 \tag{R26}$$

$$Cl + H_2O_2 \longrightarrow HCl + HO_2 \tag{R27}$$

This is another effective ozone loss cycle involving SPE induced $NO_x$ enhancements between 15 and 40 km (Lary, 1997).
ClO can react with nitric oxide (Reactions R28, R29, R30 and R31), that is most important in the 15 to 50 km altitude range
as suggested by J. C. Farman and Shanklin (1985). $NO_x$ enhancements are also essential regarding the production of $ClONO_2$.
López-Puertas et al. (2005) and von Clarmann et al. (2005) reported the first experimental confirmation of Reaction R32 under
SPE conditions. ClO reacts with nitrogen dioxide, $NO_2$ which is most efficient in the lower stratosphere, forming $ClONO_2$
(Reactions R32, R33, R34, R35 and R36).

$$ClO + NO \longrightarrow Cl + NO_2 \tag{R28}$$

$$NO_2 + O(^3P) \longrightarrow NO + O_2 \tag{R29}$$

$$Cl + O_3 \longrightarrow ClO + O_2 \tag{R30}$$

$$Net : O(^3P) + O_3 \longrightarrow 2O_2 \tag{R31}$$

$$\text{ClO} + \text{NO}_2 \longrightarrow \text{ClONO}_2 \tag{R32}$$

$$\text{ClONO}_2 + \text{h}\nu \longrightarrow \text{Cl} + \text{NO}_3 \tag{R33}$$

$$\text{NO}_3 + \text{h}\nu \longrightarrow \text{NO} + \text{O}_2 \tag{R34}$$

$$\text{Cl} + \text{O}_3 \longrightarrow \text{ClO} + \text{O}_2 \tag{R35}$$

$$\text{Net} : \text{O}(^3\text{P}) + \text{O}_3 \longrightarrow 2\text{O}_2 \tag{R36}$$

The present paper deals with changes of HOCl, ClONO$_2$, ozone and NO$_y$ occurring in the Northern Hemisphere at a high latitude of 67.5°N during the Halloween SPE from mid October to early November 2003, peaking around October 28-29. The Halloween SPE was one of the largest SPEs in the satellite era and consisted of a series of solar flares and coronal

mass ejections. Such large events mainly occur in the declining phase of the solar maximum. Changes in the composition of HOCl, ClONO$_2$, ozone and NO$_y$ species during the Halloween SPE have been previously reported in Funke et al. (2011) and Jackman et al. (2008). Funke et al. (2011) used different models to investigate the SPE induced changes and Jackman et al. (2008) used version 3 of the Whole Atmosphere Community Climate Model (WACCM). Both studies compared with the MIPAS observations from polar orbit satellite ENVISAT. Damiani et al. (2012) also looked at chlorine species (i.e., HOCl,

ClONO$_2$, ClO and HCl) using MLS and MIPAS data and version 4 of the WACCM model during SPEs of 17 and 20 January 2005. However they did not consider the D region ion-chemistry. Here, we studied the temporal evolution of changes of the respective chemical constituents considering the D region ion-chemistry in the 1D stacked box model, Exoplanetary Terrestrial Ion Chemistry (ExoTIC). The ion-chemistry was implemented by Winkler et al. (2009) upon which ExoTIC is based.

In order to have a better comparison of ExoTIC simulations with MIPAS observations, we ensured that they are sampled

inside the polar vortex. The polar vortex is a large circumpolar cyclone that is formed due to decreased solar insolation in the polar winter stratosphere as a manifestation of a strong meridional temperature gradient caused by a lack of high-latitude solar heating during polar night and dominates the dynamics (Harvey et al., 2015). The assumption is that the air inside the polar vortex is horizontally well mixed and separated from air masses outside the vortex. That allows us to simulate it in a 1D vertical model. The ionisation during particle precipitation in the polar cap is also assumed to be inside the polar vortex where NO$_x$ is

conserved which makes it better comparable to the 1D model. The procedure is described in detail in the Sect. 3.1.

The Halloween SPE is later compared with an exceptionally strong cosmic ray event that occurred in 775 A.D. It was derived from the historical records in radiocarbon $^{14}$C measured in tree ring archives and later confirmed by $^{10}$Be and $^{36}$Cl cosmogenic nuclides. Although various scenarios were initially proposed, it is concluded now that the event was caused by solar energetic particles (Sukhodolov et al., 2017). $^{10}$Be and $^{14}$C implied that the event had a very hard spectrum and thereby very high

energetic protons. It is the strongest solar energetic particle storm known for the last 11 millennia (the Holocene), serving as a likely worst-case scenario being 40–50 times stronger than the largest directly observed event on 23$^{\text{rd}}$ February 1956 (Usoskin et al., 2013). This event was transient, as estimated using the ratio of different cosmogenic isotopes (Mekhaldi et al., 2015).

This paper is organised as follows. Sect. 2 describes the ionisation rates, model framework, simulations and the satellite observations used to evaluate the model. Sect. 3 presents the results of the model evaluation with MIPAS satellite observations.

An overview of the changes in chlorine species, ozone and NO$_y$ induced by the SPE is presented. Sect. 4 presents a case study

comparing model simulations of the Halloween SPE with the extreme solar event. Sect. 5 shows some results describing the impact of chlorine ion-chemistry on ozone loss. In sect. 6, a conclusion is provided to check if the data is well understood, a summary of how our results compare to previous studies. Finally an assessment is given how further studies could improve our current knowledge on SPE induced ozone loss due to chlorine ion-chemistry.

## 2  Data and methods

### 2.1  Ionisation rates

The ionisation rates (IRs) used for the Halloween SPE were obtained from the Atmospheric Ionisation during Substorm (AIS-storm) model which is an enhanced version of the Atmospheric Ionisation Module Osnabrück (AIMOS) model (Wissing and Kallenrode, 2009). The AIMOS model computes ionisation rates by precipitating electrons, protons and alpha particles for the whole atmosphere based on particle flux measurements from Polar Operational Environmental Satellites (POES), the Meteorological Operational satellites (Metop) and the Geostationary Operational Environmental Satellites (GOES). The treatment of the electron fluxes is in the energy range (0.154–300 keV), protons with an energy range of 0.154 eV to 500 MeV. In the AIMOS v2.0-AISstorm model, both the time resolution (0.5 hr) and spatial resolution has been improved compared to AIMOS. For a comparison of the model results with MIPAS observations, the time dependent ionisation rates were put into ExoTIC. Figure 1 on the left side shows the temporal evolution of ion pair production rates for protons, electrons and alpha particles varying over the time period, 25$^{th}$ October to 4$^{th}$ November 2003 from the AISstorm model. These IRs are averaged over the longitudes for the latitude of 67.5°N, in the polar cap region and are also daily averaged. For the extreme event, integrated ionisation rates were taken from an extreme SPE of 23 February 1956 (SPE 56) (Meyer et al., 1956), which was the strongest observed event with ground-level enhancement (GLE) > 4000 %. Cliver et al. (2022) estimated this factor 70 × particle fluence compared to the 1956 event and the ionisation rates were scaled accordingly. This factor was a rough estimate to scale the fluxes of particles and excess radiation such that the energy spectrum of SPE 56 was comparable to the isotope signals of the extreme event. Figure 1 on the right shows the ionisation rate profiles for both the events. The profiles for the Halloween SPE show average IRs for October 27 (day 301) and October 28 (day 302) before the SPE (in blue) and the average IRs for October 28 and October 29 during the main SPE phase (in green). It can be observed that the ionisation rates for the stratosphere and lower mesosphere in case of the extreme event is about 1-2 orders of magnitude higher compared to the Halloween SPE main phase. This is because the extreme event contained protons of energies up to a few GeV, compared to about only a few MeV protons for the Halloween SPE, the ionisation rates for the same can be seen to reach much further down to the surface.

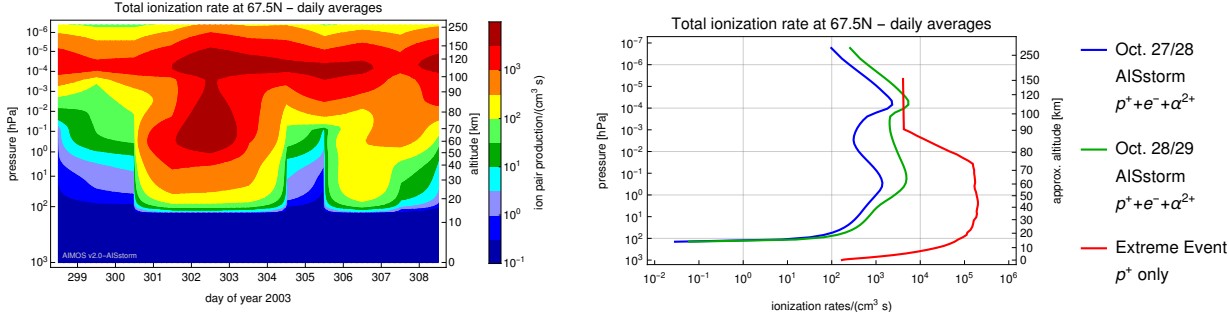

**Figure 1.** Time dependent and daily averaged ionisation rates (IRs) from $25^{\text{th}}$ October to $4^{\text{th}}$ November 2003 obtained from the AISstorm model for the latitude of $67.5°$N (left); Mean ionisation rates (IRs) before the SPE (Oct 27-28), and in SPE main phase (Oct 28-29) obtained from AISstorm and mean IRs of the extreme event for the same latitude of $67.5°$N (right).

## 2.2 Description of the 1D model and Experiments

The Exoplanetary Terrestrial Ion Chemistry model ExoTIC is a 1-dimensional stacked-box model of the atmospheric neutral
and ion composition (Herbst et al., 2022). The ion-chemistry is based on the UBIC (University of Bremen ion-chemistry) model developed by Winkler et al. (2009) for the terrestrial middle atmosphere. The neutral chemistry is based on the SLIMCAT model by Chipperfield (1999). It accounts for photo-ionisation of NO by Lyman-$\alpha$ radiation, photo-dissociation of charged species and photo-detachment of electrons but doesn't contain any diffusion or horizontal and vertical transport. It first simulates a neutral atmosphere and contains the time evolution of 106 charged and 58 neutral species that interact due to neutral, neutral–ion,
and ion–ion gas-phase reactions, as well as photolysis and photo-electron attachment and detachment reactions (Sinnhuber et al., 2012). The ExoTIC model extends the applicability of UBIC to atmospheres of (rocky) planets other than Earth with a wide range of orbital parameters, stellar systems and base compositions as discussed by Herbst, Konstantin et al. (2019). More neutral species have been added to the ion-chemistry in ExoTIC since studies by Winkler et al. (2009). The ionisation of $CO_2$ was recently included. Another small change is that the equilibrium is calculated for the ions, which stabilised the model. The
model contains boxes of 2.7 km each in height. For the studies performed here, the background atmosphere was taken from the EMAC (ECHAM/MESSy) atmospheric chemistry climate model with T42 horizontal truncation. It has 74 levels in the vertical direction and covers an altitude up to 200-220 km, depending on latitude and season, with a vertical resolution of 2.7 km. The neutral chemistry and the ion-chemistry model are calculated iteratively as follows:

1. The neutral model is time dependent and calculates the volume mixing ratios of the neutral species with a variable time
step and feeds them to the ion-chemistry model.

2. The ion-chemistry in the equilibrium state is calculated, calling it hourly from the neutral model. The highest level for which the ion-chemistry is calculated is 1 (207.4 km) and the lowest level is 53 (25.4 km), which depends on the initialisation.

3. The net effective production or loss rates of neutral species due to primary ionisation, positive and negative ion-chemistry which can also be used as a parameterisation for global chemistry-climate models (Nieder et al., 2014), are computed using an iterative chemical equilibrium approach.

4. The production rates resulting from the ion-chemistry computation are then fed back to the 1-D neutral chemistry model, which solves for the neutral atmospheric state transiently using the net effective production/loss reactions as well as neutral photo-chemistry reactions.

5. Lastly, this state is again returned to the ion-chemistry model for the following computation.

The model settings used for the sensitivity studies were mainly variations of full ion-chemistry containing both positive and negative ions from the D-region: setting reactive $O(^1D)$ in photo-chemical equilibrium and switching off the chlorine ion-chemistry. Parameterised $NO_x$ and $HO_x$ model simulations based on Porter et al. (1976) and Solomon et al. (1981) were also carried out to assess the performance of the ion-chemistry model.

## 2.2.1 Ion-chemistry

The ionisation in this case is driven by prescribed ionisation rates and by photo-ionisation of NO, with the primary positive charges being distributed onto $N_2$, N, $O_2$ and O and balanced with electrons (Sinnhuber et al., 2012). These rates of the primary ions are calculated by ionisation cross-sections based on Rusch et al. (1981) and Jones and Rees (1973). All of the processes like dissociation and dissociative ionisation of $O_2$ and $N_2$ as well as ionisation of $O_2$, $N_2$ and O can form the excited states of N, O, $N_2^+$, $N^+$, $NO^+$ which are also included in the model. More details with a full list of the reactions, reactions rates and references for the reactions rates used for the positive ion-chemistry can be found in Sinnhuber et al. (2012) and the newer versions in Herbst et al. (2022).

The simulation results are sensitive to the changes of the $O(^1D)$ branching ratio, $\beta = \Delta O(^1D)/(\Delta O(^1D) + \Delta O(^3P))$ which is discussed in Winkler et al. (2009). Winkler et al. (2011) has reported the $O(^1D)$ corrections in the UBIC model in more detail. $O(^1D)$ is a reactive species which is formed from the dissociation of $O_2$ and $CO_2$ by particle impact ionisation but also in the ion-chemistry reactions itself. It generally goes into photo-chemical equilibrium but that is not considered in the ion-chemistry part of the model, so the rate of $O(^1D)$ formation passed to the neutral chemistry is too large. There are a few other short lived neutral species in the ion-chemistry model, like the excited states of N, for example, $N(^2D)$, that are treated like ions. Basically $O(^1D)$ is produced through the Reaction R1. Since $O(^1D)$ is short lived mostly but not considered to go into equilibrium in the ion-chemistry stage, a large rate of formation of $O(^1D)$ is produced and added to the neutral chemistry. The time constants for the quenching $O(^1D) + M \longrightarrow O(^3P) + M$ in the stratosphere are significantly smaller than the chosen integration time step in the ion-chemistry model which was also reported in Winkler et al. (2011). That causes too high $O(^1D)$ concentrations and an unrealistically strong effect through reaction R37. $O(^1D)$ can react with species like $H_2O$, $H_2$ and $CH_4$ in the lower stratosphere and also with HCl in the stratosphere to produce OH. Therefore, setting either the formation rates of $O(^1D)$ to zero or calculating it in photo-chemical equilibrium can make a difference to the full ion-chemistry through Reactions R37, R38, R39 and R40.

$$O(^1D) + H_2O \longrightarrow 2OH \tag{R37}$$

$$O(^1D) + H_2 \longrightarrow OH + H \tag{R38}$$

$$O(^1D) + CH_4 \longrightarrow CH_3 + OH \tag{R39}$$

$$O(^1D) + HCl \longrightarrow OH + Cl \tag{R40}$$

### 2.2.2 Sensitivity tests switching off the chlorine ion-chemistry

The purpose of this sensitivity test is to study the impact of the chlorine ion. We wanted to study what difference it makes to the full ion-chemistry with a focus on the ozone loss. This is done by switching off the reactions of negative chlorine ions with neutrals or the recombination reactions with $H^+$ in ExoTIC. The relevant reactions are given in Table A1.

### 2.2.3 Parameterised $NO_x$ and $HO_x$

The assumption in case of parameterised $NO_x$ is that 1.25 N atoms are produced per ion pair when electrically charged particles collide and dissociate $N_2$. This process produces $N_2^+$ and $NO^+$ ions and, finally, atomic nitrogen. The latter is produced in its ground state $N(^4S)$ (45 % or 0.55 per ion pair) and the excited state $N(^2D)$ (55 % or 0.7 per ion pair). These values are mostly used in stratospheric and mesospheric models. In case of $HO_x$, each ion pair typically results in the production of around two $HO_x$ constituents, i.e. a pair of H and OH per ion pair during recombination of the protonised water cluster ions in the upper stratosphere and lower mesosphere (Reaction R5) which was first estimated by Swider and Keneshea (1973). Andersson et al. (2016) calculated the parameterised $HO_x$ production using a fixed $H_2O$ profile. In contrast, ExoTIC assumes a zero abundance of water vapour above 80 km, while below, water vapour is modelled as a pair of H and OH. The $HO_x$ formation stops when there is no more water vapour. For high water vapour, two $HO_x$ per ion pair are formed, but the rate decreases with decreasing $H_2O$. As the $H_2O$ profile used in Andersson et al. (2016) decreases strongly above 80 km, the rate of $HO_x$ production also goes to zero above 80 km. Jackman et al. (2005) also considered this and an ion pair is computed to produce less than two $HO_x$ constituents per ion pair in the middle and upper mesosphere. In ExoTIC, 2 $HO_x$ are produced per ion pair everywhere, but the production of $HO_x$ is balanced by loss of water vapour, and the production therefore stops when all water vapour is consumed, effectively also reducing the amount of $HO_x$ production in regions of low water vapor. We choose 2 $HO_x$ per ion pair because we want to mainly concentrate on middle mesosphere to stratosphere and not upper mesosphere, i.e. above 80 km.

## 2.3 MIPAS on ENVISAT

The Michelson Interferometer for Passive Atmospheric Sounding (MIPAS) was a Fourier transform spectrometer for the detection of mid-infrared limb emission spectra in the middle and upper atmosphere on the ENVISAT (Environmental Satellite, 2000) mission (Fischer et al., 2008). ENVISAT was launched in 2002 into a sun-synchronous polar orbit (800 km) and stopped operation in April 2012. The atmospheric spectra were inverted into vertical profiles of atmospheric pressure, temperature and volume mixing ratios (vmrs) of more than 30 trace constituents. MIPAS observed a spectral range of 4.15 $\mu$m to 14.6 $\mu$m with a high spectral resolution, where a wide variety of trace gases have absorption lines and signals that are generally higher than

in other parts of the spectrum. This is because the Planck function maximises at about 10 $\mu$m for atmospheric temperatures. The measurement strategy of the MIPAS instrument was based on trace gases having characteristic emission and absorption lines, represented by their absorption coefficients, which are unambiguous "fingerprints" of the particular trace gases. The MIPAS mission is separated into two phases, caused by a malfunction of the instrument around March 2004. The first phase of the mission (2002–2004) is usually referred to as the MIPAS full-resolution (FR) period. After the malfunction, operation was resumed with a reduced optical path difference, resulting in deteriorated spectral resolution. The second phase starting in January 2005 is called the reduced-resolution (RR) period. Because of the long optical path through the atmospheric layers, MIPAS could also detect trace gases with very low mixing ratios. Vertical information was gained by scanning the atmosphere at different elevation angles with different tangent altitudes. MIPAS could observe atmospheric parameters in the altitude range from 5 to 68 km nominally with minimum and maximum vertical steps of 1 and 8 km respectively. The MIPAS data are used here for evaluation of the model results with different parameterisations. Data presented here are IMK version V5 data for HOCl, ozone, ClONO$_2$ and NO$_y$ species (NO, NO$_2$, HNO$_3$, N$_2$O$_5$) that are updates of those published by von Clarmann et al. (2006), von Clarmann et al. (2012), Glatthor et al. (2006), Höpfner et al. (2007a) and Funke et al. (2005).

### 2.3.1 Averaging Kernels

Different vertical resolutions of the MIPAS observations and the model need to be accounted for a meaningful comparison. The ExoTIC model has a vertical resolution of 2.7 km whereas MIPAS has different vertical resolutions for different species. For example, in case of HOCl, the maximum vertical resolution can be 17 km and for ClONO$_2$, it can be 13 km at an altitude of 40 km and above as seen from an example Figure 2 for a specific time point.

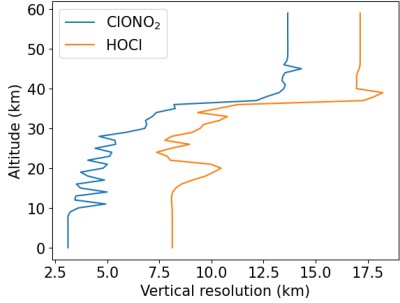

**Figure 2.** Example of typical profiles for the vertical resolution of HOCl and ClONO$_2$

To remove the discrepancy of different vertical resolutions between the model and MIPAS observations, the original model profiles have to be convolved and adjusted to the MIPAS altitude resolution. This adjustment procedure yields new species profiles that MIPAS would see if it were to sound the model atmosphere. For this purpose, we make use of the averaging kernels (Rodgers, 2000) and use a scheme suggested by Connor et al. (1994) to adjust the better resolved model profiles to those of MIPAS and the new adjusted model profiles $x_{new}$ are calculated as:

$$x_{new} = \mathbf{A}x_{orig} + (\mathbf{I} - \mathbf{A})x_a \tag{1}$$

where **A** is the MIPAS averaging kernel matrix, $x_{\mathrm{orig}}$ is the original model profile, **I** is a unity matrix and $x_{\mathrm{a}}$ is the a priori information used in the MIPAS retrievels. The rows of the averaging kernel matrix give the contribution of the true values to the retrieved values and the columns give the response of the delta peak like perturbations at each altitude. Figure 3 shows an example of averaging kernels for the different species and for a profile retrieved from spectra measured at latitude of 67.5°N on 27 October 2003 at 00:00 UT. From the figure, it is seen that the trace gas retrievals result in different sensitivities at different altitudes. For example, the maximum sensitivity is seen at 20 km for HOCl in this specific case, whereas for ClONO$_2$, it is around 15 km. To characterize the vertical resolution, the typical measures are either the full width at half maximum of the rows of the **A** or the gridwidth divided by the respective diagonal of **A** (Rodgers, 2000).

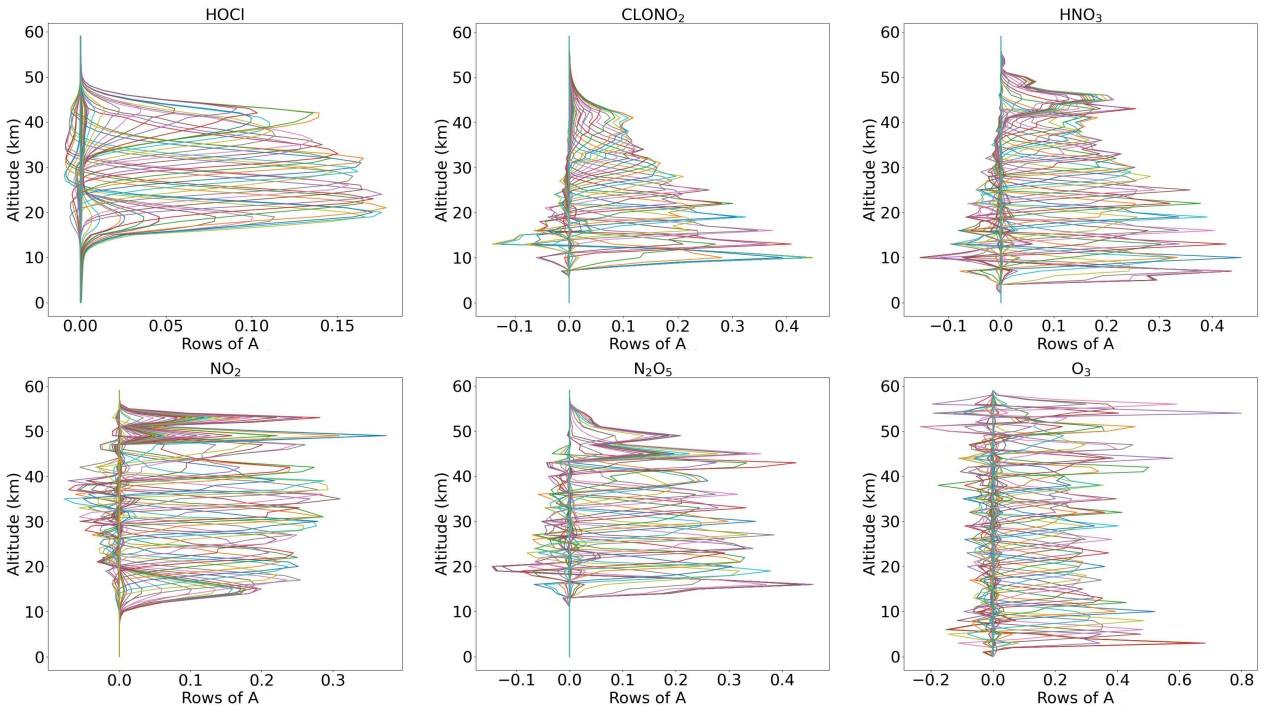

**Figure 3.** Example of rows of Averaging Kernels (**A**) for typical MIPAS HOCl, ClONO$_2$, HNO$_3$, NO$_2$, N$_2$O$_5$ and O$_3$ at a latitude of 67.5°N on 27 October 2003 at 00:00 UT.

## 3 Comparison of ExoTIC simulations to MIPAS observations for the Halloween SPE 2003

In this section, a comparison study between the ExoTIC model results and MIPAS observations has been carried out for the chlorine species of HOCl, ClONO$_2$, ozone and odd oxides of nitrogen (NO$_y$) for the Halloween SPE 2003. The comparison is done for the model simulations with different settings of ion-chemistry, i.e. calculating the photo-chemical equilibrium of O($^1$D) and switching off the uptake of chlorine ions, and parameterised NO$_x$ and HO$_x$. The model simulations are performed for a high latitude of 67.5°N and the MIPAS data were taken for the polar cap region, averaged over geographic latitudes such

that it's inside the vortex, either vortex core or vortex edge depending on the tracer properties. The model data is sampled in the MIPAS altitude grid as well. The day and night for the MIPAS data are sorted according to the solar zenith angle (day $\leq$ 90°; night > 98°). The solar zenith angles for the 1D model were chosen such that, for each day, it is the mean solar zenith angle for the MIPAS data plus/minus the standard error of mean (SEM) with N being the number of data points for each day.

$$\text{SEM} = \frac{\text{Standard Deviation}}{\sqrt{\text{N}}} \tag{2}$$

Since ExoTIC doesn't have diffusion or horizontal and vertical transport, the comparison can only be done for a short period of time. The model results are compared with the MIPAS observations for a total of 9 days from 26th October to 3rd November 2003. Due to different vertical resolutions between the model and MIPAS observations, averaging kernels were applied. The averaging kernels were applied after sampling the model data in the MIPAS altitude grid. Now, ExoTIC being a 1D column model doesn't produce the output at the same geolocations as MIPAS hence the application of the MIPAS averaging kernels was based on the temperature criteria. The following procedure was applied for the convolution:

1. The model profiles were selected, one at a time, from the entire time series.

2. All the profiles from MIPAS within 57.5 and 77.5 degrees N latitude and +/- 6 hours of the model profile's time were selected.

3. For this obtained MIPAS sample of temperature profiles, the root mean square value was calculated with the model's temperature profile which is fixed for the entire time series.

4. The geolocation for which the root mean square value of the temperature difference profile was minimal was selected, and averaging kernels for this geolocation were applied to the trace gas profiles from the model.

Using this procedure, we have obtained model profiles that were adjusted to the vertical resolution of MIPAS. The data was then averaged daily and the absolute or relative differences w.r.t a day before the event, i.e. 26th October 2003 (day 299), was calculated.

## 3.1 Estimation of the polar vortex edge

There are different methods of estimating the vortex edge and one of the methods widely used is using CO as a tracer of vortex air. Due to its strong vertical gradient and longevity in the polar winter vortex, carbon monoxide is commonly used as a tracer of vortex air originating from the upper mesosphere and lower thermosphere. Hence it can be used to estimate the vortex edge. Here, we have used a CO vmr threshold (discriminating mesospheric air from the background) as vortex criterion. We need an altitude-independent criterion (as we need to differentiate entire profiles) and our short time period (a few days) allows for a time-independent definition. A chemical vortex definition (via CO) is also widely used in the mesosphere (Harvey et al., 2015). And for the chemical species and the SPE responses discussed here, the relevant altitude range is more in the stratosphere. In the stratosphere, the vortex might be considerably smaller and is commonly determined from the potential vorticity. Here,

we use the tracer gradient instead to be consistent with the MIPAS observations that we use for comparison. Figure 4 shows volume mixing ratios of CO versus latitude in the Northern Hemisphere for different longitude bins of size 20 (shown by different colours) and averaged over latitude bins of size 5 from 27[th] October (day 300) to 3[rd] November (day 307) 2003 separated by day-time and nighttime. Choosing a threshold of 0.5 ppm of CO, the polar vortex boundary can be determined from the corresponding x-axis where an increase of the volume mixing ratios start to occur. An increase is observed starting at a latitude of approximately around 55°N for the different days. The estimation of the polar vortex boundary helps to choose the MIPAS sampling of the zonal averages for a better inter-comparison. One can also check the vortex boundary by looking at $CH_4$ zonal means, for example, Figure 9 of Funke et al. (2011). From that figure and as also discussed in Funke et al. (2011), the boundary is around 60°N which also works well for the stratosphere. We chose the latitude 57°N as to where the vortex begins and defined the latitude bands 57-77°N as "the edge region of the vortex" and the high latitude bands 70-90°N as "deep in the vortex".

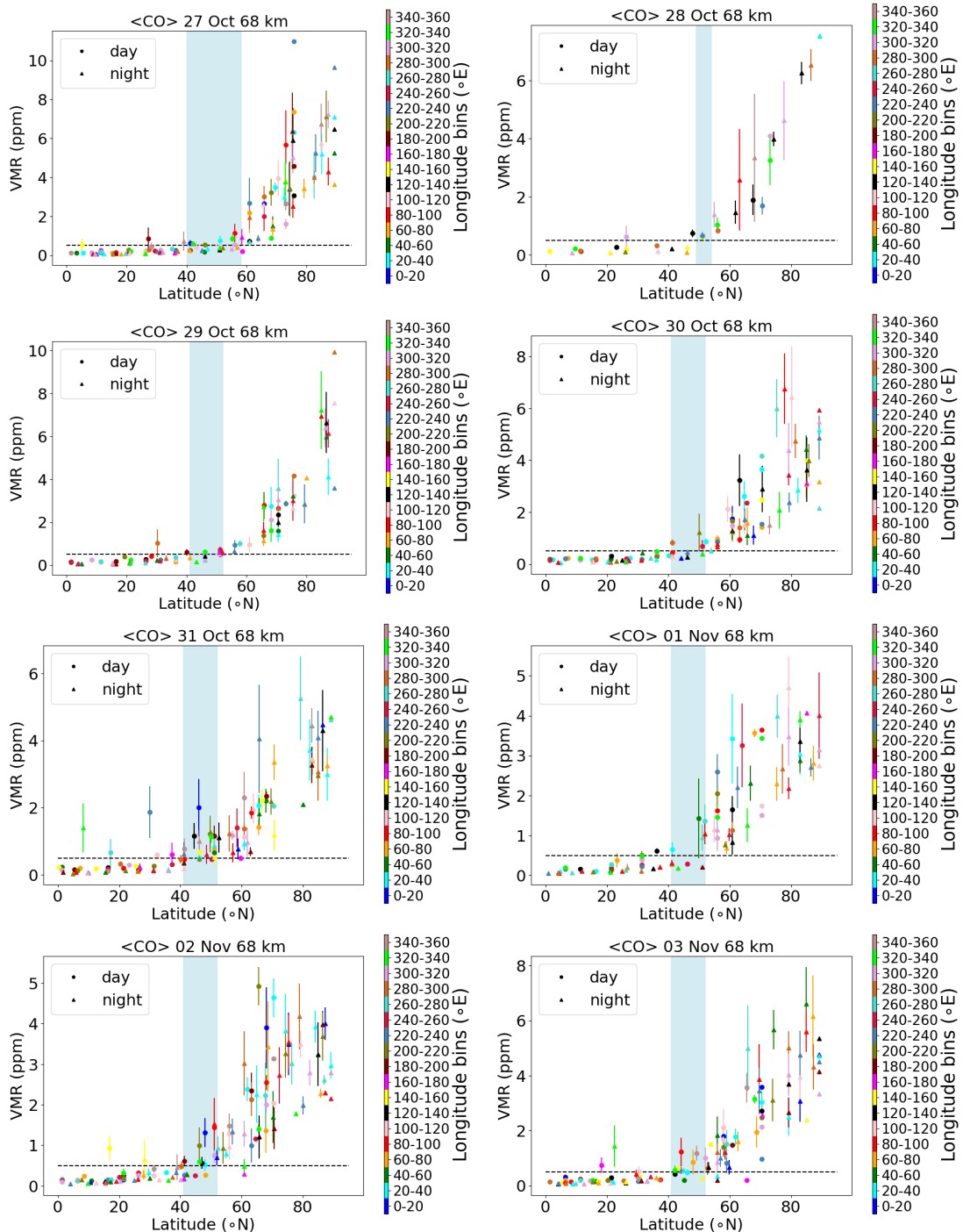

**Figure 4.** MIPAS daily averaged CO (ppm) as a function of latitude (°N) for longitude bins of size 20 and averaged over latitude bins of size 5 for 27$^{th}$ October-3$^{rd}$ November 2003 (day and night) at 68 km altitude. Colors mark the longitude bins from 0-20°E to 340-360°E running from blue to rosybrown. The error bars mark the standard error of mean. The light-blue shaded region marks the latitude range of the vortex edge and the dashed horizontal line is the CO threshold of 0.5 ppm.

## 3.2 HOCl change

A comparison of Envisat MIPAS V5 HOCl measurements (von Clarmann et al., 2006) for the polar Northern Hemisphere (57-77°N) and ExoTIC computations with different settings of HOCl simulation is presented in figures 5 and 6 for daytime and nighttime respectively. The model data is read out in the solar zenith angle range of MIPAS daytime and nighttime observations. The values in both figures 5 and 6 start on the 26[th] October 2003 (day 299). MIPAS observations for HOCl showed enhancements with peak values of 0.2 ppb for daytime and 0.24 ppb for nighttime on the 29[th] of October. The model results with full ion-chemistry also overestimated the observed enhancements for nighttime by a factor of 4 (around 1.25 ppb), and for day-time by a factor of 3 (around 0.65 ppb) produced at an altitude of 35-40 km. HOCl enhancements below 30 km was also observed for both daytime and nighttime with full ion-chemistry and a peak producing 0.12 ppb was observed in the mesosphere during nighttime. Sensitivity studies were performed setting $O(^1D)$ to photo-chemical equilibrium that showed a decrease in the enhancements from 1.25 ppb to 0.55 ppb during nighttime and 0.65 ppb to 0.08 ppb during daytime produced by the model also at 35-40 km during the event. The higher mixing ratios of HOCl below 30 km for both daytime and nighttime also disappeared with this setting. Switching off the chlorine ion-chemistry led to the removal of 0.2 ppb HOCl observed in the mesosphere during nighttime. And similar behaviour was also observed for the parameterised $NO_x$ and $HO_x$ model. However, much lower values were observed during daytime for the model, setting $O(^1D)$ to photo-chemical equilibrium, without chlorine ion-chemistry and parameterised $NO_x$ and $HO_x$ compared to MIPAS observations. Reaction rate constants and photo-chemical data follow in general the JPL-2006 recommendations from Sander et al. (2006).

After the application of averaging kernels the higher mixing ratios produced by the model were smeared out over altitudes and reduced in their peak value. For daytime, the peak value of the full ion-chemistry model decreased from 0.65 ppb to 0.4 ppb. And for the sensitivity studies with $O(^1D)$ in photo-chemical equilibrium, without chlorine ion-chemistry and parameterised $NO_x$ and $HO_x$, the peak value of 0.08 ppb decreased to 0.03 ppb. This peak value occurs due to the increased availability of chlorine atoms due to the catalytic ozone destruction cycle. The HOCl concentration reaches a peak around 35 km during daytime because this altitude represents the optimal conditions for the ClO-HOCl catalytic cycle (Reactions R20, R21, R22 and R19) to occur. For nighttime, full ion-chemistry peak value of 1.25 ppb went down to 0.84 ppb after applying the MIPAS averaging kernels. Setting $O(^1D)$ to photo-chemical equilibrium decreased the peak value from 0.55 ppb to 0.36 ppb which is in better agreement with the MIPAS observations. For the model without chlorine ion-chemistry and parameterised $NO_x$ and $HO_x$, the enhancements of 0.58 ppb went down to 0.33 ppb that also agrees quite well. Jackman et al. (2008) compared results from the Whole Atmosphere Community Climate Model (WACCM3) with MIPAS observations and applied MIPAS averaging kernels for the Halloween SPE 2003 and found the HOCl peak at an altitude of 48 km on the 29[th] of October and the MIPAS averaging kernels moved it down to 40 km. ExoTIC however produced the peak around 35-40 km itself for both daytime and nighttime for all the test cases. This is actually quite in agreement with the MIPAS observations and the application of the averaging kernels also didn't shift the peak in terms of altitude. This difference in the peak altitude between the results from Jackman et al. (2008) and ExoTIC might be due to the fact that WACCM3 has fully interactive dynamics, radiation, chemistry and other parameterizations whereas ExoTIC includes only the chemistry. Damiani et al. (2012) considered the

SPE of January 2005 where they observed 0.2 ppb increase of HOCl during the event in the polar cap region also using the WACCM model that agreed quite well with Microwave Limb Sounder (MLS) observations. Enhancements of HOCl results from enhanced $HO_x$ constituents. In the middle stratosphere, it is mainly accelerated by odd hydrogen chemistry (via Reaction
365   R19). The morphology of the HOCl distribution and it's temporal variation is a combined effect of photolysis, temperature and availability of ClO, $HO_2$ and OH, which in themselves show pronounced diurnal variation.

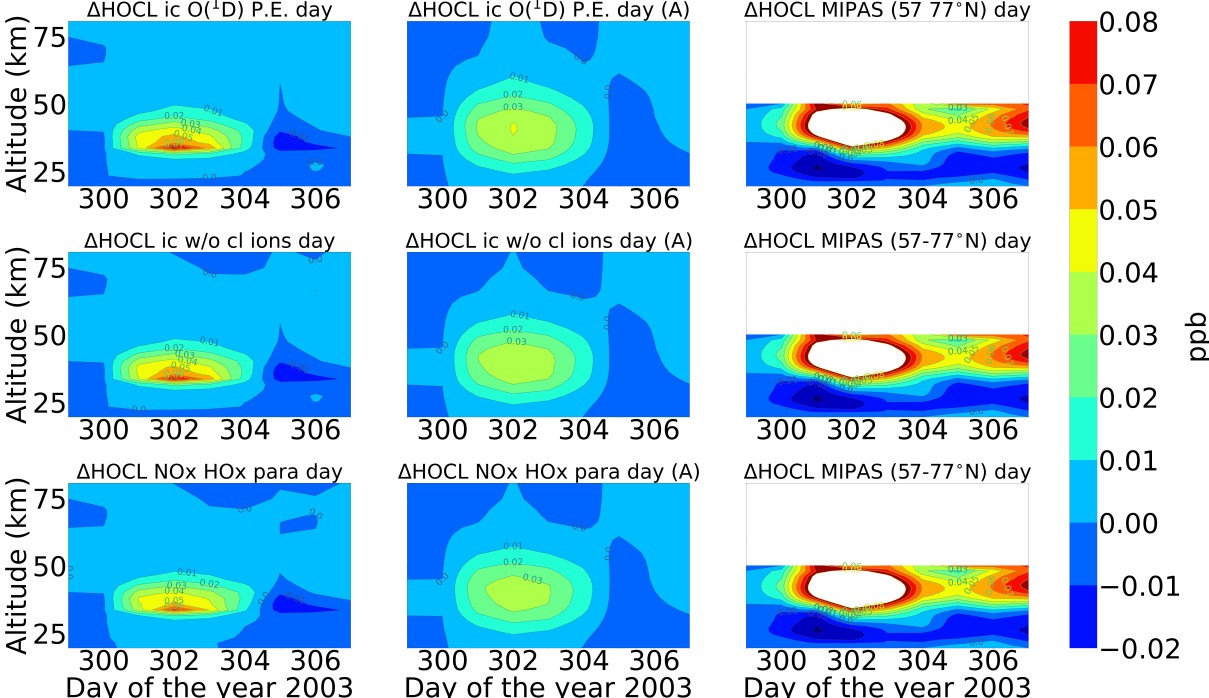

**Figure 5.** Absolute differences of daily averaged data for HOCl w.r.t. a day before the event, i.e. $26^{th}$ October 2003. Starting point is $26^{th}$ October 2003 and for the three different model settings (Sensitivity tests (row-wise): ion-chemistry with $O(^1D)$ in photo-chemical equilibrium, switching off chlorine ion-chemistry and parameterised $NO_x$ and $HO_x$); column-wise: without Averaging kernal (A), with Averaging kernal (A) applied and MIPAS observations averaged over 57-77°N for day-time (sza $<=$ 90°). For daytime, the white region below 50 km is the MIPAS peak (0.2 ppb) and the colorbar is adjusted to the lower mixing ratios predicted by the model (first plot). The white region above 50 km for the MIPAS observations represent meaningless data, where the values of Averaging kernal (A) diagonal elements are close to zero ($<$ 0.03) that indicate no sensitivity to the retrieved parameter at the corresponding altitude. Colorbar interval: (-0.02, -0.01, 0.00, 0.01, 0.02, 0.03, 0.04, 0.05, 0.06, 0.07, 0.08)

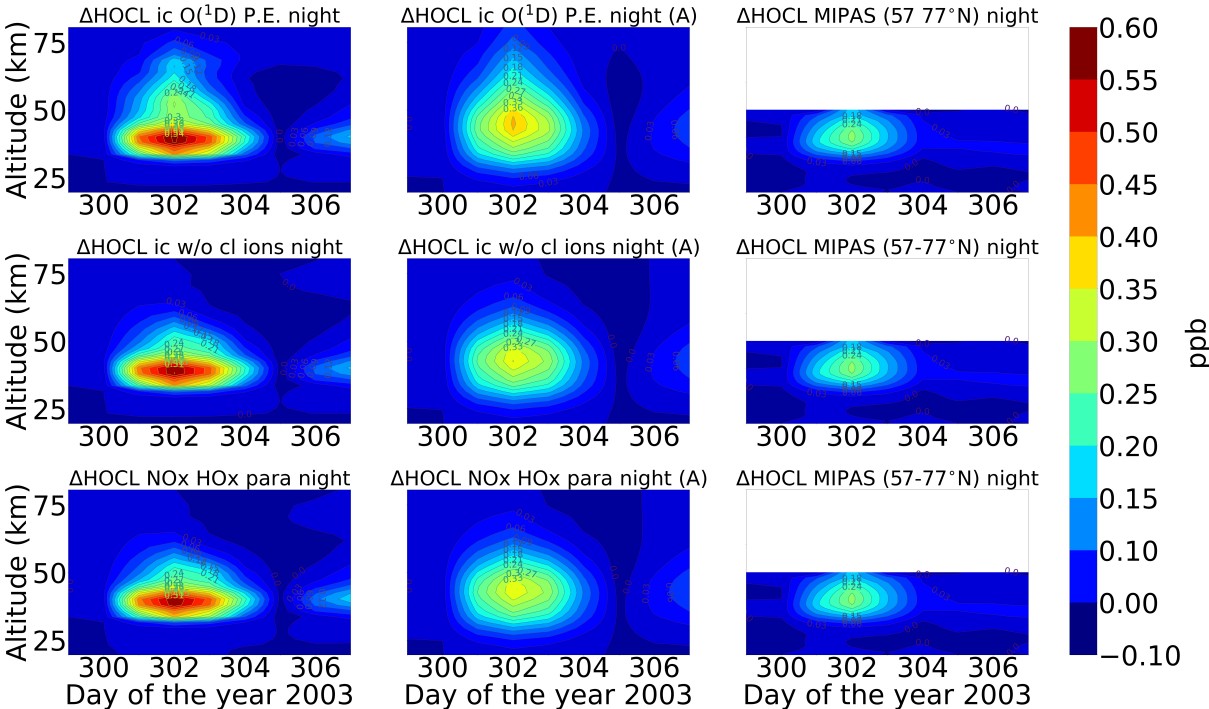

**Figure 6.** Same as figure 5 but for nighttime (sza > 98°). Colorbar interval: (-0.10, 0.00, 0.10, 0.15, 0.20, 0.25, 0.30, 0.35, 0.40, 0.45, 0.50, 0.55, 0.60)

### 3.3 ClONO$_2$ change

Figures 7 and 8 show the daily averaged absolute differences for ENVISAT MIPAS V5 (Höpfner et al., 2007a) and modelled chlorine nitrate w.r.t. 26[th] October 2003 for day-time and nighttime. The zonal average for ClONO$_2$ observations was also taken over a latitude range of 57-77°N, the edge region of the polar vortex. Continuous enhancements of ClONO$_2$ is observed for the full ion-chemistry model with peak values of 0.96 ppb and 1.15 ppb also approximately two days after the event for daytime and nighttime respectively starting from the onset of the event on 28[th] of October. The peak was observed around 25 km. In case of the model with ion-chemistry but O($^1$D) set to photo-chemical equilibrium, a peak value of around 0.18 ppb was observed for both day-time and nighttime which compared much better with the MIPAS observations. Similar results were also observed for the sensitivity study without chlorine ion-chemistry and parameterised NO$_x$ and HO$_x$ model. The increase was seen also approximately two days after the event in the altitude range of 35-40 km. After the application of averaging kernels the peak value for the full ion-chemistry daytime and nighttime decreased down to 0.66 ppb and 0.78 ppb respectively. For the rest, the peak value decreased from 0.12 ppb to 0.09 ppb for daytime and from 0.18 ppb to 0.12 ppb for nighttime. Jackman et al. (2008) observed ClONO$_2$ maximum enhancements of 0.3-0.4 ppb with the peak production at a higher altitude of 40-45 km with WACCM3. The peak however was produced several days later than MIPAS. The application of MIPAS averaging

kernels moved the peak down to 40 km and the predicted peak increases are reduced substantially to about 0.2 ppbv, about a factor of 2 less than MIPAS observations.

The enhanced $ClONO_2$ production happens due to SPE produced $NO_x$ via Reaction R32. $ClONO_2$ is removed mainly by photolysis in the sunlit atmosphere and, to a lesser extent, by reaction with atomic oxygen. And due to it's pressure dependence, $ClONO_2$ formation by Reaction R32 is more effective at lower altitudes (Funke et al., 2011). The zonal average of MIPAS observations were tested for latitude bands 57-77°N, i.e., at the edge region of the polar vortex and 70-90°N, deep in the polar vortex. The sample of 57-77°N works better for the inter-comparison of $ClONO_2$ compared to 70-90°N. In case the sample is taken deep in the vortex, the model seemed to fairly underestimate the peak values. This can be explained by the Reaction R32, where formation of ClO needs sunlight, which is again available more at the edge region of the polar vortex. But $ClONO_2$ can also photolyse in the presence of sunlight and due to this, there is a balance between the two processes and $ClONO_2$ can form at the edge region which can be transported deep into the vortex and conserved there at high latitudes. This however cannot be reproduced by the 1D model because it is fixed at a certain location and has no transport.

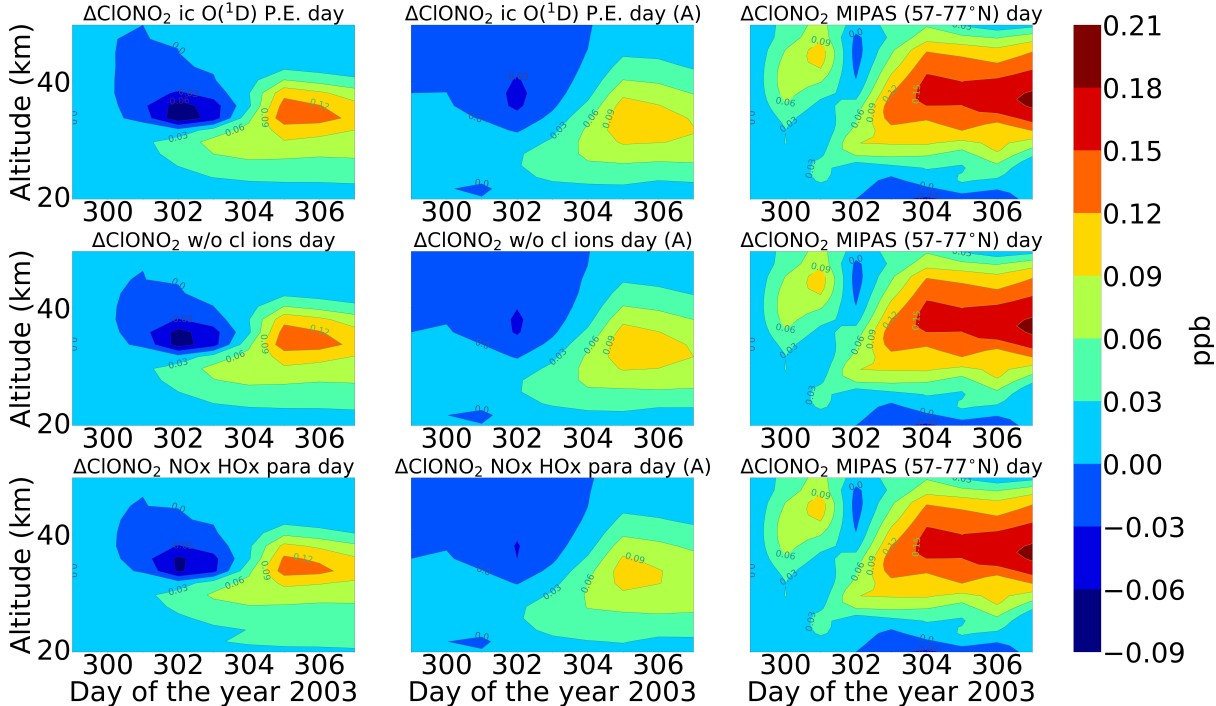

**Figure 7.** Same as figure 5 but for $ClONO_2$. Colorbar interval: (-0.09, -0.06, -0.03, 0.00, 0.03, 0.06, 0.09, 0.12, 0.15, 0.18, 0.21)

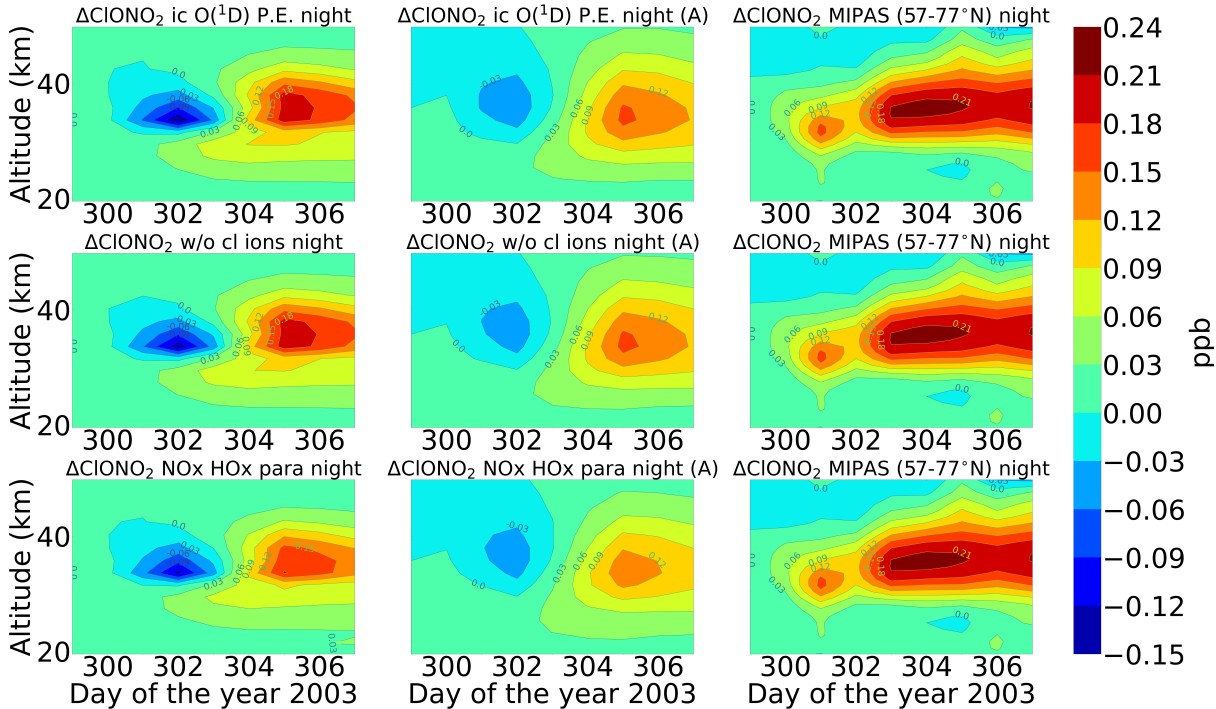

**Figure 8.** Same as figure 6 but for ClONO$_2$. Colorbar interval: (-0.15, -0.12, -0.09, -0.06, -0.03, 0.00, 0.03, 0.06, 0.09, 0.12, 0.15, 0.18, 0.21, 0.24)

### 3.4 Odd oxides of nitrogen (NO$_y$)

An important impact of proton precipitation in the middle atmosphere is the formation of NO$_x$ which happens by the disso-
ciation of molecular nitrogen by ionisation and subsequent recombination with oxygen. In order to assess the agreement of
the ENVISAT MIPAS V5 (Funke et al., 2005) and modelled SPE related odd nitrogen enhancements, total NO$_y$ (=NO + NO$_2$
+ HNO$_3$ + 2N$_2$O$_5$ + ClONO$_2$) is compared. The observed and modelled NO$_y$ enhancements w.r.t. 26$^{th}$ October is shown in
figures 9 and 10 for daytime and nighttime conditions respectively. Averaging kernels are also applied to the model profiles
for the different NO$_y$ species for the different model settings and then added, except for NO. In case of MIPAS NO and NO$_2$
data, there is a complication which is, that instead of mixing ratios the logarithms of the mixing ratios are retrieved; also the
averaging kernels refer to the logarithms of the mixing ratios. The application of MIPAS averaging kernels to a better resolved
profile on the basis of the coarse-grid averaging kernel **A** of the logarithm of the mixing ratio then is (Stiller et al., 2012):

$$x_{\mathrm{new}} = \exp(\mathbf{A}\log(x_{\mathrm{orig}}) + (\mathbf{I} - \mathbf{A})\log(x_{\mathrm{a}})) \tag{3}$$

There is a general issue with logarithmic retrievals, because regularization is self-adaptive and depends on the actual state
of the atmosphere. For an SPE response, if the NO peaks around 50-60 km and if there is a better sensitivity at this altitude the
Jacobian and the averaging kernels scale with the volume mixing ratio. For NO$_2$ however, the logarithmic averaging kernels

behave well and are not dependent on the actual conditions (for a deeper discussion of the problem of time and state dependent averaging kernels, see von Clarmann et al. (2020)). Due to this complication, for the total $NO_y$ in the second column of figures 9 and 10 for both daytime and nighttime, NO is added without the application of the averaging kernels as compared to the rest of the species.

The magnitude of $NO_y$ enhancements is found to be larger for the ExoTIC model with ion-chemistry settings compared to the MIPAS observations for both daytime and nighttime. However the SPE induced $NO_y$ layer is reproduced well in terms of vertical distribution. The MIPAS observations showed a production of 30-40 ppb in the upper stratosphere during nighttime and 20-30 ppb during day-time. The MIPAS observations show much higher dynamics than ExoTIC for both day-time and nighttime. The nighttime values show stronger dynamics with decreased ionisation rates on day 305-306 and connected to this less $\Delta NO_y$ reaches a constant state after the maximum of the first SPE on day 303. This could be an indication that some of the $NO_y$ recombination speeds are faster than expected. The results are shown upto 50 km since above 50 km NO is the largest contributor (Funke et al., 2011) and averaging kernels are not applied to NO. Another reason is that large uncertainties of small vmr values of $ClONO_2$ (Höpfner et al., 2007b) will spoil the $NO_y$ sum and it's uncertainty. The nighttime results compare better with the observations, specially for the parameterised $NO_x$ and $HO_x$ model. For day-time, the model with and without averaging kernel (**A**) didn't make too much of a difference because NO was added without averaging kernel (**A**) and was abundant during daytime.

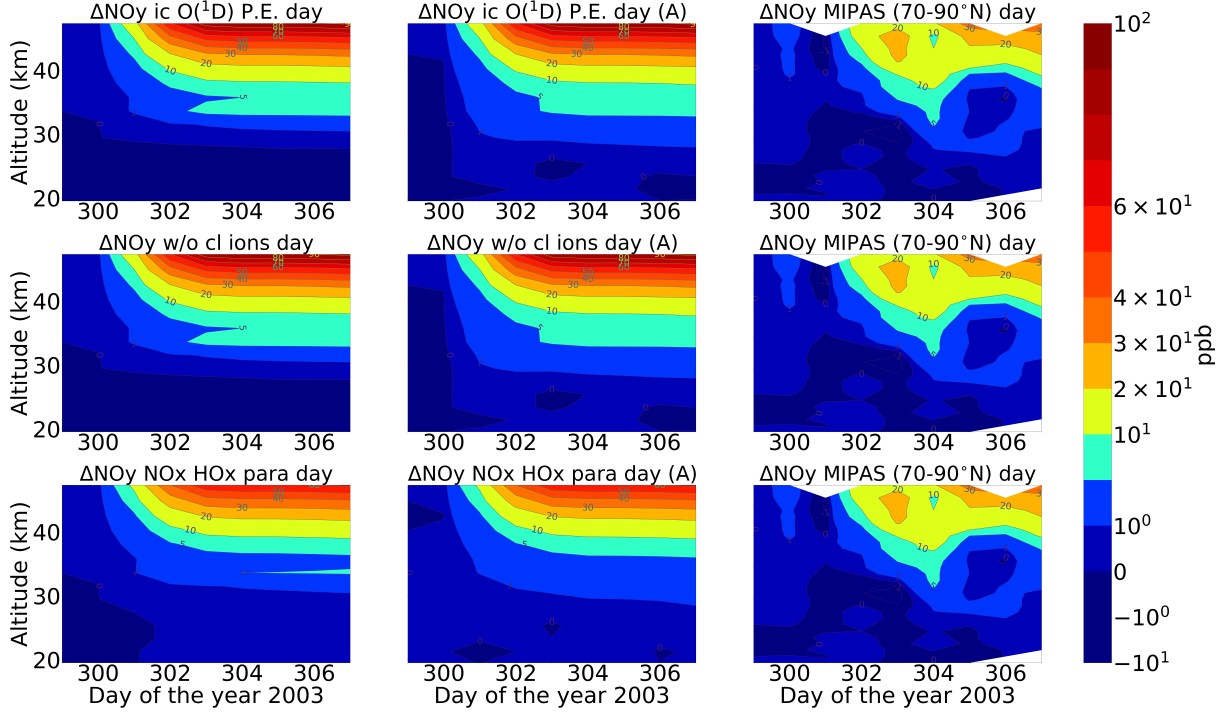

**Figure 9.** Same as figure 5 but for $NO_y$ species. Colorbar interval: ($-10^{-1}$, $-10^0$, $0$, $10^0$, $10^1$, $2*10^1$, $3*10^1$, $4*10^1$, $6*10^1$, $10^2$)

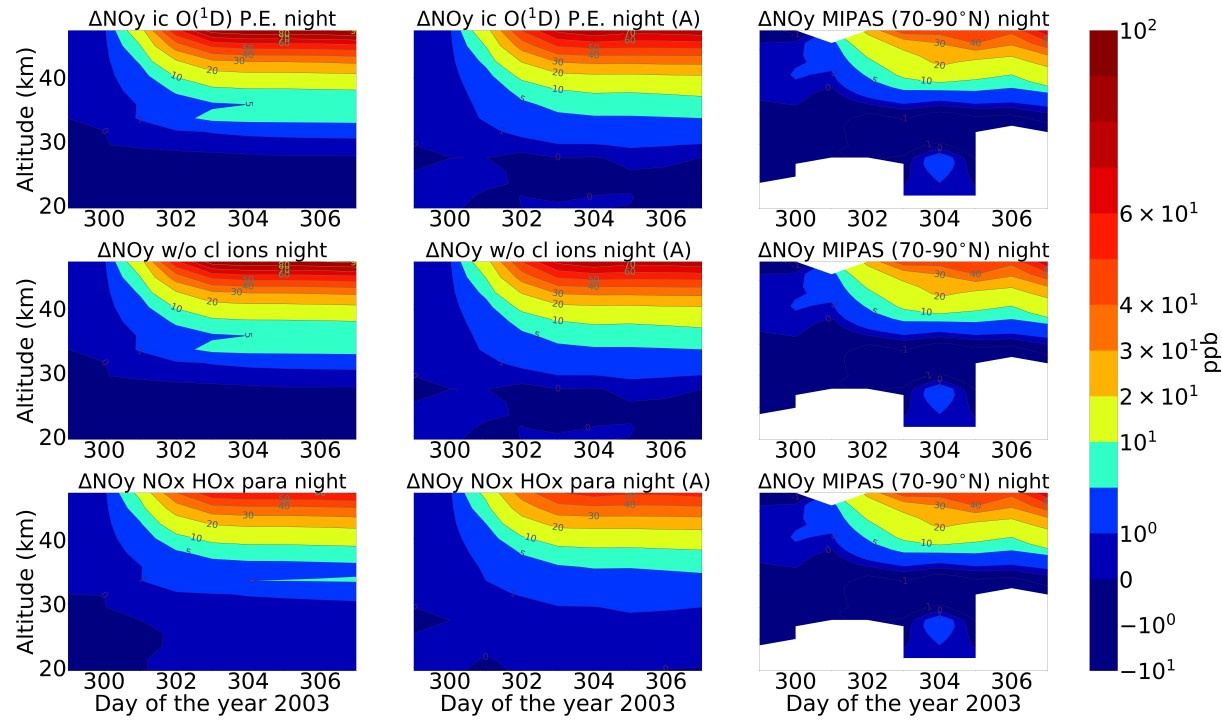

**Figure 10.** Same as figure 6 but for $NO_y$ species. Colorbar interval: $(-10^{-1}, -10^0, 0, 10^0, 10^1, 2*10^1, 3*10^1, 4*10^1, 6*10^1, 10^2)$

### 3.5 Ozone

Energetic particles in the polar atmosphere enhances the production of $NO_x$ and $HO_x$ in the winter stratosphere and meso-
sphere. Both $NO_x$ and $HO_x$ are powerful ozone destroyers. An important aspect in the evaluation of the ability of models is the
reproduction of the observed ozone destruction caused by the catalytic cycles of $NO_x$ and $HO_x$ during SPE induced chemical
composition changes. As inferred from observations, stratospheric ozone decreases due to the indirect effect of EPP by about
10–15 % as observed by satellite instruments (Meraner and Schmidt, 2018). López-Puertas et al. (2005) found $HO_x$ related
mesospheric ozone losses upto 70% and $NO_x$ related stratospheric loss of around 30% during the Halloween SPE.
Figures 11 and 12 show ENVISAT MIPAS V5 (Glatthor et al., 2006) and modelled temporal evolution of the relative ozone
changes w.r.t. 26[th] October, averaged over 70-90°N for daytime and nighttime respectively. For ozone, the long term history of
air parcels is more important as air parcels that are ozone depleted gets dispersed into the mid-latitudes if they are at the edge
region of the vortex. So a sample deeper in the vortex for ozone is better, the reason we chose 70-90°N here. A loss of 60-75 %
is observed during the event itself in the mesosphere that is short lived and is related to the $HO_x$ catalytic cycle (Reactions R6,
R7, R9 and R10, (Funke et al., 2011)(Bates and Nicolet, 1950)). The ozone recovers after the event, since $HO_x$ is short-lived.
A second peak is observed on the 3rd of November which is related to a weaker coronal mass ejection event. $NO_x$ related
loss of 15 % is observed in the stratosphere that lasts longer and is also related to the polar winter atmosphere (Reactions R12
and R13). The full ion-chemistry shows an ongoing loss of 45 % starting from the event day and the sensitivity study with

O($^1$D) in photo-chemical equilibrium confirmed that this loss is due to Reactions like R40 and R37, which produces OH and
Cl contributing further to ozone loss. The agreement between the observations and the model results, for nighttime, for the
three model results except for the full ion-chemistry is excellent in the mesosphere indicating a good ability of the model to
reproduce HO$_x$ related ozone loss for SPEs. However, the ozone loss shifted to lower altitudes for both daytime and nighttime
in all the model settings as compared to MIPAS observations. This could be explained as a result of the AISstorm ionisation
rates that was used in the model which could be different to what MIPAS might have experienced during the SPE. AISstorm
here uses proton fluxes from GOES 10 and the ionisation rates should be a lower estimate. The SPE ionisation happens in
the denser atmosphere, therefore the conversion from particle fluxes into ionisation should be more or less precise in terms
of total ionisation and altitude. The main uncertainty of SPE ionisation in AISstorm is the size of the area that is affected by
high energetic particle precipitation. This cannot be derived from these channels but is taken from a lower energy channel on
another satellite and thus it might be an underestimate of the polar cap size. However if that would be important for the spatial
average 70-90°N, we should see an underestimation of the ionisation (and NOy) as well, which doesn't seem to be the case
(and which is unlikely anyway as the equatorward boundary should be at about 60°N).

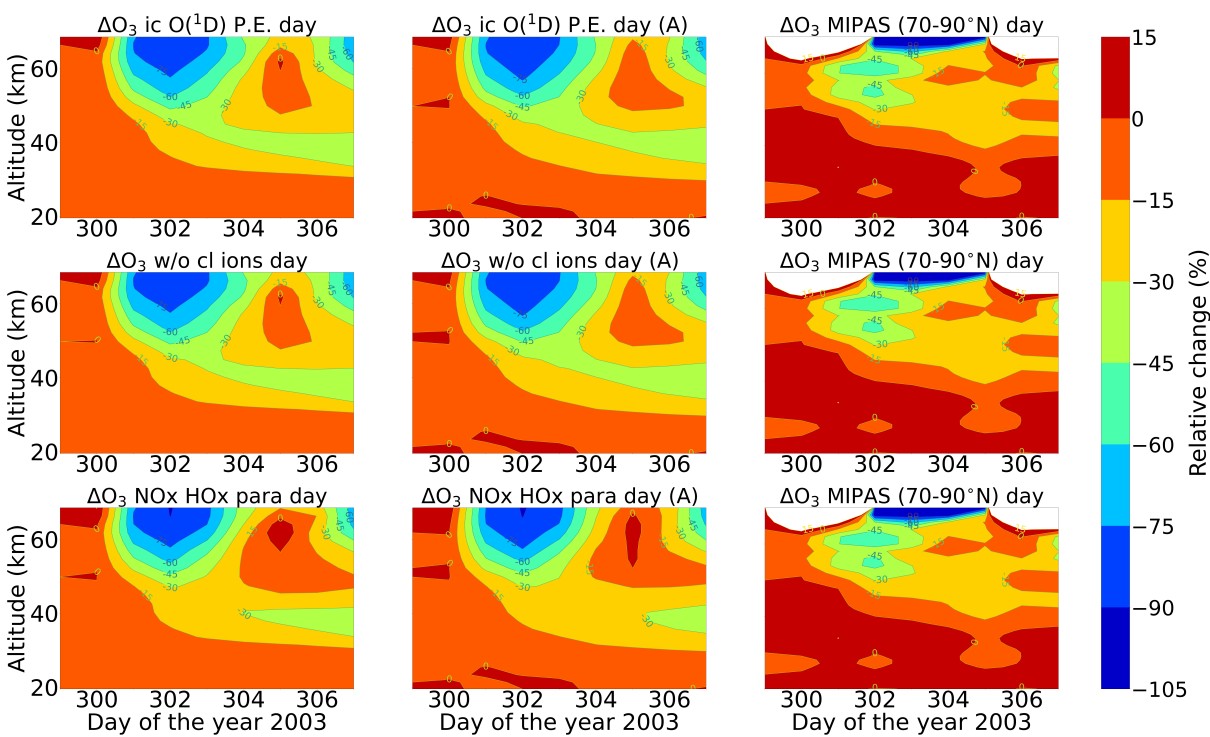

**Figure 11.** Relative difference of ozone w.r.t. 26$^{th}$ October. Rest is same as Figure 5. Colorbar interval: (-105, -90, -75, -60, -45, -30, -15, 0, 15)

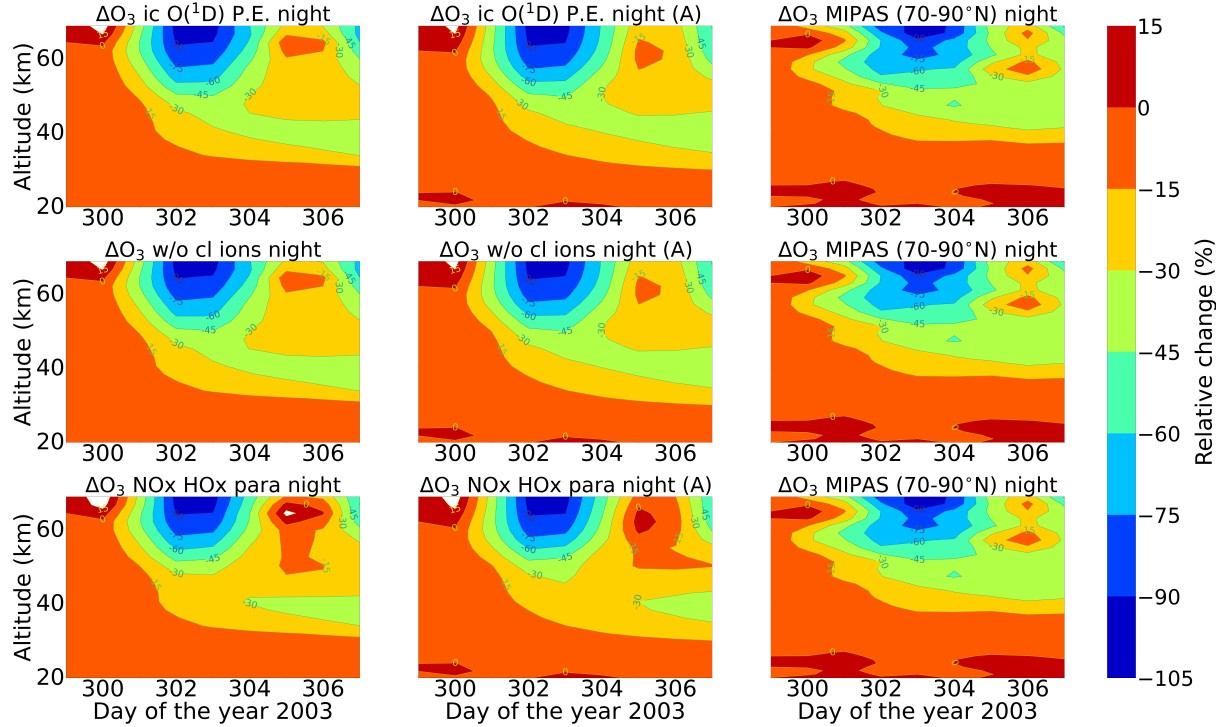

**Figure 12.** Same as figure 11 but for nighttime. Colorbar interval: (-105, -90, -75, -60, -45, -30, -15, 0, 15)

## 4 Comparison of the Halloween storm and the extreme solar event of 775 A.D.

In this section, a comparison study between the Halloween storm of 2003 and an extreme event of 775 A.D. is presented. The model simulations are performed at a latitude of 67.5°N and begin at the noon of 27$^{\text{th}}$ October (day 300). The ionisation rate profiles are obtained from AISstorm for both the events and are input from the noon of October 28 to noon of October 29. Since the studies were performed without horizontal and vertical transport, the results shown here are restricted to a short time period. The results are shown for the model simulations for the settings that compared well with MIPAS observations.

### 4.1 $\tilde{\text{NO}}_{\text{y}}$ and $\text{HO}_{\text{x}}$:

Figure 13 shows the formation of $\tilde{\text{NO}}_{\text{y}}$ during the Halloween SPE and the extreme scenario with different settings of the ion-chemistry. $\tilde{\text{NO}}_{\text{y}}$ consists species of odd nitrogen as shown in equation 4:

$$\tilde{\text{NO}}_{\text{y}} = \text{NO} + \text{NO}_2 + \text{N} + \text{HNO}_3 + 2\text{N}_2\text{O}_5 + \text{NO}_3 + \text{ClONO}_2, \tag{4}$$

After the onset of the event, $\tilde{\text{NO}}_{\text{y}}$ starts accumulating over time in the stratosphere and lower mesosphere. For the extreme scenario, the volume mixing ratio (VMR) of $\tilde{\text{NO}}_{\text{y}}$ is about one order of magnitude larger compared to the Halloween SPE in the upper stratosphere and lower mesosphere. For example, the amount of $\tilde{\text{NO}}_{\text{y}}$ at 60 km is found to be 50 ppb for the

Halloween event compared to 500 ppb for the extreme event. Additionally for the extreme event, $\tilde{NO}_y$ is seen to be formed even in the lower stratosphere (below 30 km). This is because of the high values of ionisation rates that reached further down in altitude in this case. A small difference was observed between the sensitivity study without the chlorine ion-chemistry and the model setting of ion-chemistry with $O(^1D)$ in photo-chemical equilibrium around 75 km. The full ion-chemistry and the ion-chemistry setting $O(^1D)$ to photo-chemical equilibrium didn't show much of a difference. For the parameterised model,

$\tilde{NO}_y$ enhancements are observed in the mesosphere and lower thermosphere with higher values seen for the extreme scenario compared to the Halloween event. Since only N, NO and $NO_2$ are present for the parameterised $NO_x$, the impact of the scavenging Reaction, R2 is stronger and the partitioning between N and NO is different compared to the ion chemistry which also contains other $\tilde{NO}_y$ species like $HNO_3$ etc. R2 drives the $NO_x$ parameterisation and makes the main difference w.r.t. the ion-chemistry. In contrast to our results, which show the total $NO_y$ that includes $NO_x$=N+NO, Andersson et al. (2016)

showed that WACCM-D with the D-region ion-chemistry predicts more $NO_x$ in the mesosphere compared to WACCM with the standard parameterisation of $NO_x$ and $HO_x$ production for the northern polar cap region, latitude > 60°N during the SPE of January 2005. Kalakoski et al. (2020) also reported the same when considering SPEs using proton flux data from satellite-based GOES observations. They considered an event in which the peak proton flux exceeded 100 particle flux units (pfu), with pfu defined as the 5 min average flux in units of particles $cm^{-2}s^{-1}sr^{-1}$ for protons with energy larger than 10 MeV. The rate

of formation of $NO_x$=N+NO is indeed higher in the mesosphere when full ion-chemistry is considered, but the partitioning between the formation of N and $N(^2D)$ forming NO is also increasingly in favour of N, meaning that the rate of loss of NO is also faster in this altitude range when full ion-chemistry is considered. This can lead to less NO depending on the absolute rate of NOx formation, and the partitioning between N and NO in this formation. The $HO_x$ parameterisation doesn't make much of a difference for $\tilde{NO}_y$.

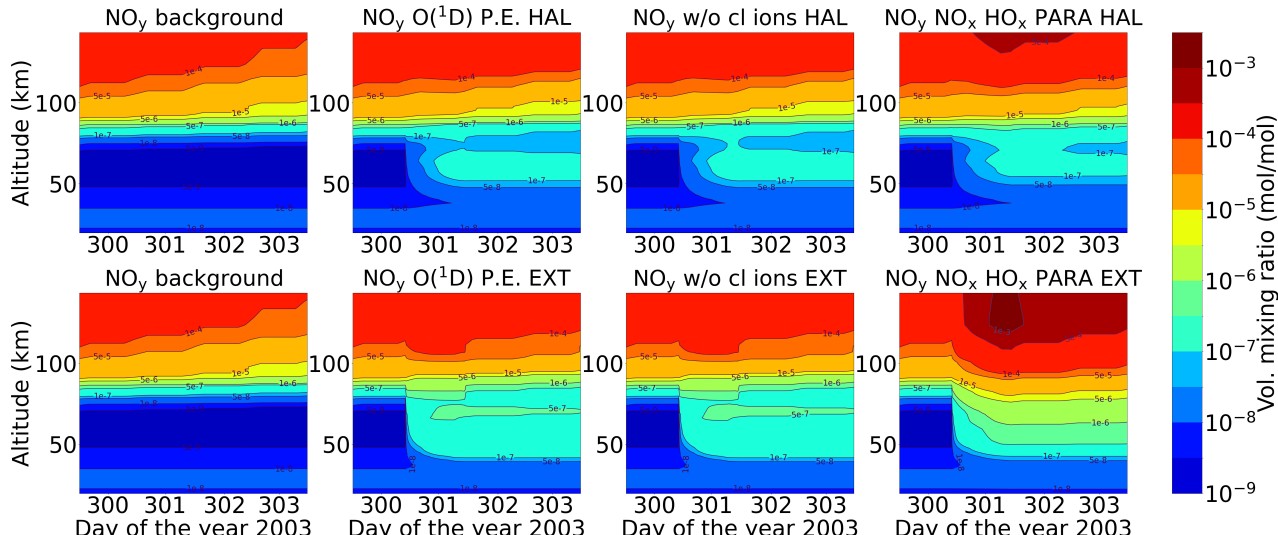

**Figure 13.** Comparison of the Halloween SPE and the extreme scenario (row-wise) for $\tilde{NO}_y$: reference run (background atmosphere), full ion-chemistry with $O(^1D)$ set to photo-chemical equilibrium, without chlorine ions and parameterised $NO_x$ and $HO_x$ model (column-wise) for a high latitude of $67.5°N$.

Figure 14 shows the temporal evolution of $HO_x$ for the Halloween SPE and extreme scenario that consists of odd hydrogen species (equation 5);

$$HO_x = H + OH + HO_2 + 2H_2O_2 \tag{5}$$

$HO_x$ was not shown for the MIPAS comparison due to the fact that H and OH are not provided by MIPAS. $HO_x$ enhancements are seen during the event. For the Halloween SPE, $HO_x$ enhancements of 0.1 ppm were observed in the mesosphere during the event whereas for the extreme scenario these enhancements were seen to penetrate deep down. However after the event stops, the $HO_x$ disappears at higher altitudes, since it is short-lived up there. But at lower altitudes, around 25-40 km, $HO_x$ enhancements of around 1 ppb was found to be continuous and more persistent, specially for the extreme scenario. The full ion-chemistry shows $HO_x$ enhancements below 30 km which is due to the presence of $O(^1D)$, that can react with water vapour, hydrogen and methane via Reactions R37, R38 and R39. However with $O(^1D)$ set to photo-chemical equilibrium, the $HO_x$ enhancements below 30 km disappeared for both the events. The impact of chlorine ion-chemistry is seen to be rather small. For the parameterised model, the recovery of $HO_x$ after the event was found to be slower compared to the ion-chemistry model. The high values of $HO_x$ are balanced by the continuous destruction of water vapour and when it is completely destroyed, the $HO_x$ formation breaks down. So the full parameterisation with both $NO_x$ and $HO_x$ is different but the water vapour is a limiting factor for both. The ion-chemistry with more $HO_x$ can also lead to a faster reaction between OH and $NO_2$ (R41), which can produce $HNO_3$ contributing a difference to the recovery of $HO_x$.

$$OH + NO_2 \longrightarrow HNO_3 \tag{R41}$$

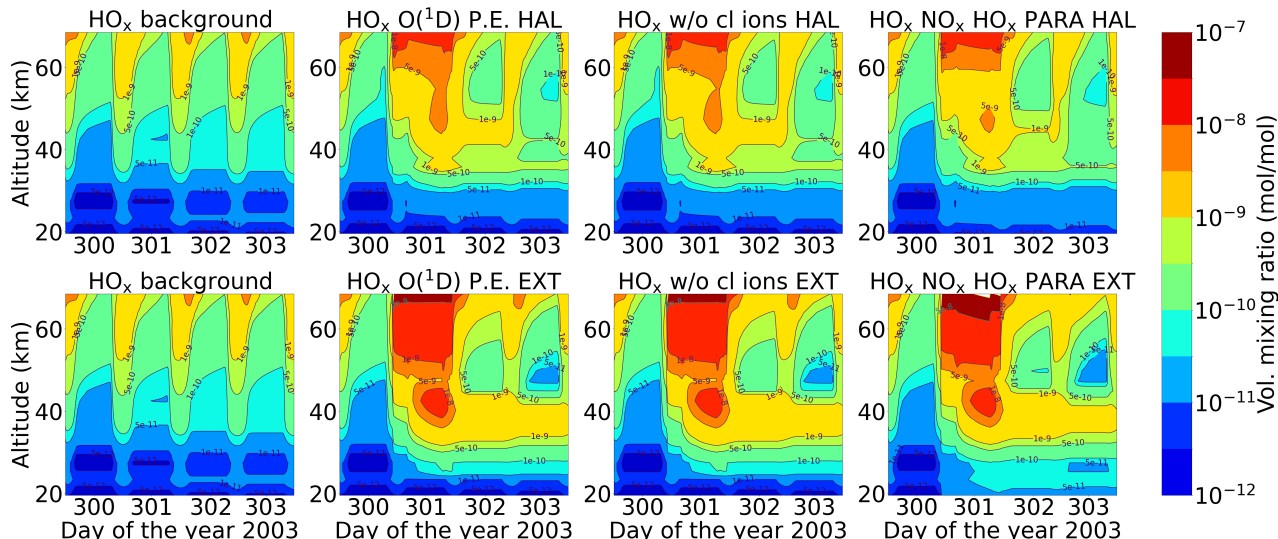

**Figure 14.** Same as Figure 13 but for odd hydrogen species, $HO_x$

## 4.2 Chlorine species:

Figure 15 shows the volume mixing ratios of HCl for the two events. Loss of HCl, which can occur via transformation into reactive chlorine as it is taken up from the gas phase into negative ions, is seen during the event in both cases, pronounced more for the extreme scenario. This occurred both in the stratosphere and the mesosphere for full ion-chemistry. Negative ions upon reaction with HCl can form $Cl^-$ ions, which forms large cluster ions thereby releasing Cl upon recombination (Sinnhuber et al., 2012),

$$HCl + X^- \longrightarrow Cl^- + HX \tag{R42}$$

$$Cl^- + Y \longrightarrow Cl^-Y \tag{R43}$$

$$Cl^-(Y) + Z^+ \longrightarrow Cl + Z + Y \tag{R44}$$

where X = O, $O_2$, $CO_3$, OH, $NO_2$, $NO_3$; Y = HCl, $H_2O$, $CO_2$ and Z = positive ions (Kopp and Fritzenwallner, 1997). $Cl^-$ and $Cl^-$ cluster ions like $Cl^-(H_2O)$ can also release HCl via reaction with H (Kopp and Fritzenwallner, 1997). HCl can react with $O(^1D)$ via Reaction R40 resulting in an enhanced loss of HCl below 40 km but that disappears when $O(^1D)$ is set to photochemical equilibrium. At 40-50 km, ExoTIC with ion-chemistry and $O(^1D)$ set to photo-chemical equilibrium observed 5-15% HCl loss for the Halloween SPE and 20-45% HCl loss for the extreme event. From the sensitivity study without the chlorine ions, it is evident that the huge loss of HCl, around 75% observed in the mesosphere is due to the chlorine ion-chemistry. The parameterised model underestimates the loss of HCl which was also found in studies by Winkler et al. (2009).

The primary neutral reaction that leads to the decrease in HCl below 50 km is a series of reactions involving $HO_x$ species that are part of the catalytic ozone destruction cycle (Reactions R6 and R7). The decrease in ozone concentration has a secondary effect on the concentration of HCl. In the absence of an SPE, ozone plays a role in the conversion of ClO back to HCl. However,

during an SPE, the enhanced ionisation and subsequent formation of ions disrupt this ozone destruction and formation cycle. This leads to an increase in the concentration of ClO and a subsequent reduction in the concentration of HCl. The excess ClO can further participate in additional ozone depletion cycles, leading to a decline in ozone levels during the event. Andersson et al. (2016) reported daily averaged anomalies of HCl in both hemispheres for the latitudinal band 60–82.5°N at altitudes between 40 and 52 km. They compared WACCM-D consisting of the D-region ion-chemistry and WACCM consisting of the standard $NO_x$ and $HO_x$ parameterisation with MLS observations. They found a rapid HCl decrease of about 2-6% during the January 2005 SPE in both hemispheres. With WACCM-D, a loss of around 4% was found compared to a loss of 3% in standard WACCM in the Northern Hemisphere. They also showed that WACCM-D showed better agreement with MLS observations.

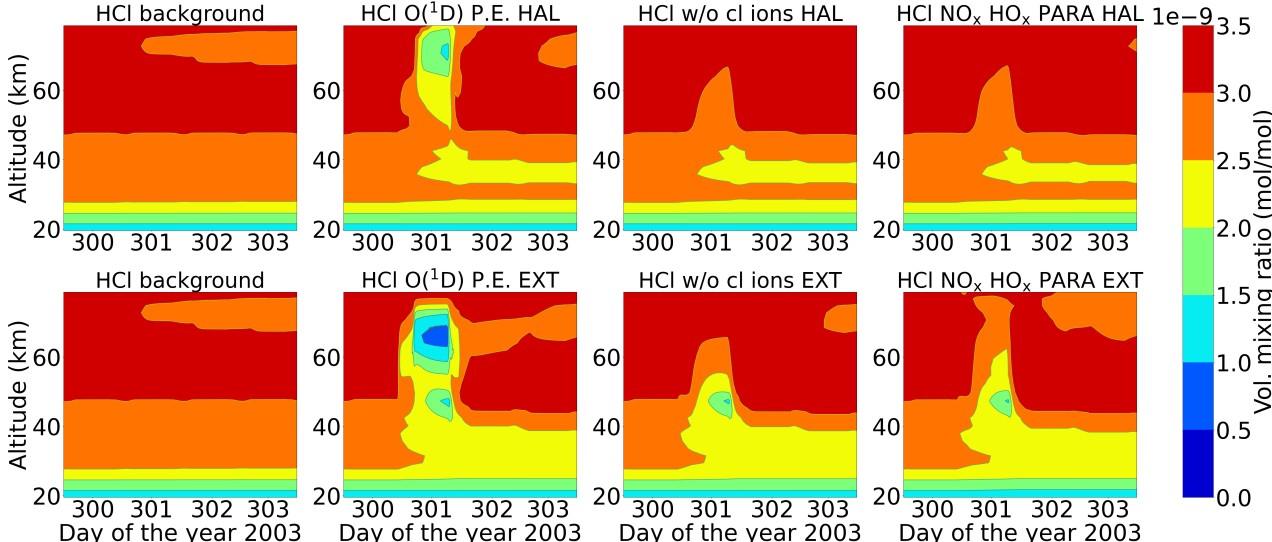

**Figure 15.** Same as Figure 13 but for HCl

Since $\tilde{NO}_y$ and $HO_x$ production is enhanced during SPEs, which is evident from figures 13 and 14, they can react with reactive chlorine species like ClO. ClO can either react with $HO_x$ producing short term enhancement in HOCl (R19), or with $NO_x$ slowly producing $ClONO_2$ (R32). ClO is formed from the reactive Cl via Reaction R21 by neutral gas phase reactions of Cl with ozone in the altitude range of 35-40 km.

From figure 16, it can be seen that ClO decreases during the event for the same altitude range but recovers straight after the event stops. ClO can react with $HO_x$ species during SPEs, particularly $HO_2$ to produce HOCl (R19), which is also a short-lived species. Therefore, a short term enhancement of HOCl is seen during both the events (figure 17) which was also observed by von Clarmann et al. (2005) for the Halloween SPE. However, for the extreme scenario, loss of ClO continues specially around 30-40 km even after the event stops. This is because of the excess $NO_x$ available for the extreme event due to which R32 can happen continuously which is supplemented by an increase in $ClONO_2$ after the extreme event as seen from figure 18. Loss of HOCl seen after the extreme event is due to excess $HO_x$ as well. HOCl can react with OH to form ClO (R24) and then ClO can react back with $HO_2$ and this catalytic cycle thereby results in the loss of both species due to excess $HO_x$.

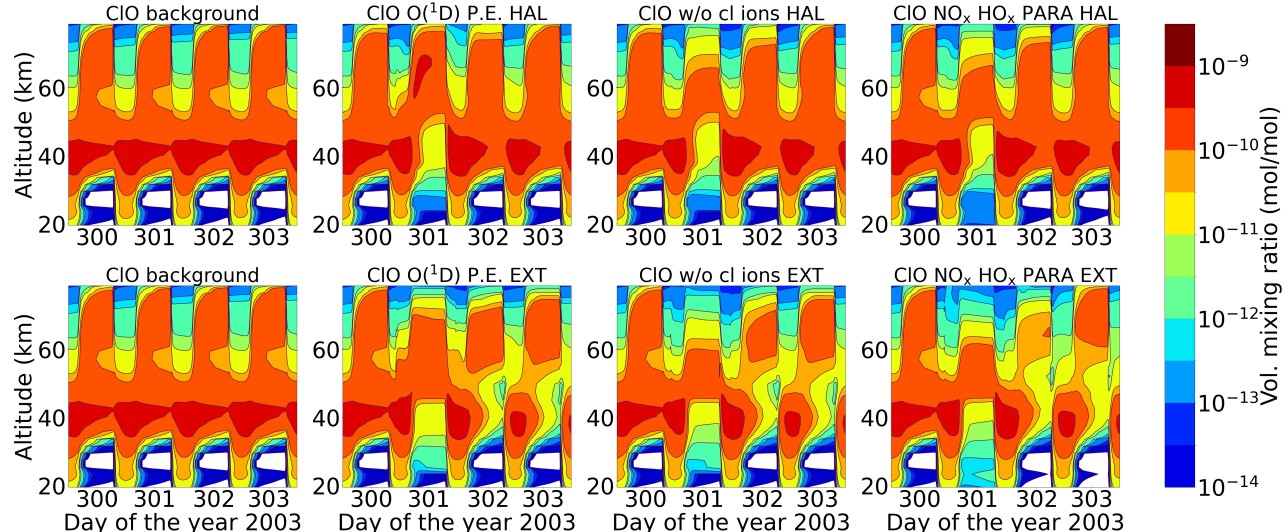

**Figure 16.** Same as figure 15 but for ClO

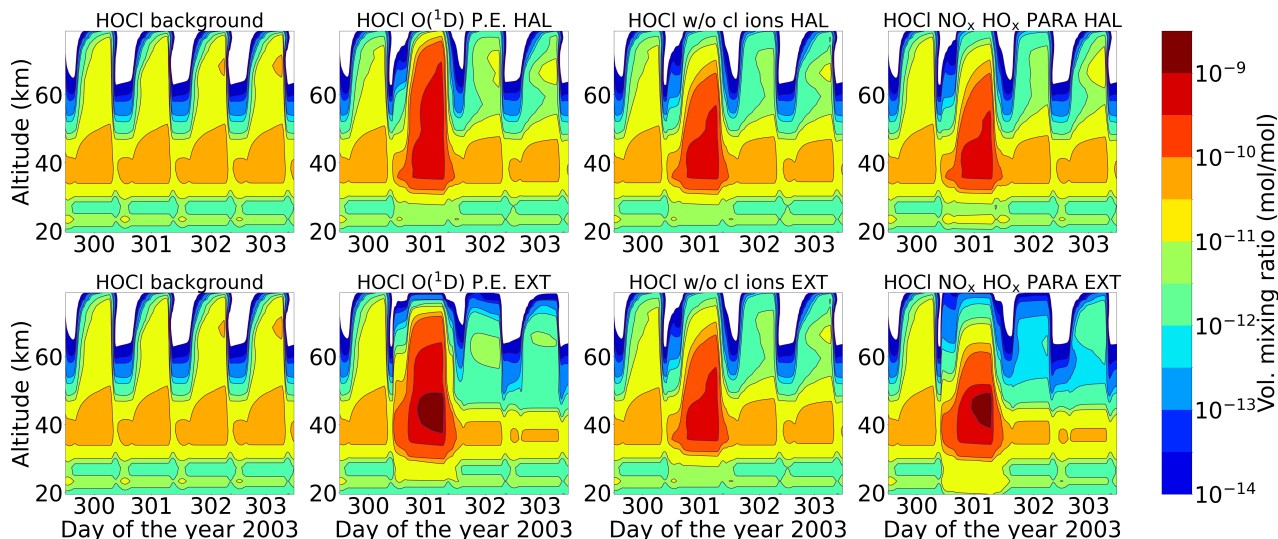

**Figure 17.** Same as figure 15 but for HOCl

An increase in $ClONO_2$ is also observed during both the events at an altitude of 60 km (figure 18). Solomon et al. (1981) pointed out that the increasing $NO_x$ concentrations after a SPE interact with chlorine species, forming chlorine nitrate at the expense of reactive radicals. This reaction is of importance in the lower and middle stratosphere around 40 km, (Jackman et al., 2000) but not so important at higher altitudes. It can be seen from figure 18 that at an altitude of 40 km, $ClONO_2$ increases after the event has stopped. This is because after the event ClO is lost via Reaction R32 since $NO_x$ is formed slowly accumulating over time. So the formation of $ClONO_2$ via R32 is slow, hence leading to high production after the event in the lower and

540

middle stratosphere. The chlorine ion-chemistry plays a small role for ClONO$_2$ around 45-50 km specially during nighttime on day 301. The production of ClONO$_2$ when the chlorine ion-chemistry is switched off is comparatively higher at that altitude.

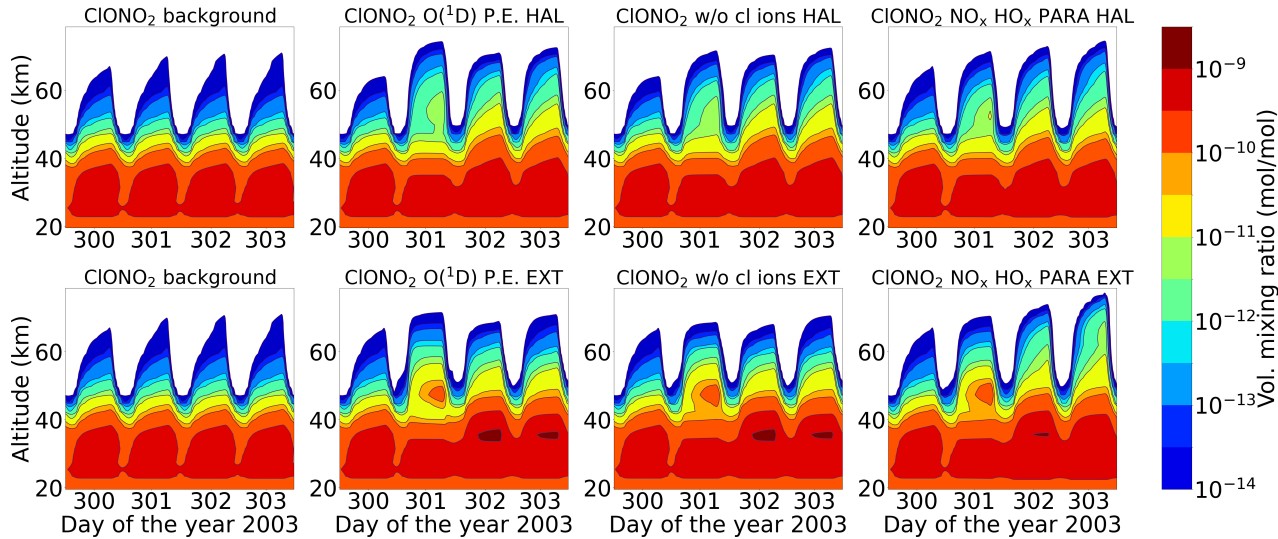

**Figure 18.** Same as figure 15 but for ClONO$_2$

545    Figure 19 shows the chlorine species at an altitude of 40 km for the various model set-ups. There is a small impact of chlorine ion-chemistry on the loss of HCl, whereas the parameterised NO$_x$ and HO$_x$ underestimates it. ClO decreases during the event and transfers to HOCl via Reaction R19. After the event, it recovers and the HOCl enhancements also decrease. For the extreme event however, ongoing loss of ClO during nighttime is observed which is due to the excess HO$_x$ produced during the extreme event. Reformation of HCl after the event is observed that happens via Reactions R26 and R27.

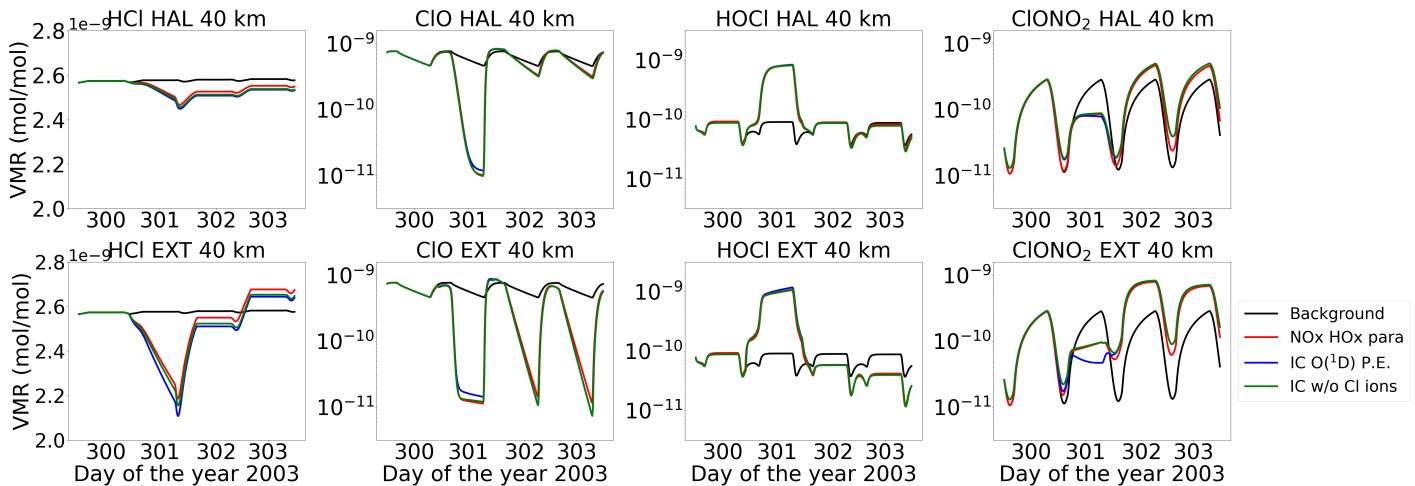

**Figure 19.** Comparison of the Halloween SPE and the extreme scenario for (column wise): HCl, ClO, HOCl and ClONO$_2$ at 40 km.

550 An interesting observation for HCl can be seen from the figure 19 for the extreme scenario, where the recovered HCl after the event shows a positive excursion. This depends on the diurnal cycle of ClO and happens mainly during night time. The reformation of HCl can happen via Reactions R26 and R27 during night time because during daytime $HO_2$ and $H_2O_2$ photolyses. This can be seen from the figure 20 where $HO_2$ and $H_2O_2$ production increases during the event mainly at night time. A steep increase in HCl is observed during the transition from day to night where Cl, $HO_2$ and $H_2O_2$ increases which is 555 constant over the night and increases again afterwards at the beginning of the day. This is due to the fact that during nighttime the concentration of free chlorine atoms is typically low since the primary source of these atoms is the dissociation of chlorine-containing reservoir species, such as chlorine nitrate ($ClONO_2$). $ClONO_2$ occurs predominantly during daytime due to the presence of sunlight, where it is photolysed to release chlorine atoms and hence at sunrise this renewed increase in chlorine atoms results in a subsequent increase in HCl levels.

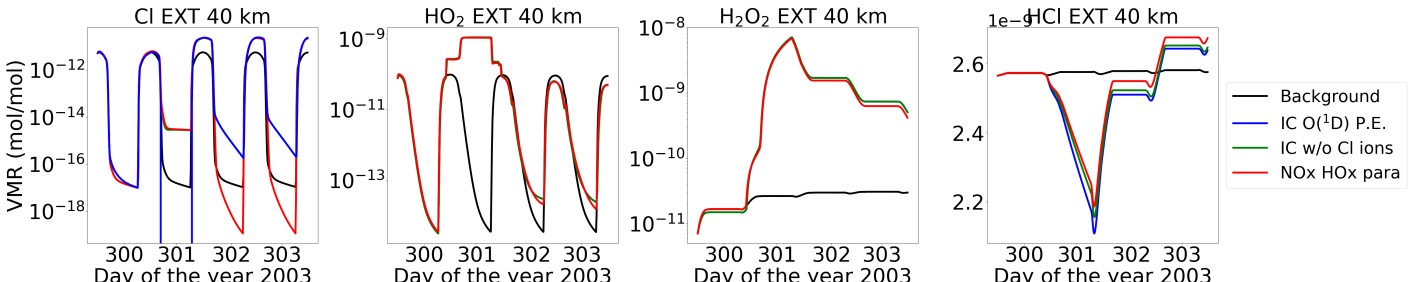

**Figure 20.** Volume mixing ratios of species (Cl, $HO_2$, $H_2O_2$ and HCl) at 40 km for the extreme event. The different lines are for the model settings: reference (black), ion-chemistry with $O(^1D)$ in photo-chemical equilibrium (blue), without chlorine ions (green) and parameterised $NO_x$ and $HO_x$ (red)

 **4.3 Ozone:**

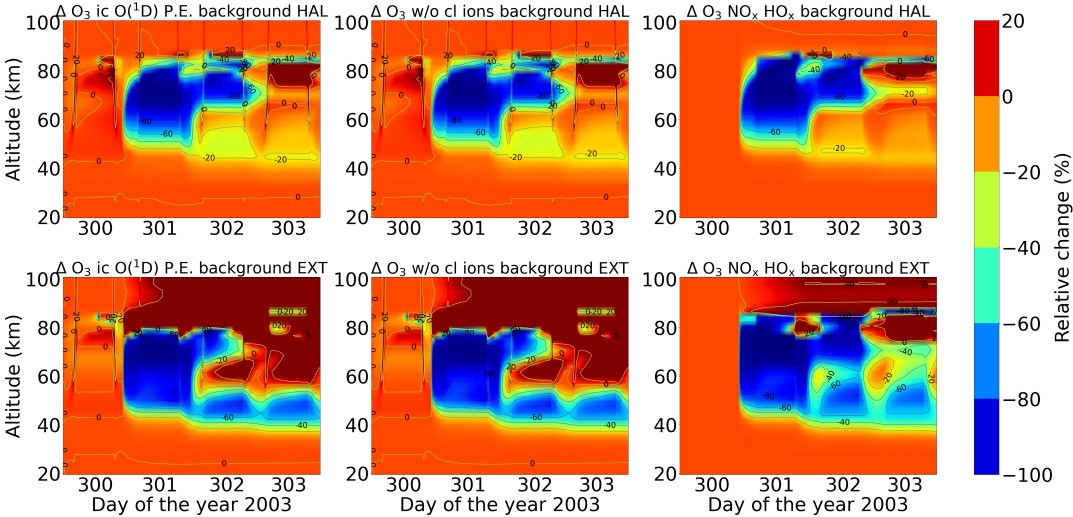

**Figure 21.** Percentage difference of ozone for the different model runs w.r.t. the reference run (row-wise): Halloween event and extreme scenario; (column-wise): ion-chemistry with $O(^1D)$ in photochemical equilibrium, without Cl ions and parameterised $NO_x$ and $HO_x$.

Figure 21 shows the percentage difference of ozone for the two events for the different model runs calculated w.r.t. the reference run. It is seen that with the onset of the event, ozone is completely lost which is due to $HO_x$ enhancements in the mesosphere above 55 km. This net ozone loss in the upper stratosphere and mesosphere is mainly due to odd hydrogen ($HO_x$) catalytic destruction cycles (Reactions R6 and R7) (Jackman et al., 2005) and is short-lived. Since $HO_x$ species have a shorter lifetime, the recovery of ozone is faster. For the extreme event, after the event stops, ozone enhancements upto 25 % is observed in the mesosphere and above 80 km.

NO is quite long lived down at 60 km altitude and below so the ozone loss due to the $NO_x$ catalytic cycle seems to be persistent. The ozone loss occurs during day time when NO reacts with ozone to form $NO_2$, which is then photolysed back to NO and this catalytic cycle between NO and $NO_2$, related to daytime continues (see Reactions R12 and R13). In the beginning, the amount of $\tilde{NO}_y$ was not enough for ozone loss but at the end of the event, the accumulation is large enough for significant ozone depletion which stays on and produces a diurnal cycle between 40 km and 80 km due to $NO_x/HO_x$ cycles as seen from the figure 21. The magnitude of depletion with respect to the diurnal cycle and altitude has some considerable variation. Verronen et al. (2005) used a one dimensional chemical model, Sodankylä Ion and neutral Chemistry model (SIC) for ionospheric D-region studies. They studied the effects of ion-chemistry on the neutral atmosphere and also the diurnal variation of $NO_x/HO_x$ increases, as well as ozone depletions. This diurnal variation during a solar proton event or an energetic particle precipitation (EPP) event has been previously reported by Aikin and Smith (1999). Verronen et al. (2005) observed the diurnal cycle of $O_x=O+O_3$ depletion between 40 and 85 km. They found substantial ozone loss at sunset of 28 October and even greater loss at sunrise of 29 October followed by a recovery at 55–75 km during the noon and afternoon hours. The maximum depletion is

reached just after sunset, with a 95% reduction in the $O_x$ values at 78 km. During daytime on 30 October, $O_x$ partly recovers but is again depleted during sunset. Rohen et al. (2005) also studied the Halloween SPE using SCHIAMACHY observations and considered 60°N magnetic latitude, which compares quite well to our 67.5°N geographic latitude. Since SCHIAMACHY measures only during daytime, they don't see a diurnal cycle in their results. With SCHIAMACHY, they reported a 20-30% ozone loss at 40-50 km in the Northern Hemisphere during the event and a 20-40% ozone loss, also during the event at 40-55 km observed by a photochemical model. This is related to the $NO_x$ catalytic cycle that was long lived. Above 50 km and at higher altitudes, ozone recovery was faster after 50-60% loss during the event observed with SCHIAMACHY and the model which was due to the short lifetime of $HO_x$ and photolytical reproduction of ozone. In our case, the continuing ozone loss at 40-55 km, related to the diurnal variation of $NO_x$ is found to be 60-80% for the extreme scenario as compared to the Halloween event which is just around 20%. At 60-80 km, 80-100% ozone loss is observed during the event and also the continuing loss due to the $HO_x$ related diurnal cycle afterwards. Other two examples that provide valuable insights into the significant ozone variations that can occur during extreme space weather events were studied by Calisto et al. (2013) and Rodger et al. (2008). Calisto et al. (2013) investigated the potential effects of a Carrington-like solar event on ozone using a global 3D chemistry-climate model SOCOL v2.0. They found that the enhanced ionisation during the event led to substantial ozone depletion in the polar regions, particularly in the middle atmosphere. Due to the $NO_x$ and $HO_x$ enhancements, ozone depletion was found to be 60% in the mesosphere and 20% in the stratosphere for several weeks after the event started. They also showed total ozone decreased more than 20 DU in the northern hemisphere. Rodger et al. (2008) examined the relationship between SPEs and ozone depletion using ground-based observations and modeling. They used the Sodankylä Ion and Neutral Chemistry (SIC) model and investigated the Carrington event of August/September 1859 and found that SPEs can cause localized ozone depletion in the polar regions, primarily through the production of $NO_x$. The most important SPE-driven atmospheric response is an unusually strong and long-lived $O_x$ decrease in the upper stratosphere ($O_x$ levels drop by 40%) primarily caused by the very large fluxes of >30 MeV protons. Considering these studies, it is crucial to recognize that the ion chemistry processes during SPEs can lead to ozone changes that go beyond what is typically captured in fixed parameterizations or standard models. The enhanced ionisation and subsequent chemical reactions can influence ozone concentrations, particularly in the polar regions. Therefore, when studying the impact of SPEs on ozone, it is important to consider the effects of ion chemistry processes and their potential role in generating substantial ozone variations. By incorporating these processes into models, we can better understand the complex interplay between extreme space weather events, ion chemistry, and ozone dynamics, ultimately improving our ability to assess the impacts of such events on Earth's atmosphere.

## 5   Impact of chlorine ions on ozone loss

The evaluation of the model results with MIPAS observations gave us confidence in the model. Thus, the impact of chlorine ion-chemistry on ozone loss could be assessed using the model. According to the model, we found the ozone loss in the stratosphere and lower mesosphere during the event. Figure 22 shows the relative difference of the ion-chemistry model with $O(^1D)$ in photo-chemical equilibrium including chlorine ions w.r.t. the model without chlorine ions for daytime and nighttime.

The difference in this case here is calculated for daily averaged data for each day. A loss of 2.5 % during daytime at an altitude range of 40-60 km and about 10 % during nighttime, at an altitude of 50-70 km is observed during the event day. Negative chlorine species directly increase the concentrations of uncharged active chlorine compounds. Through their catalytic

cycles, these uncharged chlorine compounds can be responsible for ozone loss at different altitudes which is also dependent on illumination conditions. The $ClO_x$ catalytic cycle (R15 and R16) is responsible for the ozone loss at 40-50 km. There is a slight difference between day-time and nighttime in the loss observed in terms of altitude range, which is expected. This difference can be explained by the difference of the diurnal cycle of ClO during daytime and nighttime. The catalytic ozone loss cycles relevant in the stratosphere-mesosphere are the ClO+O (Reactions R15, R16 and R17) and ClO+$HO_2$ (Reactions R18, R21 and

R19) that also need solar light, since O is formed by photolysis. During daytime, ClO photolyses in the mesosphere but not in the stratosphere, so ClO is not observed in the mesosphere. The ClO+$HO_2$ cycle produces HOCl which also photolyses during daytime producing Cl through Reaction R20. So during daytime, Cl is more important than ClO. But during nighttime, ClO accumulates in the mesosphere and in the stratosphere, it is mainly $ClONO_2$ due to Reaction R32. As we see from our results, during the event day on October 28 and also on October 29, the ClO mixing ratios were found to be higher for night-time

around 60-70 km compared to daytime. Hence, the ozone loss occurs more in the upper stratosphere-mesosphere around 50-70 km for nighttime, thereby producing the difference of the ozone loss in the altitude range. Loss of 0.6% during day-time and 2% during nighttime is observed in the altitude range of 30-40 km. The $ClO_x$ cycles with Reactions R28, R29, R30 and Reactions R32, R33, R34 and R35 are responsible at this altitude range for both daytime and nighttime. Furthermore, a continuous ozone formation of 2% both during daytime and nighttime is observed. This increase is linked to enhanced atomic oxygen production

by $O_2$ photolysis in solar maximum conditions (Marsh et al., 2007). It is observed at an altitude range of 60-70 km for daytime and 50-70 km for nighttime.

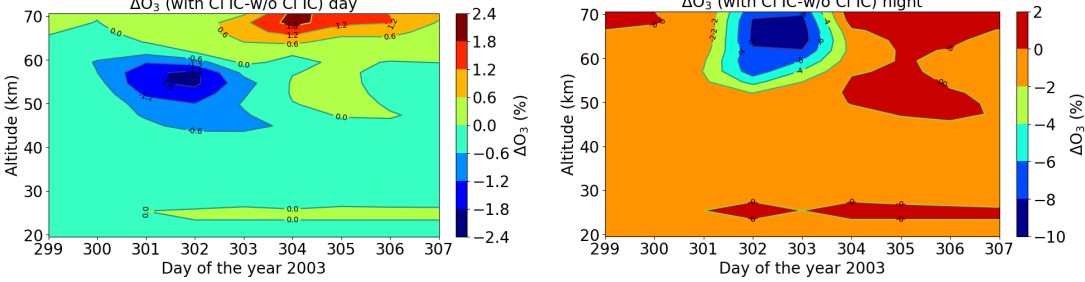

**Figure 22.** Relative difference of the model with full ion-chemistry and $O(^1D)$ in photo-chemical equilibrium including chlorine ions w.r.t. the model without chlorine ion-chemistry for the Halloween SPE: daytime (left) and nighttime (right). The difference here is calculated for daily averaged data.

Figure 23 shows the relative differences of the model setting with ion-chemistry and $O(^1D)$ in photo-chemical equilibrium w.r.t. the model setting without chlorine ion-chemistry comparing the Halloween SPE and the extreme event in order to assess the impact of chlorine ion-chemistry on ozone loss during the event day. An impact of the chlorine ions around 10-20% is

observed on the event day. Qualitatively, it was a bit more for the extreme event compared to the Halloween SPE that could

be more important for higher forcing. An increase of around 5% for ozone is also seen after the event stops for the extreme scenario. The extreme run doesn't show any impact of chlorine ion-chemistry on ozone loss at 70-75 km while the Halloween SPE does. This can be explained by the ClO and HOCl responses to SPEs and due to the $ClO_x$ catalytic cycle. At an altitude of 70-75 km, ClO decrease for the sensitivity runs for both ion-chemistry with $O(^1D)$ in photo-chemical equilibrium and without

chlorine ions is larger for the extreme event compared to the Halloween SPE (figure 16). This is one contributing factor as to why we don't see an impact on ozone loss at these altitudes. Kalakoski et al. (2020) used WACCM-D to investigate ozone depletion around 50-60 km after the onset of the SPE as explained in Sect. 4. They studied the effect in both the hemispheres and the duration and altitude range of this extra ozone loss correspond to $NO_x$ and enhanced $Cl_x$(=Cl+ClO) mixing ratios. An ozone loss of 0.2 ppm in both the hemispheres was observed after the event. Around 70 km, the ozone loss was due to short

lived $HO_x$. And around 50 km, it is driven by $NO_x$ and $Cl_x$ that lasts longer with maximum ozone decrease seen about 30 days after the event onset. Since HCl response to SPEs is partly due to chlorine ion-chemistry which converts it to Cl, ClO and HOCl (Winkler et al., 2009), this is also indirectly an effect of the chlorine ion-chemistry. They also see an increase in ozone, around 0.2 ppm throughout the period near the secondary ozone maximum above 80 km, which is also due to $O_2$ photolysis as discussed above. We observe a continuous increase of $O_3$ after the event, which is about 5%. This increase was seen around

50-60 km for the Halloween SPE and 50-75 km for the extreme scenario.

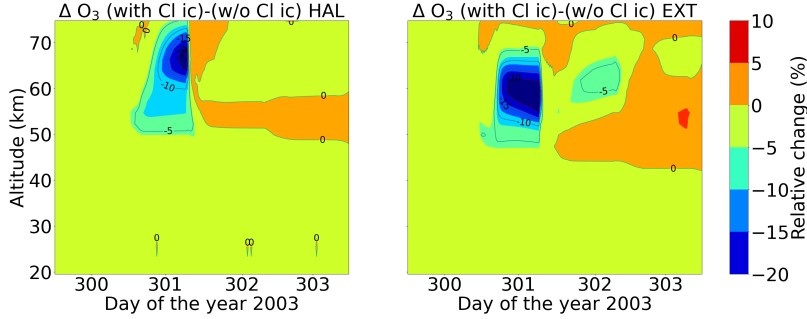

**Figure 23.** Relative difference of the model simulations: full ion-chemistry with $O(^1D)$ in photo-chemical equilibrium and with chlorine ions w.r.t. the model setting without chlorine ion-chemistry comparing the Halloween SPE (left) and extreme scenario (right). The data shown here is not daily averaged but the real model time step.

## 6    Discussions and conclusions

Using satellite data (MIPAS on ENVISAT), the state of the art 1D ion-chemistry ExoTIC model has been validated. Two event classes were modelled and chlorine ion-chemistry and its impact on ozone has been evaluated. ExoTIC has been used to study the short-term constituent changes caused by the Halloween SPE of 2003. The results demonstrated here show a comparison

of stratospheric and mesospheric composition changes observed by MIPAS in the polar cap region with simulations performed by ExoTIC. The inter-comparison of the model and MIPAS observations allowed for an evaluation of the overall ability of the ExoTIC model to reproduce observed atmospheric perturbations generated by solar proton events, particularly with respect to

changes in the chlorine species, SPE induced $NO_y$ enhancements and ozone depletion. Polar upper stratospheric and lower mesospheric $NO_y$ increased by over 40 ppbv and mesospheric ozone decreased by over 70 % during the SPE period. The inter-comparison also tested and identified deficiencies in the chemical schemes, particularly with respect to nitrogen and chlorine chemistry which is relevant for stratospheric ozone. Short-time enhancements of HOCl and $ClONO_2$ were observed by MIPAS and also reproduced in the simulations with different model settings. Application of MIPAS averaging kernels to the ExoTIC output made the inter-comparison much better. HOCl enhancements were reproduced best for the model simulation with ion-chemistry where $O(^1D)$ was set to photo-chemical equilibrium. An HOCl enhancement of more than 0.2 ppbv occurred roughly during the event as observed by MIPAS. The HOCl enhancements with averaging kernels as found in ExoTIC and its temporal variation agree quite well with MIPAS. The ozone depletion simulated by the model extends over a large altitude range as compared to MIPAS observations. Jackman et al. (2008) found the opposite from WACCM3 results in the northern hemisphere in terms of altitude range. The encountered differences between the model and observations for $ClONO_2$ enhancements, it's underestimation by the model are related to a smaller availability of ClO in the polar region before the SPE. The $ClONO_2$ peak is observed at the same altitude for both the model and MIPAS, but the enhancements in MIPAS is observed earlier and the peak values seem to agree quite well even without the averaging kernels applied. Jackman et al. (2008) also found the $ClONO_2$ peak at the same altitude with WACCM3 but the values were underestimated by a factor of two compared to MIPAS.

The comparison of the Halloween SPE and the extreme solar event of 775 A.D. showed long lasting stratospheric ozone loss for the extreme scenario. A long lasting impact was also found for the chlorine species like HOCl and HCl in case of the extreme scenario. Loss of HCl was underestimated by the parameterised model which was also found by Winkler et al. (2009) during the solar proton event in July 2000 in the northern polar region. For the extreme event, the parameterised model showed much higher $NO_y$ enhancements, about a 1000 ppm in the mesosphere and lower thermosphere. $HO_x$ enhancements of 0.1 ppm was found during the extreme event which went further down in altitude upto 40 km, for all the model case studies. An impact of around 10-20 % on ozone loss was found due to the chlorine ions for the two events, a bit stronger for the extreme scenario that is more important for higher forcing. Ozone formation was observed after the event which is also due to the impact of chlorine ion-chemistry. For the Halloween event with temporal ionisation rates, ozone loss of 2.4 % during day-time and 10 % during nighttime was observed during the event that is also due to the included chlorine ion-chemistry. Ozone formation of 2-4 % was found after the event both during day-time and nighttime.

In general, ExoTIC simulations reproduced the impacts of the Halloween SPE quite well, mainly for HOCl and $NO_y$. However, the initial state of the atmosphere in the simulations could be an important factor for some variability in the model results and MIPAS observations. Future work will focus on including the D-region ion-chemistry into the global 3D chemistry climate model EMAC (ECHAM/MESSy) and the evaluation of the chemistry with MIPAS observations in a setup considering atmospheric dynamics. While previous results with UBIC focused on the solar proton event in July 2000 in the northern polar region and compared to the HCl measurements from Halogen Occultation Experiment instrument (HALOE), we compare with MIPAS observations. Since MIPAS observations provide a better picture of the polar cap region as compared to HALOE observations that are less densely sampled, these results suggest that the validated D-region ion-chemistry setup in the ExoTIC model can be trusted to implement in a global 3D model. The problem of $O(^1D)$ in the ion-chemistry in ExoTIC should be taken

into account which was however related to the neutral chemistry model as explained in Sect. 2.2.1. For a global 3D chemistry climate model, at least in EMAC (ECHAM/MESSy), that we are considering for the implementation, the ion-chemistry is part of the normal chemistry solver and in that case, the $O(^1D)$ formation rate should be not too large and it should work without doing anything to $O(^1D)$. This setup will be first evaluated with MIPAS observations and since EMAC can already provide the data at the MIPAS footprints, the modelled data can be sampled at MIPAS measurement local times. The model will then considered for experiments in different setups to look at the dynamical impacts with and without the D-region ion-chemistry with important chemical reactions involving water, chlorine and $NO^+$ cluster ions.

In this Appendix, the reactions involving the chlorine ions and their rate coefficients used in ExoTIC is listed.

**Table A1.** T is the temperature in K, and M is the total air density in $cm^{-3}$

| Reactants | Products | Rate coefficient | Source |
|---|---|---|---|
| $Cl^- + Cl_2$ | $Cl_3^-$ | $9\times10^{-30}\times(M)$ | Amelynck et al. (1994) |
| $Cl^- + CO_2$ | $Cl^-(CO_2)$ | $6\times10^{-29} \times \left(\frac{300}{T}\right)^2\times(M)$ | Kopp and Fritzenwallner (1997) |
| $Cl^- + H_2O$ | $Cl^-(H_2O)$ | $2\times10^{-29} \times \left(\frac{300}{T}\right)^2\times(M)$ | Turco (1977) |
| $Cl^- + HCl$ | $Cl^-(HCl)$ | $1\times10^{-27}\times(M)$ | Kazil et al. (2003) |
| $Cl^-(CO_2)$ | $Cl^- + CO_2$ | $2.6\times10^{-5} \times \left(\frac{300}{T}\right)^3 \times e^{\frac{-4000}{T}}\times(M)$ | Kopp and Fritzenwallner (1997) |
| $Cl^-(H_2O)$ | $Cl^- + H_2O$ | $9.2\times10^{-5} \times \left(\frac{300}{T}\right)^3 \times e^{\frac{-7450}{T}}\times(M)$ | Kopp and Fritzenwallner (1997) |
| $Cl^-(HCl)$ | $Cl^- + HCl$ | $3.33\times10^{-5} \times \left(\frac{300}{T}\right) \times e^{\frac{-11926}{T}}\times(M)$ | Kopp and Fritzenwallner (1997) |
| $NO_3^- + HCl$ | $NO_3^-(HCl)$ | $5.22\times10^{-28} \times \left(\frac{300}{T}\right)^{2.62}$ | Kopp and Fritzenwallner (1997) |
| $OH^- + HCl$ | $Cl^- + H_2O$ | $1.5\times10^{-9} \times \left(\frac{300}{T}\right)^5$ | Kopp and Fritzenwallner (1997) |
| $Cl^- + ClONO_2$ | $NO_3^- + Cl_2$ | $9.2\times10^{-10}$ | Turco (1977) |
| $Cl^- + HNO_3$ | $NO_3^- + HCl$ | $2.8\times10^{-9}$ | Huey (1996) |
| $Cl^- + H$ | $e + HCl$ | $9.6\times10^{-10}$ | Turco (1977) |
| $Cl^- + N_2O_5$ | $NO_3^- + ClNO_2$ | $9.4\times10^{-10}$ | Amelynck et al. (1994) |
| $Cl^-(H_2O) + Cl_2$ | $Cl_3^- + H_2$ | $1.09\times10^{-9}$ | Kopp and Fritzenwallner (1997) |
| $Cl^-(H_2O) + HCl$ | $Cl^-(HCl) + H_2O$ | $1.30\times10^{-9}$ | Kopp and Fritzenwallner (1997) |
| $Cl^-(H_2O) + HNO_3$ | $NO_3^-(HCl) + H_2O$ | $2.85\times10^{-9}$ | Kopp and Fritzenwallner (1997) |
| $Cl^-(H_2O) + H$ | $e + H_2O + HCl$ | $8\times10^{-11}$ | Kopp and Fritzenwallner (1997) |
| $Cl^-(HCl) + Cl_2$ | $Cl_3^- + HCl$ | $5.3\times10^{-10}$ | Kopp and Fritzenwallner (1997) |
| $Cl^-(HCl) + HNO_3$ | $NO_3^-(HCl) + HCl$ | $2.48\times10^9$ | Kopp and Fritzenwallner (1997) |
| $Cl^- + NO_2$ | $NO_2^- + Cl$ | $6.0\times10^{-12}$ | Kopp and Fritzenwallner (1997) |
| $Cl^- + O_3$ | $ClO^- + O_2$ | $5.0\times10^{-13}$ | Turco (1977) |
| $Cl_2^- + HNO_3$ | $NO_3^-(HCl) + Cl$ | $4.8\times10^{-10}$ | Amelynck et al. (1994) |
| $Cl_2^- + NO_2$ | $Cl^- + ClNO_2$ | $4.0\times10^{-11}$ | Kopp and Fritzenwallner (1997) |
| $Cl_2^- + O_3$ | $O_3^- + Cl_2$ | $2.0\times10^{-12}$ | Kopp and Fritzenwallner (1997) |
| $Cl_3^- + HNO_3$ | $NO_3^-(HCl) + Cl_2$ | $1.3\times10^{-9}$ | Amelynck et al. (1994) |
| $CO_3^- + ClONO_2$ | $NO_3^- + ClO + CO_2$ | $2.1\times10^{-9}$ | Kopp and Fritzenwallner (1997) |
| $CO_3^- + HCl$ | $Cl^- + OH + CO_2$ | $3.0\times10^{-11}$ | Kopp and Fritzenwallner (1997) |
| $CO_4^- + HCl$ | $Cl^-(HO_2) + CO_2$ | $1.2\times10^{-11}$ | Kopp and Fritzenwallner (1997) |

| Reactants | Products | Rate coefficient | Source |
|---|---|---|---|
| $NO_2^- + Cl_2$ | $Cl_2^- + NO_2$ | $6.8 \times 10^{-10}$ | Kopp and Fritzenwallner (1997) |
| $NO_2^- + HCl$ | $Cl^- + HNO_2$ | $1.4 \times 10^{-9}$ | Kopp and Fritzenwallner (1997) |
| $NO_3^- + HCl$ | $Cl^- + HNO_2$ | $1.0 \times 10^{-12}$ | Kopp and Fritzenwallner (1997) |
| $NO_3^-(HCl) + HNO_3$ | $NO_3^-(HNO_3) + HCl$ | $7.6 \times 10^{-10}$ | Amelynck et al. (1994) |
| $O^- + HCl$ | $Cl^- + OH$ | $2.0 \times 10^{-9}$ | Turco (1977) |
| $O_2^- + HCl$ | $Cl^- + HO_2$ | $1.6 \times 10^{-9}$ | Turco (1977) |
| $O_3^- + Cl_2$ | $Cl^- + ClO + O2$ | $1.3 \times 10^{-9}$ | Turco (1977) |
| $ClO^- + NO$ | $Cl^- + NO_2$ | $2.9 \times 10^{-11} \times 0.5$ | Turco (1977) |
| $ClO^- + O_3$ | $Cl^- + O_2 + O_2$ | $6.0 \times 10^{-11}$ | Turco (1977) |
| $ClO^- + O_3$ | $ClO + O_3^-$ | $1.0 \times 10^{-11}$ | Turco (1977) |
| $O^- + Cl$ | $Cl^- + O$ | $1.0 \times 10^{-10}$ | Turco (1977) |
| $O^- + ClO$ | $Cl^- + O_2$ | $1.0 \times 10^{-10}$ | Turco (1977) |
| $O_2^- + Cl$ | $Cl^- + O_2$ | $1.0 \times 10^{-10}$ | Turco (1977) |
| $O_2^- + ClO$ | $ClO^- + O_2$ | $1.0 \times 10^{-10}$ | Turco (1977) |
| $OH^- + Cl$ | $Cl^- + OH$ | $1.0 \times 10^{-10}$ | Turco (1977) |
| $OH^- + ClO$ | $ClO^- + OH$ | $1.0 \times 10^{-10}$ | Turco (1977) |
| $CO_3^- + Cl$ | $Cl^- + O + CO_2$ | $1.0 \times 10^{-10}$ | Turco (1977) |
| $CO_3^- + Cl$ | $ClO^- + CO_2$ | $1.0 \times 10^{-10}$ | Turco (1977) |
| $CO_3^- + ClO$ | $Cl^- + CO_2 + O_2$ | $1.0 \times 10^{-10}$ | Turco (1977) |
| $CO_4^- + Cl$ | $Cl^- + O_2 + CO_2$ | $1.0 \times 10^{-10}$ | Turco (1977) |
| $CO_4^- + ClO$ | $ClO^- + O_2 + CO_2$ | $1.0 \times 10^{-10}$ | Turco (1977) |
| $NO_2^- + Cl$ | $Cl^- + NO_2$ | $1.0 \times 10^{-10}$ | Turco (1977) |
| $NO_2^- + ClO$ | $Cl^- + NO_3$ | $1.0 \times 10^{-10}$ | Turco (1977) |
| $HCO_3^- + Cl$ | $Cl^- + OH + CO_2$ | $1.0 \times 10^{-10}$ | Turco (1977) |
| $HCO_3^- + ClO$ | $Cl^- + HO_2 + CO_2$ | $1.0 \times 10^{-10}$ | Turco (1977) |
| $ClO^- + O$ | $Cl^- + O_2$ | $2.0 \times 10^{-10}$ | Turco (1977) |
| $H^+ + Cl^-$ | $Cl$ | $6 \times 10^{-8} \times \frac{300}{T})^{0.5} + 1.25 \times 10^{-25} \times$ $\left(\frac{300}{T}\right)^4 \times (M)^*$ | Arijs et al. (1987) |
| $H^+ + Cl_2^-$ | $Cl_2$ | | |
| $H^+ + Cl_3^-$ | $Cl_2 + Cl$ | | |
| $H^+ + Cl^-(HCl)$ | $Cl + HCl$ | | |
| $H^+ + Cl^-(H_2O)$ | $Cl + H_2O$ | | |

* The coefficient is the same for all the reactions below

| Reactants | Products | Rate coefficient | Source |
|---|---|---|---|
| $H^+ + Cl^-(CO_2)$ | $Cl + CO_2$ | | |
| $H^+ + Cl^-(HO_2)$ | $Cl + HO_2$ | | |
| $H^+ + ClO^-$ | $ClO$ | | |

*Code availability.* The Exoplanetary Terrestrial Ion Chemistry (EXoTIC) is continuously developed and applied in the group 'Middle Atmosphere Solar Variability and Climate Interactions (MSK)' at IMK-ASF. The exact code version used to produce the simulation results can be made available upon request from Miriam Sinnhuber (miriam.sinnhuber@kit.edu).

*Data availability.* The "raw" MIPAS V5H data set, i.e. ClONO2, HNO3, HOCl, N2O5, NO, NO2, and O3 vertical profiles and corresponding averaging kernel matrices for October and November 2003 is available on doi: 10.5445/IR/1000156935.

*Author contributions.* MB and MS discussed the ideas. MB wrote the paper. MB and MS worked on the code and simulation results for the ExoTIC ion-chemistry model. MB worked on the visualisation and analysis of the MIPAS data and AL helped with the MIPAS averaging kernels. TC, GS and BF provided access to MIPAS data and helped with technical questions regarding the correct use of MIPAS data. JW
and OY developed AISstorm and provided us with the ionisation rates. IU calculated the ionisation rates for the extreme solar event of 775 A.D. All authors contributed to reviewing and editing the manuscript.

*Competing interests.* At least one of the co-authors is a member of the editorial board of Atmospheric Chemistry and Physics.

*Acknowledgements.* This research has been supported by the German Research Foundation (DFG) under the project SI 1088/7-1. The AISstorm model is funded by the German Science Foundation (DFG project WI4417/2-1). J.M. Wissing also thanks to support from the
German Aerospace Center (DLR). The authors acknowledge the NOAA National Centers for Environmental Information (https://ngdc.noaa.gov/stp/satellite/poes/dataaccess.html) for the POES and Metop particle data used in this study.

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
