# Peer review of "Impact of chlorine ion chemistry on ozone loss in the middle atmosphere during very large solar proton events"

_EGUsphere, 2023_

## Author Response (AR1)

**Point by point response to reviews and changes made in impact of chlorine ion chemistry on ozone loss in the middle atmosphere during very large solar proton events**

Monali Borthakur[1], Miriam Sinnhuber[1], Alexandra Laeng[1], Thomas Reddmann[1], Peter Braesicke[1], Gabriele Stiller[1], Thomas von Clarmann[1], Bernd Funke[2], Ilya Usoskin[3], Jan Maik Wissing[4], and Olesya Yakovchuk[4]

[1]Institute of Meteorology and Climate research, Karlsruhe Institute of Technology, Karlsruhe, Germany
[2]Instituto de Astrofísica de Andalucía, CSIC, Granada, Spain
[3]University of Oulu, Oulu, Finland
[4]University of Rostock, Rostock, Germany

Dear editor,

Below are the responses of the authors to the reviewers and the changes in the revised manuscript in blue indicating the line number from the document with tracked changes. The document is structured in the same way as the author's response to the reviewers. And other relevant changes in the abstract and different sections are also added below for which the line numbers are indicated.

**Response to referee 1:**

Dear referee,

Thank you for your thorough review of our work and the numerous comments. Here are our responses to your comments below.

– Major changes include: A comprehensive revision of the text concerning recent publications on ion-chemistry and chlorine species and revision of the abstract and summary. The figures are re-evaluated. Research implications and recommendations are also added in the conclusions.

– The comments are numbered and in bold and each of them is followed by a response from the authors.

Yours sincerely and on behalf of all co-authors,
Monali Borthakur

**1 Detailed Remarks:**

**1.1 L 91-92: "The chlorine negative ion is the most important ion of the lower D region during day and night time." I don't think this is true. In the winter polar region, large amounts of $NO_x$ and lower level of dissociating solar radiation contribute to creation of $NO_3^-$ and its clusters, for example. For daytime and at lower latitudes, chlorine ions are abundant in the mesosphere but not dominant, I would say. The Authors could actually check this easily from their model results.**

You are right. The composition and dominance of ions in different regions of the ionosphere vary depending on several factors, including altitude, latitude, time of day, and season. While chlorine ions can be present and play a role in the mesosphere, they are not typically considered the most important ions in the lower D region during both day and night. So, you are right and we restructured the sentence to "The chlorine negative ion is an abundant ion of the lower D region during day and night time" in the revised manuscript.

Line 102-103: "The chlorine negative ion is an abundant ion of the lower D region during daytime and nighttime".

**1.2 L 170: "The ion-chemistry is based on the UBIC (University of Bremen ion-chemistry) model developed by Winkler et al. (2009) for the terrestrial middle atmosphere." The Authors could briefly describe the main development and changes between UBIC and Exotic, if any.**

UBIC (University of Bremen Ion-Chemistry model) is a model developed by the University of Bremen to simulate ion-chemistry processes in Earth's atmosphere. The UBIC model as described by Winkler et al. (2009) simulates the time evolution of 137 species, 55 positive, 49 negative and 33 neutral species and computes more than 600 reactions. ExoTIC (Exoplanetary Terrestrial Ion-chemistry) builds upon the foundations of UBIC but is extended to study the ion-chemistry of exoplanetary atmospheres with a wide range of orbital parameters, stellar systems and base compositions as discussed by Herbst et al 2019. In the ion-chemistry part of ExoTIC, we have added more neutral species compared to what was present in Winkler et al 2009. Some details have been added to the revised manuscript.

Line 209-213: The ExoTIC model extends the applicability of UBIC to atmospheres of (rocky) planets other than Earth with a wide range of orbital parameters, stellar systems and base compositions as discussed by Herbst, Konstantin et al. (2019). More neutral species have been added to the ion-chemistry in ExoTIC since studies by Winkler et al. (2009). The ionisation of $CO_2$ was recently included. Another small change is that the equilibrium is calculated for the ions, which stabilised the model.

**1.3** **Section 3.1. There are indeed methods for estimation of the polar vortex edge vortex from the gradient of daily**
45 **CO mixing ratio (e.g. Harvey et al. 2018, https://doi.org/10.1029/2018JD028815). Also, Funke (2005) and the**
**gradient method is mentioned by the Authors, so why use a fixed CO mixing ratio? Also, CO is good for the**
**mesosphere but the Authors also show a lot of results from the stratosphere where the potential vorticity would**
**be a better vortex measure, I assume. Please comment on this.**

Funke et al. (2005) indeed use a vertically and temporally varying vortex boundary and this was determined from the CO
50 latitudinal gradient. We think it was a mistake to use the Funke et al. (2005) paper as a reference for the method chosen here.
Regarding the fixed CO threshold of 0.5 ppmv, a chemical vortex definition (via CO) is widely used in the mesosphere (70
km) as also studied by Harvey et al. (2015). And as you pointed out, below in the stratosphere the vortex is indeed commonly
defined via the potential vorticity gradient from reanalysis data. Since we are determining the polar vortex in the MIPAS orbit,
it is better to use MIPAS data as compared to reanalysis data. That is why we used MIPAS CO both in the stratosphere and the
55 mesosphere and also due to the fact that potential vorticity is not provided together with the MIPAS data. We have included
details and revised the manuscript accordingly.

Line 351-357: Here, we have used a CO vmr threshold (discriminating mesospheric air from the background) as vortex
criterion. We need an altitude-independent criterion (as we need to differentiate entire profiles) and our short time period (a
few days) allows for a time-independent definition. A chemical vortex definition (via CO) is also widely used in the mesosphere
60 (Harvey et al. 2015). And for the chemical species and the SPE responses discussed here, the relevant altitude range is more in
the stratosphere. In the stratosphere, the vortex might be considerably smaller and is commonly determined from the potential
vorticity. Here, we use the tracer gradient instead to be consistent with the MIPAS observations that we use for comparison.
Line 363-366: One can also check the vortex boundary by looking at $CH_4$ zonal means, for example, Figure 9 of Funke et al.
(2011). From that figure and as also discussed in Funke et al. (2011), the boundary is around 60°N which also works well for
65 the stratosphere. We chose the latitude 57°N as to where the vortex begins and defined the latitude bands 57-77°N as "the edge
region of the vortex" and the high latitude bands 70-90°N as "deep in the vortex".

**1.4** **Figure 5 (and similar figures of MIPAS comparisons). For clarity, I would suggest to remove the model data**
**without-averaging-kernels-applied and harmonize the color scales. Firstly, the x axis is almost unreadable.**
**Secondly, if averaging kernel impact needs to be shown then one good example should do it. Thirdly, changing**
70 **the color scales makes the plot confusing (at least to me). Finally, it's not necessary to plot the same MIPAS data**
**four times. Maybe it would make sense to plot the difference between the model runs and MIPAS?**

We decided to keep the model calculations without-averaging-kernels-applied for the following reason: what MIPAS sees, is
a distorted picture of the "truth". We apply the same distortion to the model data when applying the averaging kernels. If AK-
convolved model results and MIPAS observations fit, then the original model results represent the "truth" (at least one of the
75 possible ones). Therefore, not the MIPAS results represent the true distribution of trace gases in the atmosphere, but the original
model result. The data plotted here are the differences w.r.t. 26[th] October 2003, a day before the event. However, we also added

the absolute differences between model and MIPAS data in the supplement. In the revised manuscript, the figures are enlarged and we decided to keep the format with MIPAS data along each row because we thought it's better for the comparison of the different sensitivity runs. The color scales are harmonized in the revised version and the x-axis ticks are also enlarged.

80      In the revised manuscript: Figures 5-12. And for the absolute differences between model and MIPAS data in the supplement: figures 5-8.

**1.5 Figure 5 (and similar figures). It seems to me that plotting the results from the "full ion chemistry" simulation makes no sense because that model setup cannot handle O(1D) properly. The Authors basically admit this and it's something that was already noted by Winkler (2009, and the follow-up correction paper). Therefore, I would**

85      **suggest to remove all data from the "full ion chemistry" simulation and clarify in the model description why "O(1D) in photo-chemical equilibrium" is necessary and sound approach.**

We removed the full ion-chemistry plots from the main text and put them in a supplement but kept the discussion of the results from full ion-chemistry in the main text. We made some changes in the model description discussing more on "O(1D) in photo-chemical equilibrium" as a necessary and sound approach in Sect. 2.2.1.

90      Full ion-chemistry plots for the MIPAS comparison are in the supplement: figures 1-4.

     **Line 243-245**: The simulation results are sensitive to the changes of the $O(^1D)$ branching ratio, $\beta = \Delta O(^1D)/(\Delta O(^1D) + \Delta O(^3P))$ which is discussed in Winkler et al. (2009). Winkler et al. (2011) has reported the $O(^1D)$ corrections in the UBIC model in detail.

95

     **Line 249**: Basically $O(^1D)$ is produced through the reaction R1.

     **Line 251-253**: The time constants for the quenching $O(^1D) + M \longrightarrow O(^3P) + M$ in the stratosphere are significantly smaller than the chosen integration time step in the ion-chemistry model which was also reported in Winkler et al. (2011). That causes

100      too high $O(^1D)$ concentrations and an unrealistically strong effect through reaction R37.

**1.6 Figure 5. Why is there a strong peak in HOCL increase at about 35 km in the daytime simulation panels?**

The enhanced HOCl production is due to the increased availability of chlorine atoms due to the catalytic ozone destruction cycle. The strong peak in HOCl increase at about 35 km during daytime is because this altitude represents the optimal conditions for the ClOx cycle to occur. More details are included in the revised manuscript in Sect 1.3 and Sect 3.2.

105      **Line 121-125 (along with the reactions below)**: The catalytic cycle involving hypochlorous acid (HOCl) and ClO acts as a link between chlorine and $HO_x$ enhancements as a result of the SPEs (Reactions R19, R20, R21, R22 and R23) (Lary, 1997). HOCl can photolyse during daytime and the OH formed can react with $O_3$ reforming $HO_2$ and Cl reforming ClO thereby recycling HOCl again through Reaction R19. This cycle mainly plays a role in the sunlit polar lower stratosphere (Lary, 1997). Reactive Cl can also be formed via reaction of OH with HCl.

$$OH + HCl \longrightarrow H_2O + Cl \tag{R18}$$

$$ClO + HO_2 \longrightarrow HOCl + O_2 \tag{R19}$$

$$HOCl + h\nu \longrightarrow Cl + OH \tag{R20}$$

$$Cl + O_3 \longrightarrow ClO + O_2 \tag{R21}$$

$$OH + O_3 \longrightarrow HO_2 + O_2 \tag{R22}$$

$$Net : 2O_3 \longrightarrow 3O_2 \tag{R23}$$

110

**Line 389-392**: This peak value occurs due to the increased availability of chlorine atoms due to the catalytic ozone destruction cycle. The HOCl concentration reaches a peak around 35 km during daytime because this altitude represents the optimal conditions for the ClO-HOCl catalytic cycle (Reactions R20, R21, R22 and R19) to occur.

**1.7 Figure 6. The data above 50 km could be excluded from the figures because nothing is going on.**

115 The new figures upto 50 km are included in the revised manuscript.

The new figures are: Figure 7 and Figure 8.

**1.8 Figure 8. Is the modeled ozone data sampled at MIPAS measurement local times? There could be variability with local time in the mesosphere.**

We did not sample the modelled ozone data at the MIPAS measurement local times. It is a 1D model and doing that probably
120 makes better sense for a 3D model. Instead we sampled in the solar zenith angle range of MIPAS. To sample the model data at the MIPAS measurement local time, a 3D model is needed. We are in the process of implementing the ion-chemistry into a global 3D chemistry climate model EMAC (ECHAM/MESSy) to test that.

**Line 753-754**: This setup will be first evaluated with MIPAS observations and since EMAC can already provide the data at the MIPAS footprints, the modelled data can be sampled at MIPAS measurement local times.

125 **1.9 Figure 9. I don't understand why the parameterised NOx production makes more NOy than the ion chemistry. The parameterization uses a fixed value of 1.25 NOx molecules/ion pair while it was shown by e.g. Nieder et al. (J. Geophys. Res. Space Physics, 119, 2137-2148, doi:10.1002/2013JA019044, 2014) that this number is an underestimation in the upper mesosphere and lower thermosphere. Also, Andersson et al. (J. Geophys. Res. Atmos., 121, 10328–10341, https://doi.org/10.1002/2015JD024173, 2016) and Kalakoski et al. (Atmos. Chem.**
130 **Phys., 20, 8923-8938, https://doi.org/10.5194/acp-20-8923-2020, 2020) have shown that in the mesosphere, the ion chemistry produces more NOx than the fixed parameterization during SPEs. Please discuss and clarify this issue.**

The rate of formation of $NO_x = N + NO$ is higher in the mesosphere if ion-chemistry is used but the ratio between N and NO to the initial formation is more to N and that means that the rate of loss is also faster and one ends up with less NO. It is

the absolute numbers of NO that determines if the rate of loss or the rate of formation is larger. Nieder et al. (J. Geophys. Res. Space Physics, 119, 2137-2148, doi:10.1002/2013JA019044, 2014) indeed showed that in terms of mixing ratio that is less. More discussions are added in the revised manuscript and also discussion of results from Andersson et al. (J. Geophys. Res. Atmos., 121, 10328–10341, https://doi.org/10.1002/2015JD024173, 2016) and Kalakoski et al. (Atmos. Chem. Phys., 20, 8923-8938, https://doi.org/10.5194/acp-20-8923-2020, 2020) are included in Section 4.1.

Line 520-529: In contrast to our results, which show the total $NO_y$ that includes $NO_x$=N+NO, Andersson et al. (2016) showed that WACCM-D with the D-region ion-chemistry predicts more $NO_x$ in the mesosphere compared to WACCM with the standard parameterisation of $NO_x$ and $HO_x$ production for the northern polar cap region, latitude > 60°N during the SPE of January 2005. Kalakoski et al. (2020) also reported the same when considering SPEs using proton flux data from satellite-based GOES observations. They considered an event in which the peak proton flux exceeded 100 particle flux units (pfu), with pfu defined as the 5 min average flux in units of particles $cm^{-2}s^{-1}sr^{-1}$ for protons with energy larger than 10 MeV. The rate of formation of $NO_x$=N+NO is indeed higher in the mesosphere when full ion-chemistry is considered, but the partitioning between the formation of N and N($^2$D) forming NO is also increasingly in favour of N, meaning that the rate of loss of NO is also faster in this altitude range when full ion-chemistry is considered. This can lead to less NO depending on the absolute rate of $NO_x$ formation, and the partitioning between N and NO in this formation.

**1.10 Figure 10. The ionic HOx production depends on the level of H2O at least in the upper mesosphere, but the parameterised HOx production is calculated with a fixed H2O profile (Andersson et al., J. Geophys. Res. Atmos., 121, 10328–10341, https://doi.org/10.1002/2015JD024173, 2016). Could H2O explain some of the differences between the parameterized-HOx and the other runs?**

The way we include the parameterisation is not using that of Andersson et al., J. Geophys. Res. Atmos., 121, 10328–10341, https://doi.org/10.1002/2015JD024173, 2016, but as 2 $HO_x$, one H and one OH per ion pair. In the upper mesosphere, above 80 km, water vapour is low. So if the ionisation rate is not present, the $HO_x$ will go into pre-event very quickly. The high value of $HO_x$ are only maintained by the continuing destruction of water vapour. So if the water vapour is completely destroyed, then there will be no more $HO_x$ formation and it will break down. This means $H_2O$ does explain some of the differences. Details about the $HO_x$ parameterisation and some explanation have been added to the description of the model in Sect. 2.2.4 and in Sect. 4.1.

Now section 2.2.3 and Line 269-278: Andersson et al. (2016) calculated the parameterised $HO_x$ production using a fixed $H_2O$ profile. In contrast, ExoTIC assumes a zero abundance of water vapour above 80 km, while below, water vapour is modelled as a pair of H and OH. The $HO_x$ formation stops when there is no more water vapour. For high water vapour, two $HO_x$ per ion pair are formed, but the rate decreases with decreasing $H_2O$. As the $H_2O$ profile used in Andersson et al. (2016) decreases strongly above 80 km, the rate of $HO_x$ production also goes to zero above 80 km. Jackman et al. (2005) also considered this and an ion pair is computed to produce less than two $HO_x$ constituents per ion pair in the middle and upper mesosphere. In ExoTIC, 2 $HO_x$ are produced per ion pair everywhere, but the production of $HO_x$ is balanced by loss of water vapour, and the production therefore stops when all water vapour is consumed, effectively also reducing the amount of $HO_x$

production in regions of low water vapor. We choose 2 $HO_x$ per ion pair because we want to mainly concentrate on middle mesosphere to stratosphere and not upper mesosphere, i.e. above 80 km.

**Line 543-545**: The high values of $HO_x$ are balanced by the continuous destruction of water vapour and when it is completely destroyed, the $HO_x$ formation breaks down. So the full parameterisation with both $NO_x$ and $HO_x$ is different but the water vapour is a limiting factor for both.

**1.11 Figure 10. The Authors discuss HOx recovery but I think it's worth to note that the SPE ionization stays at an elevated level for the duration of the simulation, i.e. also after the peak on day 302, as shown in Figure 1. Therefore, the recovery is only partial as EPP-HOx production continues albeit with a lower rate.**

For figure 10, we have applied the mean ionisation rate of October 28/29 for day 301 (just one profile), i.e. those shown in Figure 1 (right). Nevertheless, even after the ionisation/event stops, the recovery of HOx is indeed partial.

**1.12 L 458: "due to the same" => due to the chlorine ion chemistry**

We replaced "due to the same" to "due to the chlorine ion chemistry" in the revised manuscript.

**Line 557-558**: From the sensitivity study without the chlorine ions, it is evident that the huge loss of HCl, around 75% observed in the mesosphere is due to the chlorine ion-chemistry.

**1.13 Figure 11. Also Andersson et al. (J. Geophys. Res. Atmos., 121, 10328–10341, https://doi.org/10.1002/2015JD024173, 2016) have previously reported a HCl decrease from ion chemistry between 40 and 50 km in during an SPE. However, the Authors show here that a quite similar decrease is seen below 50 km even without ion chemistry. Also, the decrease at 40-50 km seems to be much larger than what was reported by Andersson et al. Perhaps the Authors can briefly discuss this matter.**

In Figure 11, the ion-chemistry shows an HCl decrease of around 10-15 % at 40-50 km for the Halloween SPE of 2003 which is indeed larger compared to what is reported by Andersson et al. (2016). Below 50 km, a 5-10 % decrease is observed even without ion-chemistry. A figure for the relative changes of HCl for the different sensitivity runs w.r.t. the reference run is included in the supplement (figure 16) to the revised manuscript. Further discussions are added in Sect. 4.2.

**Line 556-557**: At 40-50 km, ExoTIC with ion-chemistry and $O(^1D)$ set to photo-chemical equilibrium observed 5-15% HCl loss for the Halloween SPE and 20-45% HCl loss for the extreme event.

**Line 560-571**: The primary neutral reaction that leads to the decrease in HCl below 50 km is a series of reactions involving $HO_x$ species that are part of the catalytic ozone destruction cycle (Reactions R6 and R7). The decrease in ozone concentration has a secondary effect on the concentration of HCl. In the absence of an SPE, ozone plays a role in the conversion of ClO back to HCl. However, during an SPE, the enhanced ionization and subsequent formation of ions disrupt this ozone destruction and formation cycle. This leads to an increase in the concentration of ClO and a subsequent reduction in the concentration of

HCl. The excess ClO can further participate in additional ozone depletion cycles, leading to a decline in ozone levels during
the event. Andersson et al. (2016) reported daily averaged anomalies of HCl in both the hemispheres for the latitudinal band
60–82.5°N at altitudes between 40 and 52 km. They compared WACCM-D consisting of the D-region ion-chemistry and
WACCM consisting of the standard $NO_x$ and $HO_x$ parameterisation with MLS observations. They found a rapid HCl decrease
of about 2-6% during the January 2005 SPE in both hemispheres. With WACCM-D, a loss of around 4% was found compared
to a loss of 3% in standard WACCM in the Northern Hemisphere. They also showed that WACCM-D showed better agreement
with MLS observations.

**1.14 Figures 12,13,14. It seems to me that the extended ClO decrease below 50 km seen in the EXT run is balanced by increase and buildup of ClONO2, not HOCl. Could it be that NOx plays a more important role at these altitudes than HOx?**

You are right. NOx does play a more important role than HOx below 50 km. We have revised and corrected the explanations.

**Line 581-583**: This is because of the excess $NO_x$ available for the extreme event due to which R32 can happen continuously which is supplemented by an increase in $ClONO_2$ after the extreme event as seen from figure 18.

**1.15 Figure 17. "NO is quite long lived down at 40 km altitude and below so the ozone loss due to the NO catalytic cycle seems to be persistent." Based on the figure, it looks more like "at 60 km and below".**

We changed from "40 km altitude and below" to "60 km altitude and below" in the sentence which seems to be the case.

**Line 619-620**: NO is quite long lived down at 60 km altitude and below so the ozone loss due to the $NO_x$ catalytic cycle seems to be persistent.

**1.16 Figure 17. It could be useful the extend the altitude range upwards to fully see the ozone depletion region.**

The figures are updated with extended altitude region in the revised manuscript.

Now Figure 21: with the extended altitude region.

**1.17 Line 503: "ozone depletion which stays on and produces a diurnal cycle as seen from the Figure 17." The diurnal variability of ozone depletion has previously been discussed by, e.g., Verronen et al. (J. Geophys. Res., 110, A09S32, https://doi.org/10.1029/2004JA010932, 2005).**

The diurnal variation of $HO_x$/$NO_x$ related ozone depletion during solar proton events have been reported by Aikin and Smith
(1999) as well as in Verronen et al. (2005). Discussion from Verronen et al. (2005) who studied the diurnal cycle of ozone
loss during the Halloween SPE using the Sodankylä Ion Chemistry Model (SIC) and Rohen et al. (2005) who studied it using
SCHIAMACHY observations have been included. We have also included details about the differences of the magnitude of
depletion with respect to the diurnal cycle and altitude in Sect. 4.3.

**Line 625-639**: The magnitude of depletion with respect to the diurnal cycle and altitude has some considerable variation. Verronen et al. (2005) used a one dimensional chemical model, Sodankylä Ion and neutral Chemistry model (SIC) for iono-
spheric D-region studies. They studied the effects of ion-chemistry on the neutral atmosphere and also the diurnal variation of $NO_x$/$HO_x$ increases, as well as ozone depletions. This diurnal variation during a solar proton event or an energetic particle precipitation (EPP) event has been previously reported by Aikin and Smith (1999). Verronen et al. (2005) observed the diurnal cycle of $O_x=O+O_3$ depletion between 40 and 85 km. They found substantial ozone loss at sunset of 28 October and even greater loss at sunrise of 29 October followed by a recovery at 55–75 km during the noon and afternoon hours. The maximum depletion is reached just after sunset, with a 95% reduction in the $O_x$ values at 78 km. During daytime on 30 October, $O_x$ partly recovers but is again depleted during sunset. Rohen et al. (2005) also studied the Halloween SPE using SCHIAMACHY observations and considered 60°N magnetic latitude, which compares quite well to our 67.5°N geographic latitude. Since SCHIAMACHY measures only during daytime, they don't see a diurnal cycle in their results. With SCHIAMACHY, they reported a 20-30% ozone loss at 40-50 km in the Northern Hemisphere during the event and a 20-40% ozone loss, also during the event at 40-55 km observed by a photochemical model. This is related to the $NO_x$ catalytic cycle that was long lived. Above 50 km and at higher altitudes, ozone recovery was faster after 50-60% loss during the event observed with SCHIAMACHY and the model which was due to the short lifetime of $HO_x$ and photolytical reproduction of ozone.

**1.18  Line 504: "continuing ozone loss in the middle and upper stratosphere, after the event stops, is found to be still 80-100% for the extreme scenario". Based on the figure, I would say it's 60-80%. I think a brief discussion on this would be needed because ozone changes quite a lot compared the the impact calculated, e.g., for the Carrington event. See, e.g., Calisto et al. (Environ. Res. Lett. 8 (2013) https://doi.org/10.1088/1748-9326/8/4/045010), Rodger et al. (J. Geophys. Res., 113, D23302, https://doi.org/10.1029/2008JD010702, 2008).**

The ozone loss during the event is 80-100% whereas the continuing ozone loss is indeed around 60-80%. Significant ozone variations that can occur during extreme space weather events are discussed by Calisto et al. (2013) and Rodger et al (2008). They investigated the Carrington event of August/September 1859 using SOCOL 3D model and SIC model respectively. They found that SPEs can cause localized ozone depletion in the polar regions, primarily through the production of NOx and HOx. Discussions related to these studies have been added in Sect. 4.3.

**Line 639-660**: In our case, the continuing ozone loss at 40-55 km, related to the diurnal variation of $NO_x$ is found to be 60-80% for the extreme scenario as compared to the Halloween event where it is just around 20%. At 60-80 km, 80-100% ozone loss is observed during the event and also the continuing loss due to the $HO_x$ related diurnal cycle afterwards. Other two examples that provide valuable insights into the significant ozone variations that can occur during extreme space weather events were studied by Calisto et al. (2013) and Rodger et al. (2008). Calisto et al. (2013) investigated the potential effects of a Carrington-like solar event on ozone using a global 3D chemistry-climate model SOCOL v2.0. They found that the enhanced ionisation during the event led to substantial ozone depletion in the polar regions, particularly in the middle atmosphere. Due to the $NO_x$ and $HO_x$ enhancements, ozone depletion was found to be 60% in the mesosphere and 20% in

the stratosphere for several weeks after the event started. They also showed total ozone decreased more than 20 DU in the northern hemisphere. Rodger et al. (2008) examined the relationship between SPEs and ozone depletion using ground-based observations and modeling. They used the Sodankylä Ion and Neutral Chemistry (SIC) model and investigated the Carrington event of August/September 1859 and found that SPEs can cause localized ozone depletion in the polar regions, primarily through the production of $NO_x$. The most important SPE-driven atmospheric response is an unusually strong and long-lived $O_x$ decrease in the upper stratosphere ($O_x$ levels drop by 40%) primarily caused by the very large fluxes of >30 MeV protons. Considering these studies, it is crucial to recognize that the ion chemistry processes during SPEs can lead to ozone changes that go beyond what is typically captured in fixed parameterizations or standard models. The enhanced ionisation and subsequent chemical reactions can influence ozone concentrations, particularly in the polar regions. Therefore, when studying the impact of SPEs on ozone, it is important to consider the effects of ion chemistry processes and their potential role in generating substantial ozone variations. By incorporating these processes into models, we can better understand the complex interplay between extreme space weather events, ion chemistry, and ozone dynamics, ultimately improving our ability to assess the impacts of such events on Earth's atmosphere.

**1.19 Figure 18: I don't quite see any use for this figure. There is no comparison to observations, so what advantage is gained from averaging the model data for day-time and night-time? I would suggest to remove Figure 18.**

Figure 18 is one of the main result to assess the impact of chlorine ions on ozone loss calculated from the model after its validation. Since we applied temporal ionisation rates for the MIPAS comparison using the Halloween SPE, we wanted to check the contribution of the included chlorine ion-chemistry for the sensitivity run that compared best with MIPAS observations. That would be the run "ion-chemistry with $O(^1D)$ in photo-chemical equilibrium". It is not possible to calculate that from the observations. We decided to keep figure 18.

**1.20 Figure 19: "A very small impact of the chlorine ions around 10-20 % is observed on the event day." I would not call a 10-20% impact small. Also there should be an explanation why the EXT run shows no impact from chlorine ion chemistry at 70-75 km while the other run does. I suspect it would better to use the same control run as the reference for all runs when calculating ozone depletion percentages, and then compare those numbers. Otherwise some of the differences shown could be simply from using different references.**

We restructured the sentence and removed the words "very small". Regarding why the extreme run shows no impact from chlorine ion chemistry at 70-75 km while the other run does, we think it is mainly to do with the ClO response to the SPEs. Comparing the ClO for both the events, more discussion on this has been added to Sect. 5. Regarding the usage of the same control run as the reference for all runs when calculating ozone depletion percentages, that is something we did for Figure 17. Figure 19 was however to assess the impact of the included chlorine ions in the ion-chemistry model on the ozone loss for the two events. Due to this reason, we calculated the percentage difference of the model run, ion-chemistry with $O(^1D)$ in

photo-chemical equilibrium (with chlorine ion-chemistry included) w.r.t. the same model run when chlorine ion-chemistry is

295    switched off.

       **Line 690-691**: An impact of the chlorine ions around 10-20% is observed on the event day.

       **Line 693-697**: The extreme run doesn't show any impact of chlorine ion-chemistry on ozone loss at 70-75 km while the Halloween SPE does. This can be explained by the ClO and HOCl responses to SPEs and due to the $ClO_x$ catalytic cycle. At an

300    altitude of 70-75 km, ClO decrease for the sensitivity runs for both ion-chemistry with $O(^1D)$ in photo-chemical equilibrium and without chlorine ions is larger for the extreme event compared to the Halloween SPE (figure 16). This is one contributing factor as to why we don't see an impact on ozone loss at these altitudes.

**1.21    Figure 19. Kalakoski et al. (Atmos. Chem. Phys., 20, 8923-8938, https://doi.org/10.5194/acp-20-8923-2020, 2020) have reported stronger ozone depletion around 60 km when D-region ion chemistry is included. The results**

305    **presented here seem to agree qualitatively, particularly in the EXT case. Perhaps a brief comment on this can be added.**

Kalakoski et al. (2020) studied ozone depletion during SPEs in both the hemispheres with WACCM-D. They found the ozone depletion to be larger for about 5 days after the onset of the SPE at around 0.5-1 hPa. Both the duration and altitude range of this extra ozone loss correspond to enhanced $Cl_x$. Some details on this are added to Sect. 5.

310    **Line 697-706**: Kalakoski et al (2020) used WACCM-D to investigate ozone depletion around 50-60 km after the onset of the SPE as explained in Sect. 4. They studied the effect in both the hemispheres and the duration and altitude range of this extra ozone loss correspond to $NO_x$ and enhanced $Cl_x(=Cl+ClO)$ mixing ratios. An ozone loss of 0.2 ppm in both the hemispheres was observed after the event. Around 70 km, the ozone loss was due to short lived $HO_x$. And around 50 km, it is driven by $NO_x$ and $Cl_x$ that lasts longer with maximum ozone decrease seen about 30 days after the event onset. Since HCl response to SPEs

315    is partly due to chlorine ion-chemistry which converts it to Cl, ClO and HOCl (Winkler et al (2009)), this is also indirectly an effect of the chlorine ion-chemistry. They also see an increase in ozone, around 0.2 ppm throughout the period near the secondary ozone maximum above 80 km, which is also due to $O_2$ photolysis as discussed above. We observe a continuous increase of $O_3$ after the event, which is about 5%. This increase was seen around 50-60 km for the Halloween SPE and 50-75 km for the extreme scenario.

**Response to referee 2:**

Dear referee,

Thank you for your thorough review of our work and the numerous comments. Here are our responses to your comments below.

– Major changes include: A comprehensive revision of the text concerning recent publications on ion-chemistry and chlorine species and revision of the abstract and summary. The figures are re-evaluated. Research implications and recommendations are also added in the conclusions.

– The comments are numbered and in bold and each of them is followed by a response from the authors.

Yours sincerely and on behalf of all co-authors,

Monali Borthakur

**2    Major Comments**

**2.1    Figures 5-8: The panels are far too small, the font is too small, the changing colour scale in each column makes comparisons extremely difficult, and the contour intervals are impossible to read in the figures (please add interval to caption). When using set contour intervals, I recommend also using set intervals in the colourbar. Please don't change the colour scale between the vertical panels.**

We enlarged the figure panels and the fonts and also put the same colour scale for the vertical panels. We added the colour intervals to the caption. As part of our response to referee 1, we decided to remove the plots for full ion-chemistry for figures 5-8 since the ion-chemistry model with $O(^1D)$ in photo-chemical equilibrium works better with the MIPAS comparison. We added more discussions on why model with $O(^1D)$ in photo-chemical equilibrium is better in the model description. However, we added the full ion-chemistry plots in a supplement to the revised manuscript but kept the discussion of results in the text.

Enlarged figures and changes in the captions indicating the colorbar intervals: Figure 5-12.

**2.2    Figures 9-15: These figures are easier to read, but why are the full ion chemistry results not shown? Again there contour intervals are not clear here.**

We wanted to mainly show the comparison for the sensitivity studies that matched best with the MIPAS observations. But keep the discussion of the results from full ion-chemistry model without setting $O(^1D)$ to photo-chemical equilibrium. We have included the comparison of full ion-chemistry for the two events along with the other sensitivity studies in the supplement for your reference. And we have updated the figures in the revised manuscript.

Now figures 13-18 in the revised manuscript. Figures 9-15 in the supplement to show the comparison with the full ion-chemistry results.

**2.3 Figure 17 and 19: Why is an interpolated colour scale used here? However, the fontsize in the axis used in these figures is the best of all the many contour panel figures and I encourage you to implement this in the above mentioned figures.**

A discrete colour scale is used for these figures in the revised version.

Now figures 21 and 23 in the revised version.

**2.4 All figures from Figure 4 onwards: Please consider the use of "_" in the figure titles, this does not help the reader.**

"_" has been removed from all the figure titles in the revised manuscript.

**2.5 1D ion-chemistry modelling of the Halloween event has been done in the past, but these studies (e.g. Verronen, et al (2005), Diurnal variation of ozone depletion during the October – November 2003 solar proton events, J. Geophys. Res., 110, A09S32, doi:10.1029/2004JA010932) were not discussed. The previous studies focused on here were those involving 3D models, with limited or no ion-chemistry. Some discussion on ion-chemistry studies would be relevant to add.**

The diurnal variation of $HO_x/NO_x$ as well as ozone depletion increase during solar proton events have been reported in Verronen et al. (2005). We have included discussions from the same and also details about the differences of the magnitude of depletion with respect to the diurnal cycle and altitude in Sect. 4.3. Regarding ion-chemistry studies, discussions were added, for example, from Kalakoski et al. (2020), also as a part of our response to referee 1.

I mentioned Kalakoski et al. (2020) regarding ion-chemistry studies but the discussions are mainly for the ion-chemistry impacts on $NO_x$ and ozone loss but not the diurnal variation of ozone as I wrote above in the reply.

Line 625-639: The magnitude of depletion with respect to the diurnal cycle and altitude has some considerable variation. Verronen et al. (2005) used a one dimensional chemical model, Sodankylä Ion and neutral Chemistry model (SIC) for ionospheric D-region studies. They studied the effects of ion-chemistry on the neutral atmosphere and also the diurnal variation of $NO_x/HO_x$ increases, as well as ozone depletions. This diurnal variation during a solar proton event or an energetic particle precipitation (EPP) event has been previously reported by Aikin and Smith (1999). Verronen et al. (2005) observed the diurnal cycle of $O_x=O+O_3$ depletion between 40 and 85 km. They found substantial ozone loss at sunset of 28 October and even greater loss at sunrise of 29 October followed by a recovery at 55–75 km during the noon and afternoon hours. The maximum depletion is reached just after sunset, with a 95% reduction in the $O_x$ values at 78 km. During daytime on 30 October, $O_x$ partly recovers but is again depleted during sunset. Rohen et al. (2005) also studied the Halloween SPE using SCHIAMACHY observations and considered 60°N magnetic latitude, which compares quite well to our 67.5°N geographic latitude. Since SCHIAMACHY measures only during daytime, they don't see a diurnal cycle in their results. With SCHIAMACHY, they reported a 20-30% ozone loss at 40-50 km in the Northern Hemisphere during the event and a 20-40% ozone loss, also during the event at 40-55 km observed by a photochemical model. This is related to the $NO_x$ catalytic cycle that was long lived. Above 50 km and at

380 higher altitudes, ozone recovery was faster after 50-60% loss during the event observed with SCHIAMACHY and the model which was due to the short lifetime of $HO_x$ and photolytical reproduction of ozone.

**2.6 Overall references should be added to where previous work is discussed. For example where in the text you discuss "Recent studies…." you need to actually include the references. One such example is the sentence on lines 39-40.**

385 We have added a couple of references overall throughout the different sections. In particular for this line 39-40, the reference Verronen et al. (2005) has been added.

**Line 46-47**: Recent studies, such as Verronen et al. (2005), that studied energetic particle precipitation events (EPP) found significant co-variability in mesospheric ozone.

**3 Minor Comments/typos**

**3.1 Pay attention to use of "Figure" and "figure" throughout the text.**

Only "figure" is used in the revised text except if it's at the beginning of a sentence.

**3.2 There are several mentions of "significant" as a measure in the text. What do you mean by significant, did you use some measure for this? For example in section 3.2 you talk about "significant low values", you probably mean clearly low values etc.**

395 In section 3.2, using "significant low values" for HOCl daytime was meant as a measure against MIPAS observations, as in "significantly low values compared to what MIPAS observed". However, we replaced it with "much lower values were observed...compared to MIPAS observations" here. And removed the word "significant" in some other places, for example, line 320, caption in figure 5 and line 438. However, for example, in line 502, it means noteworthy ozone depletion even though the event stops. So we didn't remove it there and some other places concerning ozone depletion.

400 **Line 383-384**: However, much lower values were observed during daytime for the model, setting $O(^1D)$ to photo-chemical equilibrium, without chlorine ion-chemistry and parameterised $NO_x$ and $HO_x$ compared to MIPAS observations.

**3.3 "Earth" (the planet) always starts with a capital letter.**

In the new version, we have used "Earth" everywhere in the text.

**Line 44-46**: This study involves SPEs which can also induce geomagnetic disturbances in the Earth's magnetosphere leading
405 to energetic electron precipitation (EEP) events.

**3.4 67.5°N. There should not be a space after the number and before the degree sign, or between the degree sign and N/S.**

The space after the number and before the degree sign has been removed in 67.5°N.

For every sentence, i.e. lines 14, 140, 188, 313, 323, 500 and 635 that contains 67.5°N, the space has been removed. And in figure captions of figure 1, figure 3 and figure 13.

**3.5 Check where you need to use comma before the word "which".**

We checked that and revised accordingly.

**3.6 Abstract: Last sentence: This currently reads as there was only 10/20% ozone loss and it was all due to chlorine chemistry. I know you mean that you specifically found that the inclusion of chlorine ion chemistry ADDED 10/20% to the ozone loss, so this sentence ghouls be revised.**

That is true. In figure 19, for the Halloween SPE and the extreme event, we calculated the additional ozone loss that was due to the already present chlorine ion-chemistry in the ion-chemistry model. We basically switched it off to assess that. The sentence has been restructured in the abstract accordingly.

Line 31-33: Furthermore, while comparing the Halloween SPE and the extreme scenario, with ionisation rate profiles applied just for the event day, the inclusion of chlorine ion-chemistry added an ozone loss of 10% and 20% respectively.

**3.7 Line 33: "..and HEAVIER nuclei"**

We included the word "HEAVIER" in this sentence.

Line 40: ..with protons and heavier nuclei..

**3.8 Introduction: SPEs are currently defined to mean Solar Particle Event and well as Solar Proton Event. In your case you I think you only mean the latter.**

In this case, we are mainly considering solar proton events. So yes, SPEs stand for that.

Line 2, 41 and 45: Solar particle events replaced to Solar proton events (SPEs)

**3.9 Can you comment on observed amounts of chlorine ions in the mesosphere? For example, how do we know that Cl- and Cl-(H2O) are the most abundant?**

Few older studies of chlorine ion-chemistry of the D-region like Turco (1977), Chakrabarty and Ganguly (1989) and Kopp and Fritzenwallner (1997) indicate that Cl- and Cl-(H2O) are the most abundant chlorine anions in the mesosphere for both night-time and daytime conditions. Some further explanations are also provided by Holger Winkler, Justus Notholt (2013). We included more details and revised the introduction.

**Line 105-107**: $Cl^-$ and $Cl^-(H_2O)$ are the most abundant chlorine ions in the mesosphere as indicated by previous studies by, e.g., Chakrabarty and Ganguly (1989), Fritzenwallner and Kopp (1998) and Turco (1997).

**3.10 Line 106: Shanklin (1985), remove the full stop after Shanklin and before bracket.**

The full stop after Shanklin and before bracket was removed.

**3.11 Lines 125-126: This sentence needs to be rearranged to that the three statements are in correct order: circumpolar cyclone, formed due to solar insolation, dominates circulation (rather than circulation, use dynamics).**

We have re-arranged the sentence in the revised manuscript accordingly.

**Line 154-157**: The polar vortex is a large circumpolar cyclone that is formed due to decreased solar insolation in the polar winter stratosphere as a manifestation of a strong meridional temperature gradient caused by a lack of high-latitude solar heating during polar night and dominates the dynamics.

**3.12 Line 185: Space before the reference.**

We have added that.

**Line 224**: ..which can also be used as a parameterisation for global chemistry-climate models (Nieder et al., 2014)

**3.13 Section 2.2.2 and conclusions: So why is the rate of O(1D) too large and how would you assure this was not an issue if the ion chemistry was applied to a climate model? That is, can you offer a working solution from the results of this paper which you could state in the conclusions?**

In the ExoTIC model, the ions are brought into photo-chemical equilibrium. And for the neutrals, the rate of formation is then calculated from the ion-chemistry. The neutrals in the ion-chemistry are long lived with the exception of $N(^2D)$ and $O(^1D)$. $N(^2D)$ is treated like one of the ions, so it is not transferred to the neutral chemistry but is brought into equilibrium. $O(^1D)$ however is not brought into equilibrium and hence the rate of formation is large. So if $O(^1D)$ is put into equilibrium, it is definitely formed in the ion-chemistry and then the rate of formation is zero. For a climate model, at least in the chemistry climate model EMAC, the ion-chemistry is part of the normal chemistry solver and in that case, the $O(^1D)$ formation rate should be not too large. So the ion-chemistry implementation should work without doing anything to $O(^1D)$. We have revised the "conclusions" section and discussed some implications and future work.

Now **Sect. 2.2.1 and Line 243-245**: The simulation results are sensitive to the changes of the $O(^1D)$ branching ratio, $\beta = \Delta O(^1D)/(\Delta O(^1D) + \Delta O(^3P))$ which is discussed in Winkler et al (2009). Winkler et al (2011) reported the $O(^1D)$ corrections in the UBIC model in detail.

**Line 745-756**: While previous results with UBIC focused on the solar proton event in July 2000 in the northern polar region and compared to the HCl measurements from Halogen Occultation Experiment instrument (HALOE), we compare with MIPAS observations. Since MIPAS observations provide a better picture of the polar cap region as compared to HALOE observations that are less densely sampled, these results suggest that the validated D-region ion-chemistry setup in the ExoTIC model can be trusted to implement in a global 3D model. The problem of $O(^1D)$ in the ion-chemistry in ExoTIC should be taken into account which was however related to the neutral chemistry model as explained in Sect. 2.2.1. For a global 3D chemistry climate model, at least in EMAC (ECHAM/MESSy), that we are considering for the implementation, the ion-chemistry is part of the normal chemistry solver and in that case, the $O(^1D)$ formation rate should be not too large and it should work without doing anything to $O(^1D)$. This setup will be first evaluated with MIPAS observations and since EMAC can already provide the data at the MIPAS footprints, the modelled data can be sampled at MIPAS measurement local times. The model will then considered for experiments in different setups to look at the dynamical impacts with and without the D-region ion-chemistry with important chemical reactions involving water, chlorine and $NO^+$ cluster ions.

**3.14 Averaging kernels: The text uses both A and AK, figures use AK. Please change these so that the use is consistent.**

We have changed it to A everywhere in the text and figures.

In the title of the second columns of figures 5-12 and x-axis labels and caption of figure 3 in the revised version.

**3.15 Section 3: Using the averaging kernels: Do you mean that the model is first sampled only at MIPAS altitude grid locations, then averaging kernels are applied. This just need a clarification in the text where currently it's a bit mixed up. Same section, you need a reference here for the MIPAS tracer aspects.**

Yes, the model is first sampled only at the MIPAS altitude grid locations and then averaging kernels are applied. We have clarified that in the text. The references for the MIPAS tracer aspects are present in Sect. 2.3. But we have also added them in Sect. 3.

**Line 331-332**: The averaging kernels were applied after sampling the model data in the MIPAS altitude grid.

**Line 368-370**: A comparison of Envisat MIPAS V5 HOCl measurements (von Clarmann et al., 2006) for the polar Northern Hemisphere (57-77°N) and ExoTIC computations with different settings of HOCl simulation is presented in figure 5 and 6 for daytime and nighttime respectively.

**Line 410-411**: Figures 7 and 8 show the daily averaged absolute differences for Envisat MIPAS V5 (Hoepfner et al., 2007a) and modelled chlorine nitrate w.r.t. 26th October 2003 for daytime and nighttime.

: In order to assess the agreement of the ENVISAT MIPAS V5 (Funke et al., 2005) and modelled SPE related odd nitrogen enhancements, total $NO_y$ (=$NO + NO_2 + HNO_3 + 2N_2O_5 + ClONO_2$) is compared.

: Figures 11 and 12 shows ENVISAT MIPAS V5 (Glatthor et al., 2006) and modelled temporal evolution of the relative ozone changes w.r.t. $26^{th}$ October, averaged over 70-90°N for daytime and nighttime respectively.

**3.16 Page 13: numbered point 1. What do you mean by this? numbered point 2. degrees North. numbered point 3. What do you mean by the model temperature profile being fixed? Fixed the whole time?**

Numbered point 1: It means we selected the model profiles from the entire time series, one at a time. Numbered point 2: Yes that is correct. 57.5 to 77.5 degrees North. Numbered point 3: The model temperature profile is fixed for the entire time series. It doesn't change with time. In the revised version, we did some rewording.

: The model profiles were selected, one at a time, from the entire time series.

: All the profiles from MIPAS within 57.5 and 77.5 degrees N latitude and +/- 6 hours of the model profile's time were selected.

**3.17 Section 3.1. Lines 294-295: This needs rearranging or clarifying.**

We restructured the sentence and also added more details on why CO was used as a tracer of vortex air.

: Here, we have used a CO vmr threshold (discriminating mesospheric air from the background) as vortex criterion. We need an altitude-independent criterion (as we need to differentiate entire profiles) and our short time period (a few days) allows for a time-independent definition. A chemical vortex definition (via CO) is also widely used in the mesosphere (Harvey et al. 2015). And for the chemical species and the SPE responses discussed here, the relevant altitude range is more in the stratosphere. In the stratosphere, the vortex might be considerably smaller and is determined from the potential vorticity. Here, we use the tracer gradient instead to be consistent with the MIPAS observations that we use for comparison.

**3.18 Section 3.1. Later you talk about latitude bands relating to vortex edge and deep in the vortex, but you never actually define these regions. Please add the relevant information and how these were determined here.**

We defined the regions "edge of the vortex" as 57-77N and "deep in the vortex" as 70-90N. The boundary of the "edge of the vortex" was determined using a fixed threshold of CO that was 0.5 ppm. More discussion is added in the revised manuscript in Sect. 3.1 explaining how we determined the vortex boundary where we also defined these regions.

: One can also check the vortex boundary by looking at $CH_4$ zonal means, for example, Figure 9 of Funke et al. (2011). From that figure and as also discussed in Funke et al. (2011), the boundary is around 60°N which also works well for the stratosphere. We chose the latitude 57°N as to where the vortex begins and defined the latitude bands 57-77°N as "the edge region of the vortex" and the high latitude bands 70-90°N as "deep in the vortex".

**3.19 Section 3.2. Line 310: Initially, MIPAS looks to be much higher. However, this is a good example where the current figure is difficult to read with the changing colour scale and multiple panels. After careful examination the statement in the text is probably correct, but you need to clarify the figures.**

Line 310: With full ion-chemistry, the original model results are much higher compared to MIPAS for both day and night. But for the other three sensitivity runs, for daytime, MIPAS is higher and for night-time, model shows higher values. We have updated the figures keeping the same color scale for the vertical panels and also added the colorbar intervals in the figure captions.

Figures 5-12 in the revised version.

**3.20 Section 3.2. Line 315-316: I don't see this removal of a peak in the figure.**

The word 'peak' is probably not right to use here. What we meant was that there was excess HOCl observed in the mesosphere around 60 km for the night-time plots with ion-chemistry considering $O(^1D)$ in photo-chemical equilibrium. This excess value just disappears when the chlorine ion-chemistry is switched off. We have removed the word 'peak' and restructured the sentence.

Line 381-382: Switching off the chlorine ion-chemistry led to the removal of 0.2 ppb HOCl observed in the mesosphere during nighttime.

**3.21 Section 3.3: I do not understand where the 57N-77N = "vortex edge" and 70N-90N = "deep in the vortex" comes from. This is something you need to clarify in the earlier section according to my previous comment. The overlapping latitudes are particularly strange here.**

As you pointed out before, we have clarified this in the revised version in Sect. 3.3. The latitude bands do overlap but we defined the regions "vortex edge" and "deep in the vortex" based on the experiments conducted with different samplings of the latitude bands for the zonal averages. This was done as an attempt to check which sampling works better for the different chemical species. For HOCl and $ClONO_2$, using a sampling of 57-77N showed better agreements with the model results with averaging kernels applied, whereas for ozone 70-90N worked better.

Line 365-366: We chose the latitude 57°N as to where the vortex begins and defined the latitude bands 57-77°N as "the edge region of the vortex" and the high latitude bands 70-90°N as "deep in the vortex".

**3.22 Lines 387-388: Where does this arise from? Adding a reference could be sufficient.**

We added some references for this in the revised version, for example, Funke et al. 2011, https://doi.org/10.5194/acp-11-9089-2011.

Line 462-464: The results are shown upto 50 km since above 50 km NO is the largest contributor (Funke et al., 2011) and averaging kernels are not applied to NO. Another reason is that large uncertainties of small vmr values of $ClONO_2$ (Höpfner et al., 2007b) will spoil the $NO_y$ sum and it's uncertainty.

**3.23 Lines 395-396: You need to include references here.**

We have added a citation Meraner et al. 2018, https://doi.org/10.5194/acp-18-1079-2018 in this sentence.

**Line 472-473**: As inferred from observations, stratospheric ozone decreases due to the indirect effect of EPP by about
10–15% as observed by satellite instruments (Meraner and Schmidt, 2018).

**3.24 Lines 401-402: Does this information come from examination of the reaction rates, or from a reference (not included)?**

The loss in the mesosphere being related to HOx catalytic cycle is taken from references, Funke et al. (2011) and Bates and Nicolet et al. (1950).

**Line 479-480**: A loss of 60-75% is observed during the event itself in the mesosphere that is short lived and is related to the HOx catalytic cycle (Reactions R6, R7, R9 and R10) (Funke et al., 2011)(Bates and Nicolet, 1950).

**3.25 Lines 410-412: Why do you expect the AISstorm ionisation rates being different to real ones for the event? Particularly the precipitating proton fluxes are were well known for the Halloween event.**

The proton fluxes for the Halloween event are known from different satellites indeed, for example, GOES 10 and GOES 11. However, AISstorm here uses GOES 10 and the ionisation rates should be a lower estimate. The SPE ionization is already happening in the denser atmosphere, therefore the conversion from particle fluxes into ionization should be more or less precise in terms of total ionization and altitude. The main uncertainty of SPE ionization in AISstorm is the size of the area that is affected by high energetic particle precipitation. This cannot be derived from these channels but is taken from a lower energy channel on another satellite and thus it might be an underestimate of the polar cap size. However if that would be important for the spatial average 70-90N, we should see an underestimation of the ionization (and NOy) as well, which doesn't seem to be the case (and which is unlikely anyway as the equatorward boundary should be at about 60N).

**Line 490-497**: AISstorm here uses proton fluxes from GOES 10 and the ionisation rates should be a lower estimate. The SPE ionisation happens in the denser atmosphere, therefore the conversion from particle fluxes into ionisation should be more or less precise in terms of total ionisation and altitude. The main uncertainty of SPE ionisation in AISstorm is the size of the area that is affected by high energetic particle precipitation. This cannot be derived from these channels but is taken from a lower energy channel on another satellite and thus it might be an underestimate of the polar cap size. However if that would be important for the spatial average 70-90°N, we should see an underestimation of the ionisation (and NOy) as well, which doesn't seem to be the case (and which is unlikely anyway as the equatorward boundary should be at about 60°N.

**3.26 Section 4. What latitude is modelled here?**

It is the same latitude of 67.5°N as in the previous section.

**Line 499-500**: The model simulations are performed at a latitude of 67.5°N and begin at the noon of $27^{th}$ October (day 300).

**3.27** **Why are both NOy and HOx defined different to previous sections of the paper for this comparison? Why not use the same definition (the initial ones) throughout? Seems like the macron was added to NOy to distinguish it from the previous definition, but the same was not done for HOx?**

$NO_y$ was indeed defined using a macron in this section as $\tilde{NO}_y$ because we wanted to add N and $NO_3$ to $\tilde{NO}_y$ for the Halloween SPE and extreme event comparison. Since MIPAS doesn't measure these species of N and $NO_3$, we used a different definition of $NO_y$ in the previous section. $HO_x$ studies are mainly shown in section 4 and wasn't calculated for section 3 due to the same reason as H and OH not being provided by MIPAS.

Line 534: $HO_x$ was not shown for the MIPAS comparison due to the fact that H and OH are not provided by MIPAS.

**3.28** **Lines 445-446: The results for the full ion-chemistry model results are not shown so we can't see where this comparison result comes from.**

We didn't include the full ion-chemistry results because we wanted to mainly show the sensitivity studies that compared best to the MIPAS observations. But discussions of the full ion-chemistry results are included in the text. We added the full ion-chemistry results along with the other sensitivity runs as a comparison in the supplement to the revised manuscript.

Full ion-chemistry results are for the Halloween SPE and extreme event comparison are in the supplement: Figures 9-15.

**3.29** **Lines 463-464: This is repetition of previous sentences, remove.**

This sentence has been removed.

**3.30** **Figure 14: Looking at this figure, it would seem that the chlorine ion chemistry is not important for ClONO2? Perhaps you can comment in the text?**

The chlorine ion-chemistry plays a small role for $ClONO_2$ around 45-50 km during the event as seen from this figure. However the impact is indeed very small. We have clarified this in the text.

Lines 592-594: The chlorine ion-chemistry plays a small role for $ClONO_2$ around 45-50 km specially during nighttime on day 301. The production of $ClONO_2$ when the chlorine ion-chemistry is switched off is comparatively higher at that altitude.

**3.31** **Figure 15-16: It's impossible to see any detail in these figures. I think you should be able to easily clarify these by making lines thicker and adjusting the y-scale. The labelling font is far too small.**

The figures are updated with thicker lines and larger labelling font. The y-scale in figure 15 is the same for the top and the bottom row for each column (species). The axes font are also enlarged.

Now figures 19 and 20 in the revised version.

**3.32** **Lines 488-489: Not clear. Perhaps: "Because there are not enough chlorine atoms during nighttime…"?**

Yes, because there are not enough chlorine atoms during night-time.

 This is due to the fact that during nighttime the concentration of free chlorine atoms is typically low since the primary source of these atoms is the dissociation of chlorine-containing reservoir species, such as chlorine nitrate ($ClONO_2$). $ClONO_2$ occurs predominantly during daytime due to the presence of sunlight, where it is photolysed to release chlorine atoms and hence at sunrise this renewed increase in chlorine atoms results in a subsequent increase in HCl levels.

**3.33 Section 4.3: The discussion of ozone loss in the mesosphere and stratosphere is a bit mixed. Perhaps you can clearly distinguish between the HOx and NOx driven losses to clarify this section. In this section it would also be useful to contrast the Halloween ozone loss to previously published model results (e.g. Verronen et al. I mentioned earlier) as well as observations since this is a very well studied event.**

We have included discussion from Verronen et al. (2005) who studied the diurnal cycle of ozone loss during the Halloween SPE using the Sodankylä Ion Chemistry Model (SIC) and Rohen et al. (2005) who studied it using SCHIAMACHY observations. The HOx and NOx driven losses in the mesosphere and stratosphere are also clarified and revised. Details about the differences of the magnitude of ozone depletion with respect to the NOx and HOx related loss and altitude have been included as well.

 The magnitude of depletion with respect to the diurnal cycle and altitude has some considerable variation. Verronen et al. (2005) used a one dimensional chemical model, Sodankylä Ion and neutral Chemistry model (SIC) for ionospheric D-region studies. They studied the effects of ion-chemistry on the neutral atmosphere and also the diurnal variation of $NO_x$/$HO_x$ increases, as well as ozone depletions. This diurnal variation during a solar proton event or an energetic particle precipitation (EPP) event has been previously reported by Aikin and Smith (1999). Verronen et al. (2005) observed the diurnal cycle of $O_x = O + O_3$ depletion between 40 and 85 km. They found substantial ozone loss at sunset of 28 October and even greater loss at sunrise of 29 October followed by a recovery at 55–75 km during the noon and afternoon hours. The maximum depletion is reached just after sunset, with a 95% reduction in the $O_x$ values at 78 km. During daytime on 30 October, $O_x$ partly recovers but is again depleted during sunset. Rohen et al. (2005) also studied the Halloween SPE using SCHIAMACHY observations and considered 60°N magnetic latitude, which compares quite well to our 67.5°N geographic latitude. Since SCHIAMACHY measures only during daytime, they don't see a diurnal cycle in their results. With SCHIAMACHY, they reported a 20-30% ozone loss at 40-50 km in the Northern Hemisphere during the event and a 20-40% ozone loss, also during the event at 40-55 km observed by a photochemical model. This is related to the $NO_x$ catalytic cycle that was long lived. Above 50 km and at higher altitudes, ozone recovery was faster after 50-60% loss during the event observed with SCHIAMACHY and the model which was due to the short lifetime of $HO_x$ and photolytical reproduction of ozone. In our case, the continuing ozone loss at 40-55 km, related to the diurnal variation of $NO_x$ is found to be 60-80% for the extreme scenario as compared to the Halloween event which is just around 20%.

**3.34 Section 5: Lines: 516-517: Why is this expected?**

We believe that this is related to the diurnal cycle of ClO which is different for daytime and night-time. During the event, ClO formation is seen for night-time at 50-70 km whereas it is not present during day-time. Explanations regarding this are added in the revised manuscript in section 5 and a figure for the same in the supplement (Figure 15).

650     It's figure 17 now in the supplement.

**Line 672-681**: This difference can be explained by the difference of the diurnal cycle of ClO during daytime and nighttime. The catalytic ozone loss cycles relevant in the stratosphere-mesosphere are the ClO+O (Reactions R15, R16 and R17) and ClO+HO$_2$ (Reactions R18, R21 and R19) that also need solar light, since O is formed by photolysis. During daytime, ClO

655 photolyses in the mesosphere but not in the stratosphere, so ClO is not observed in the mesosphere. The ClO+HO$_2$ cycle produces HOCl which also photolyses during daytime producing Cl through Reaction R20. So during daytime, Cl is more important than ClO. But during nighttime, ClO accumulates in the mesosphere and in the stratosphere, it is mainly ClONO$_2$ due to Reaction R32. As we see from our results, during the event day on October 28 and also on October 29, the ClO mixing ratios were found to be higher for night-time around 60-70 km compared to daytime. Hence, the ozone loss occurs more in

660 the upper stratosphere-mesosphere around 50-70 km for nighttime, thereby producing the difference of the ozone loss in the altitude range.

**3.35 End of Conclusions: Based on your work, can you discuss lessons learned that should be considered for inclusions of D-region ion chemistry in EMAC.**

We have revised section 6 discussing the implications and recommendations of the work. In a nutshell, the validated D-region

665 ion-chemistry in ExoTIC consists of important chemical reactions involving water, chlorine and NO$^+$ cluster ions. We are implementing these reactions from ExoTIC into EMAC and also using the same ion pair production rates that are based on Porter, J. N. (1976). One lesson that we learned from our ion-chemistry model is to consider O($^1$D) in photo-chemical equilibrium. But for the global chemistry climate model EMAC, this shouldn't make much of a difference as also discussed in the previous comment "section 2.2.2 and conclusions". The next step would be to evaluate the performance of the ion-chemistry

670 comparing it with MIPAS observations and considering that for experiments involving atmospheric dynamics.

**Line 745-756**: While previous results with UBIC focused on the solar proton event in July 2000 in the northern polar region and compared to the HCl measurements from Halogen Occultation Experiment instrument (HALOE), we compare with MIPAS observations. Since MIPAS observations provide a better picture of the polar cap region as compared to HALOE observations that are less densely sampled, these results suggest that the validated D-region ion-chemistry setup in the ExoTIC model can

675 be trusted to implement in a global 3D model. The problem of O($^1$D) in the ion-chemistry in ExoTIC should be taken into account which was however related to the neutral chemistry model as explained in Sect. 2.2.1. For a global 3D chemistry climate model, at least in EMAC (ECHAM/MESSy), that we are considering for the implementation, the ion-chemistry is part of the normal chemistry solver and in that case, the O($^1$D) formation rate should be not too large and it should work without doing anything to O($^1$D). This setup will be first evaluated with MIPAS observations and since EMAC can already provide

680 the data at the MIPAS footprints, the modelled data can be sampled at MIPAS measurement local times. The model will then considered for experiments in different setups to look at the dynamical impacts with and without the D-region ion-chemistry with important chemical reactions involving water, chlorine and NO$^+$ cluster ions.

**4 Other relevant additions in the manuscript**

**4.1 Abstract**

685 **Line 8-11 (Regarding a sentence on the novelty of the work)**: The Halloween SPE that occurred in late October 2003 is used as a test field for our study. This event has been extensively studied before using different 3D models and satellite observations. Our main purpose is to use such a large event that has been recorded by MIPAS on ENVISAT to evaluate the performance of the ion-chemistry model.

690 **Line 14-17**: Comparison of the simulated effects against MIPAS observations for the Halloween SPE revealed a rather good temporal agreement, also in terms of altitude range for HOCl, $O_3$ and $NO_y$. For $ClONO_2$, a good agreement was found in terms of altitude range. The model showed $ClONO_2$ enhancements after the peak of the event.

**Line 23-25**: With the model applied to this scenario, assessment can be made what is to be expected at worst for effects of a 695 SPE on the middle atmosphere concentrating on effects of ion-chemistry compared to crude parameterisations.

**4.2 Section 2.1: Ionisation rates**

**Line 191-192**: Cliver et al. (2022) estimated this factor 70 $\times$ particle fluence compared to the 1956 event and the ionisation rates were scaled accordingly.

**4.3 Section 3.4: Odd oxides of nitrogen ($NO_y$)**

700 **Line 459-462**: The MIPAS observations showed a production of 30-40 ppb in the upper stratosphere during nighttime and 20-30 ppb during day-time. The MIPAS observations show much higher dynamics than ExoTIC for both day-time and nighttime. The nighttime values show stronger dynamics with decreased ionisation rates on day 305-306 and connected to this less $\Delta NO_y$ reaches a constant state after the maximum of the first SPE on day 303. This could be an indication that some of the $NO_y$ recombination speeds are faster than expected.

705 ## 4.4 Section 5: Impact of chlorine ions on ozone loss: Regarding the increase in ozone due to chlorine

**Line 685-687**: This increase is linked to enhanced atomic oxygen production by $O_2$ photolysis in solar maximum conditions (Marsh et al., 2007). It is observed at an altitude range of 60-70 km for daytime and 50-70 km for nighttime.